**Integrating field, textural and geochemical monitoring to track eruption triggers and**
**dynamics: a case-study from Piton de la Fournaise**
Lucia Gurioli[1], Andrea Di Muro[2], Ivan Vlastélic[1], Séverine Moune[1], Simon Thivet[1],
Marina Valer[1], Nicolas Villeneuve[2,3], Guillaume Boudoire[2,3], Aline Peltier[2], Patrick
Bachèlery[1], Valerie Ferrazzini[2], Nicole Métrich[2], Mhammed Benbakkar[1], Nicolas
Cluzel[1], Christophe Constantin[1], Jean-Luc Devidal[1], Claire Fonquernie[1], Jean-Marc
Hénot[1]
(1) Université Clermont Auvergne, CNRS, IRD, OPGC, Laboratoire Magmas et Volcans, F-63000
Clermont-Ferrand, France
(2) Institut de Physique du Globe (IPGP), Sorbonne Paris-Cite, CNRS UMR-7154, Université Paris
Diderot, Observatoire Volcanologique du Piton de la Fournaise (OVPF), Bourg Murat, France,
(3) Laboratoire Géosciences Réunion, Université de La Réunion, Institut de Physique du Globe de
Paris, Sorbonne Paris-Cité, UMR 7154 CNRS, F-97715 Saint-Denis, France
Corresponding author: L Gurioli, Université Clermont Auvergne, CNRS, IRD, OPGC, LMV
Campus Universitaire des Cézeaux, 6 Avenue Blaise Pascal, 63178 Aubière Cedex
(lucia.gurioli@uca.fr)
**Abstract**
The 2014 eruption at Piton de la Fournaise (PdF), La Réunion, which occurred after 41
months of quiescence, began with surprisingly little precursory activity, and was one of the
smallest so far observed at PdF in terms of duration (less than 2 days) and volume (less than
$0.4 \times 10^6$ m$^3$). The pyroclastic material was composed of golden basaltic pumice along with
fluidal, spiny-iridescent and spiny-opaque basaltic scoria. Density analyses performed on 200
lapilli reveal that while the spiny-opaque clasts are the least dense (1600 kg/m$^3$) and most
crystalline (55 vol. %), the golden pumices are the least dense (400 kg/m$^3$) and crystalline (8
vol. %). The connectivity data indicate that the fluidal and golden (Hawaiian-like) clasts have
more isolated vesicles (up to 40 vol. %) than the spiny (Strombolian-like) clasts (0-5 vol. %).
These textural variations are linked to primary pre-eruptive magma storage conditions. The
golden and fluidal fragments track the hotter portion of the melt, in contrast to the spiny
fragments and lava that mirror the cooler portion of the shallow reservoir. Exponential decay
of the magma ascent and output rates through time revealed depressurization of the source

during which a stratified storage system was progressively tapped. Increasing syn-eruptive degassing and melt-gas decoupling led to a decrease in the explosive intensity from early fountaining to Strombolian activity. The geochemical results confirm the absence of new input of hot magma into the 2014 reservoir and confirm the emission of a single, shallow, differentiated magma source, possibly related to residual magma from the November 2009 eruption. Fast volatile exsolution and crystal-melt separation (second boiling) were triggered by deep pre-eruptive magma transfer and stress field change. Our study highlights the possibility that shallow magma pockets can be quickly reactivated by deep processes without mass or energy (heat) transfer and produce hazardous eruptions with only short term elusive precursors.

**Key words**: Piton de la Fournaise, Hawaiian activity, Strombolian activity, shallow reservoire, texture, petrology, geochemistry

## 1. Introduction

A detailed characterization and understanding of eruptive dynamics and of processes driving and modulating volcano unrest is crucial in monitoring active volcanoes and fundamental for forecasting volcanic eruptions (Sparks, 2003). Many studies suggest that eruptive phenomena are strongly dependent on the physico-chemical properties of ascending magma in the conduit (e.g., temperature, viscosity, porosity, and permeability) (e.g. Sparks, 1978; Rust and Cashman, 2011; Gonnermann and Manga, 2013; Polacci et al., 2014). Integrating petrographic, chemical and textural data can thus provide critical information to constrain both the pre-eruptive storage conditions, and the processes related to magma ascent, degassing and cooling (e.g., reference in Table 1 in Gurioli et al., 2015). This multidisciplinary approach is of even greater importance in the monitoring of volcanoes which emit relatively uniform magma compositions over time, like basaltic volcanoes (e.g. Di Muro et al., 2014; Gurioli et al., 2015; Coppola et al., 2017). As a result, monitoring of textures, and petrochemical properties of lava fragments and pyroclasts is now routinely carried out on a daily basis at active volcanoes such as Kilauea, Etna, and Stromboli (e.g., Taddeucci et al., 2002; Thornber et al., 2003; Polacci et al., 2006; Swanson et al., 2009; Colo' et al., 2010; Houghton et al., 2011; 2013; 2016; Carey et al., 2012; 2013; Lautze et al., 2012; Andronico et al., 2013a; b; 2014; Corsaro and and Miraglia, 2014; Di Muro et al., 2014; Gurioli et al.; 2014; Eychenne et al., 2015; Leduc et al., 2015; Kahl et al., 2015). In the past, time series of petrographic and geochemical data have been measured for PdF basalts and particularly for effusive products.

The aim of these datasets was to constrain the spatial and temporal evolution of magma for one of the most active basaltic volcanoes of the world (e.g. Albarède et al., 1997; Vlastélic et al., 2005; 2007, 2009; Boivin and Bachèlery, 2009; Peltier et al., 2009; Schiano et al., 2012; Lénat et al., 2012; Di Muro et al., 2014; 2015; Vlastèlic and Pietruszka, 2016). However, this type of approach has seldom been coupled with detailed textural studies at PdF and instead has mostly focused on crystal textures and crystal size distribution (Welsch et al., 2009; 2013; Di Muro et al., 2014; 2015). Moreover, only sporadic data exist on the textures of pyroclasts ejected by the eruptions at PdF (Villemant et al., 2009; Famin et al., 2009; Welsch et al., 2009; 2013; Michon et al., 2013; Vlastélic et al., 2013; Di Muro et al., 2015; Morandi et al., 2016; Ort et al., 2016).

Within this paper, we present a multidisciplinary textural, chemical and petrological approach to quantify and understand the short-lived 2014 PdF eruption. This approach combines detailed study of the pyroclastic deposit (grain size and componentry) with bulk texture analysis (density, vesicularity, connectivity, permeability, morphology, vesicle distribution and crystal content) and a petro-chemical study (bulk rock, glass, minerals, melt inclusions) of the same clasts. This integrated approach has now been formalized within the French National Observation Service for Volcanology (SNOV), as routine observational systems (DynVolc), Dynamics of Volcanoes, (http://wwwobs.univ-bpclermont.fr/SO/televolc/dynvolc/) and GazVolc, Observation des gaz volcaniques, (http://wwwobs.univ-bpclermont.fr/SO/televolc/gazvolc) to provide data for the on-going activity at PdF (Harris et al., 2017).

In spite of being the first of a series of eruptions, the June 2014 event was preceded by only weak inflation and by a rapid increase in number of shallow (< 2 km below volcano summit) volcano tectonic earthquakes that happened only 11 days before the eruption (Peltier et al., 2016). The eruptive event was dominantly effusive, lasted only 20 hours and emitted a very small volume of magma (ca. 0.4 x $10^6$ m$^3$, Peltier et al., 2016), which makes this event one of the smallest, in terms of duration and volume, observed at PdF up to now. In addition, the eruption started during the night and very little direct observation exists for the first few hours of the activity, when the lava effusion was associated with very weak fountaining activity and Strombolian explosions.

This eruption occurred just outside the southern border of the summit Dolomieu caldera, at the top of the central cone of PdF (Fig. 1). This is a high risk sector because of the high number of tourists. Identification of precursors of this kind of activity represents an important challenge for monitoring systems (Bachélery et al., 2016).

Therefore this eruption represents an ideal context to apply our multidisciplinary

approach, with the aim of addressing the following key questions:

(i)      why was such a small volume of magma erupted instead of remaining

endogenic?

(ii)     what caused the rapid trigger and the sudden end to this small volume

eruption?

(iii)    which was the source of the eruption (shallow versus deep, single versus

multiple small magma batches)?

(iv)    what was the ascent and degassing history of the magma?

(v)     what was the time and space evolution of the eruptive event?

Furthermore, this eruption provides an exceptional opportunity to study processes leading to
the transition from mild Hawaiian (<20 m high fountains, following the nomenclature
proposed by Stovall et al., 2011) to Strombolian activity (<10 m high explosions), whose
products are little modified by post-fragmentation processes because of the very low intensity
of the activity.

**2 The 2014 activity**

**2.1 Precursory activity**

The 20 June 2014 summit eruption represents the first eruption at PdF after 41 months of
quiescence. The last eruption had been on 9 December 2010, with a shallow (above sea level)
intrusion on 2 February 2011 (Roult et al., 2012). From 2011, the deformation at PdF was
constant with two distinct types of behaviour: (i) a summit contraction of a few centimetres
every year (Fig. 1d), and (ii) a preferential displacement of the east flank at a rate of 1-3
centimetres per year (Brenguier et al., 2012; Staudacher and Peltier, 2015). The background
microseismicity was very low (< 5 shallow events/day below volcano summit) and low-
temperature summit intracaldera fumaroles emitted very little sulphur ($H_2S$ or $SO_2$) and
carbon ($CO_2$) (Di Muro et al., 2016). After 41 months of rest, a new intense cycle of activity
(June 2014, February 2015, May 2015, July 2015, August-October 2015; May 2016;
September 2016; January 2017 and July 2017) began with surprisingly little and ambiguous
precursory activity.

The 2014 summit eruption started during the night of June 20/21, at 21h35 GMT

(0h35 local time) and ended on June 21 at 17h09 GMT (21h09 local time), after less than 20
hours of dominantly effusive activity. The volcano reawakening was preceded, in March and

April 2014, by deep (15-20 km below sea level) eccentric seismicity and increase in soil $CO_2$ flux below the western volcano flank, 15 km NW of the volcano summit (Liuzzo et al., 2015; Boudoire et al., 2017). Background micro-seismicity and inflation of the central cone increased progressively starting on 9 June 2014. Weak inflation recorded on both distal and summit baselines (Fig. 1d) suggest that deep (below sea level) magma up-rise was pressurizing the shallow (above sea level) magma storage system (Peltier et al., 2016). On June 13, 17 and 20, three shallow (hypocentres located above sea level) intense seismic crises occurred below the summit Dolomieu caldera (Fig. 1), with hundreds of events located in a narrow depth range between 1100 and 2100 metres below the volcano summit. These seismic crises consisted of swarms of low magnitude (M: 1-2) volcano tectonic events which increased in number from the first to the third crisis. On June 20, seismicity increased progressively and a final seismic crisis started at 20h20 GMT, only 75 minutes before the eruption. This last seismic crisis was coupled with acceleration in the deformation of the summit area, which began only 60 minutes before the eruption. Interestingly, only slight inflation of the central cone (< 2 cm of dilatation) was detected 11 days before the 2014 eruption with a maximum of 1 cm and 1.6 cm enlargement at the summit and the base of the cone, respectively (Peltier et al., 2016 and Fig. 1d). A moderate increase in $CO_2$ and $H_2S$ emissions from summit intracaldera fumaroles was detected starting on June 2, but only very minor $SO_2$ emissions occurred before the eruption (mostly on June 7 and 15, unpublished data). Therefore, the acceleration in both geophysical and geochemical parameters was mostly related to the late phase of dyke propagation towards the surface just before the eruption. Following the end of the June 20-21 eruption, a long-term continuous inflation of the edifice began, at a moderate rate, and mostly at the base of the volcano. More than one year after this first eruption, the long-term deformation trends showed that the 2014 eruption marked a kink between the deflation trend which followed the caldera-forming 2007 eruption (Staudacher et al., 2009) and the currently ongoing continuous inflation trend (Fig. 1d, and Peltier et al., 2016; Coppola et al., 2017).

**2.2 Chronology of the events**

We reconstructed the chronology of the events by combining a distribution map of the fissures, pyroclastic deposits and lava flows (Fig. 1) with a review of available images and videos extracted from the observatory data base, the local newspapers, and web sites (Fig. 2). The 2014 eruption occurred at the summit and on the SE slopes of the Dolomieu Caldera (Figs. 1a, 1b and 1c) and evolved quickly and continuously over 20 hours. The full set of

fractures opened during a short period of time (minutes) and emitted short (<1.7 km long)
lava flows (Fig. 1 and Figs. 2c and 2d). Feeding vents were scattered along a 0.6 km long
fissure set (Fig. 1a) and produced very weak (low) Hawaiian to Strombolian activity (Fig. 2).
Fissures opened from west to east, initially sub-parallel to the southern border of
Dolomieu caldera and then propagated at lower altitude (Fig. 1). The summit part of the
fractures (ca. 2500 m asl, Western Fracture, WF in Fig. 1) emitted only small volumes of lava
and pyroclasts. This part of the fracture set was active only during the first few hours of the
eruption, at night. The eastern part of the fractures (Upper Fracture, UF in Fig. 1) descended
to lower altitude (between 2400 and 2300 m asl, Middle Fracture, Fig. 1) along the SE flank
of the summit cone and emitted most of the erupted volume. As often observed in PdF
eruptions, the activity progressively focused on a narrow portion of the fractures at low
altitude and finally on a single vent located at the lower tip of the fracture system (Main Vent,
at 2336 m asl, MV in Figs. 1, 2). The first in situ observations in the morning of June 21 (ca.
04h00 GMT) showed that weak Strombolian activity (Figs. 2a and 2b) was focused on a
narrow segment of the lower fractures and that a'a lavas had already attained the elevation of
1983 m asl (0.2 km before maximum runout, Fig. 2c). A small, weak gas plume was also
blowing northwards. A single sample of partially molten lava was collected from the still
active lava front and partially water quenched (REU140621-1, Table S1, Fig. 2d). During
most of June 21, the activity consisted of lava effusion in three parallel lava streams (Fig. 2c)
merging in a single lava flow (Fig. 2e) and mild-weak "Strombolian" explosions at several
closely spaced spots along the lower part of the feeding fracture. At 13.00 (GMT), only weak
explosions were observed within a single small spatter cone (Figs. 2f and 2g). Most of the
lava field was formed of open channel a'a lavas. The total volume of lava was estimated by
MIROVA service (https://www.sites.google.com/site/mirovaweb/home), with the use of the
MODIS images and the analyses of the flux from the spectral properties, to be within 0.34 +/-
0.12 x $10^6$ m$^3$, (Coppola et al., 2017). Satellite derived volume estimates are consistent with
independent photogrammetric estimates ($0.4 \pm 0.2$ x $10^6$ m$^3$; Peltier et al., 2016) and rank the
2014 eruption at the lower end of the volume range typically emitted by PdF (Roult et al.,

2012).

**3. Methodology**
**3.1 Sampling strategy**

Apart from the sample from the front of the still active lava flow (Fig. 2d), all other samples were collected in two phases: 3 days (pyroclasts on June 24, Fig. 3a and Table S1) and 11 days after the eruption (lavas on July 2, Table S1), and three months later (pyroclasts from the MV, Fig. 1, on November 18 and Table S1). June 24 samples were collected both from the main fractures (WF and UF, Fig. 1a), the MV and the active lava flow (Fig. 1 and Table S1). Twenty five scoriaceous bombs and lapilli (REU140624-9a-1 to REU140624-9a and REU140624-9b-6 to REU140624-9b-25, in Table S3) were collected from the discontinuous deposit (Fig. 3d) emplaced at the WF site (Fig. 1a), active only at the beginning of the eruptive event. Because of the short duration of the activity at the WF, the scoria fragments on the ground were scarce (Fig. 3d). The strategy was to collect a sample that was formed by the largest available number of clasts that was representative of this discrete deposit (REU140624-9 in Table S1). From the UF (Fig. 1a) only one big scoria was collected (REU140624-13, Table S1) that broke in five parts, allowing us to measure its vesiculated core and the dense quenched external part (REU140624-13-a to REU140624-13-e, in Table S3). In contrast, the sustained and slightly more energetic activity at the lower tip of the fractures, at the MV site, built a small spatter cone (Fig. 2) and accumulated a continuous, small volume deposit (Fig. 3a) of inversely graded scoria fallout (Figs. 3b and 3c). This deposit is 10 cm thick at 2 m from the vent and covers an area of about ~1000 m$^2$. For this fall deposit we collected two bulk samples, one from the base (within the lower 5 cm, REU141118-6 in Table S1) and the other from the top (within the upper 5 cm, REU140624-3, in Table S1), for the grain size (Fig. 3c) and componentry analyses. The sample at the base was collected in November because on June 24 the loose proximal lapilli blanket was still very hot (405 °C; thermocouple measurement) and fumaroles with outlet temperatures in the range 305-60 °C were sampled all along the fractures several weeks after the eruption (Fig. 1b and Table S1). These latter geochemical data are not presented in this paper. We selected 103 fragments from the coarse grained bulk deposit within the upper 5 cm of the scoria fall out deposit (Fig. 3b) at MV (REU140624-3-1 to REU140624-3-103, in Table S3) for density, connectivity, permeability, petrological and geochemical analysis. In addition, in November 2014, more than 200 clasts (comprising the REU141118-1 to REU141118-5 samples, Table S1) of similar size were collected, both close to the MV and in the 'distal' area (30 metres away from the MV site) to complete the particle bulk texture analyses and the chemical analyses.

**3.2 Grain size and componentry**

We performed grain size analyses on the two bulk samples collected from the MV, following the procedure of Jordan et al. (2015) (Table S2). The samples were dried in the oven at 90°C and sieved at ½ phi intervals in the range of -5 φ to 4 φ (Fig. 3c); the data are also shown in full phi for comparison with the deposits of the 2010 PdF fountaining episode (Hibert et al., 2015; Fig. 3f). Sieving was carried out by hand and for not longer than three minutes to avoid breaking and abrasion of the very vesicular and fragile clasts. For the scattered scoria sampled from the WF (Figs. 1, 3d and 3e), we followed the grain size strategy proposed in Gurioli et al. (2013). Within this procedure we sampled each fragment and we recorded the weight and the three main axes (a being the largest, b, and c). To allow comparison with the sieving grain size analyses (Inman, 1952), we used the intermediate b axis dimension to obtain $\varphi = -\log_2 b$.

Following the nomenclature of White and Houghton (2006) the componentry analysis is the subdivision of the sample into three broad components: i) juvenile, ii) non-juvenile particles, and iii) composite clasts. The juvenile components are vesicular or dense fragments, as well as crystals, that represent the primary magma involved in the eruption; non-juvenile material includes accessory and accidental fragments, as well as crystals that predate the eruption from which they are deposited. Finally, the composite clasts are mechanical mixtures of juvenile and non-juvenile (and/or recycled juvenile) clasts. In these mild basaltic explosions, the non-juvenile component is very scarce, so we focused on the juvenile component that is characterized by three groups of scoria: (i) spiny-opaque, (ii) spiny-glassy, and (iii) fluidal, along with golden pumice (Fig. 4). The componentry quantification was performed for each grain size fraction between -5 φ to 0.5 φ (Figs. 5a and 5b), where a binocular microscope was used for the identification of grains smaller than -1 phi (Table S2).

In the following, we will use the crystal nomenclature of Welch et al. (2009), with the strictly descriptive terms of macrocrysts (> 3 mm in diameter) mesocrysts (from 0.3 to 3 mm in diameter), and microcrysts (<0·3 mm in diameter). Regarding the June 2014 products, these ranges of size may however change in comparison to the December 2005 products studied by Welsch et al. (2009).

**3.3 Particle bulk texture (density, porosity, connectivity, permeability) and microtexture**

For each sample site (WF, UF and MV, Fig. 1a), we selected all the available particles within the 8-32 mm fraction for density/porosity, connectivity and permeability measurements (Table S3). This is the smallest granulometric fraction assumed to be still representative of the larger size class in terms of density (Houghton and Wilson, 1989; Gurioli et al., 2015), and has been used in previous textural studies (e.g., Shea et al., 2010). In addition, this size range is ideal

for vesicle connectivity measurements (e.g. Formenti and Druitt, 2003; Giachetti et al., 2010;
Shea et al., 2012; Colombier et al., 2017a, b). Density of juvenile particles was measured by
the water-immersion technique of Houghton and Wilson (1989), which is based on
Archimedes principle. A mean value for the vesicle-free rock density was determined by
powdering clasts of varying bulk densities, measuring the volumes of known masses using an
Accupyc 1340 Helium Pycnometer, then averaging. The same pycnometer was also used to
measure vesicle interconnectivity for each clast using the method of Formenti and Druitt
(2003) and Colombier et al. (2017a). Permeability measurements were performed on five
clasts: two golden pumices, one fluidal, one spiny glassy and one opaque scoria, all collected
from the MV (Table S3). Following Colombier et al. (2017a), the clasts were cut into
rectangular prisms to enable precise calculation of the cross-sectional area, which is required
to calculate permeability. These prisms were then embedded in a viscous resin, which was left
to harden for 24 h. The sample surface had been previously coated with a more viscous resin
and then wrapped with parafilm to avoid intrusion of the less viscous resin inside the pores.
The coated samples were placed with a sample holder connected to a permeameter built at
Laboratoire Magmas et Volcans (LMV, France) following Takeuchi et al. (2008). The
measurements were performed at atmospheric pressure (i.e. without confining pressure) and
the samples were measured at a range of gas flow rates and upstream air pressures to create a
curve that could be fitted using a modified version of Darcy's Law, the Forchheimer equation,
to solve for viscous ($k_1$) and inertial permeabilities ($k_2$) (Rust and Cashman, 2004, Lindoo et
al. 2016 and Colombier et al. 2017).

Vesicle size distribution was performed following the method of Shea et al. (2010) and
Leduc et al. (2015), while the total crystallinity, the percentages for both crystal phases
(plagioclase and clinopyroxene) and size-populations (meso and microcrysts) were calculated
using the raw data from FOAMS program (Shea et al 2010) and the CSDcorrections program
of Higgins (2000) and the CSDslice data base (Morgan and Jerram 2006) to have the
percentage of crystals in 3D with the corrected assumption for shape. We performed these
analyses on eight clasts picked up from each component-density distribution (stars in Figs. 6a
and 6b). The choice of the clasts was made mostly on the typologies, rather than on each
density distribution, in order to avoid the analysis of clasts with transitional characteristics.
For example, two golden pumice fragments were selected from the largest clasts that were the
less dense and didn't break, even if the values in vesicularity were similar. A larger number of
fluidal fragments were chosen (even if the density distribution was unimodal) because this
typology of clasts was the most abundant and was emitted all along the active fracture, so we
did our best in order to study products representative of the WF, the UF and the MV activities.
Only one spiny glassy and one spiny opaque were selected, because they were emitted only at
the MF. A full description of the textural measurements all performed at LMV, as well as the
raw data of these measurements are available at DynVolc Database (2017).

**3.4 Bulk geochemistry**

For the determination of the bulk chemistry (Table S4 and Fig. 7) of the different pyroclasts
we selected the largest pyroclasts of golden pumice and the largest fluidal, spiny glassy and
spiny opaque scoriae (Table S4). We also analyzed two fragments of lava, from the beginning
and the end of the eruption (Table S4). Samples were crushed into coarse chips using a steel
jaw crusher and powdered with an agate mortar. Major and trace element compositions were
analyzed using powder (whole rock composition). In addition, for a sub-set of pyroclasts,
glass chips (2-5 mm in size) were hand-picked under a binocular microscope and analyzed
separately for trace elements. For major element analysis, powdered samples were mixed with
$LiBO_2$, placed in a graphite crucible and melted in an induction oven at 1050 °C for 4.5
minutes, resulting in a homogeneous glass bead. The glass was then dissolved in a solution of
deionized water and nitric acid ($HNO_3$), and finally diluted by a factor of 2000. The final
solutions were analyzed by ICP-AES. Trace element concentrations were analysed following
a method modified from Vlastélic et al. (2013). About 100 mg of sample (powder and chip)
were dissolved in 2 ml of 28M HF and 1 ml of 14M $HNO_3$ in teflon beaker for 36 hours at
70°C. Solutions were evaporated to dryness at 70°C. The fluoride residues were reduced by
repeatedly adding and evaporating a few drops of concentrated $HNO_3$, before being fully
dissolved in ca. 20 ml of 7M $HNO_3$. These solutions were diluted by a factor of 15 with
0.05M HF (to reach rock dilution factor of ca. 4000) and trace element abundances were
determined by quadrupole ICPMS (Agilent 7500). The analyses were performed in plasma
robust mode (1550 W). The reaction cell (He mode) was used to reduce interference on
masses ranging from 45 (Sc) to 75 (As). The signal was calibrated externally (every 4
samples) with a reference basaltic standard (USGS BHVO-2) dissolved as for the samples and
using the GeoRem recommended values (http://georem.mpch-mainz.gwdg.de/). For elements
that are not well characterized in literature (As, Bi, Tl), or which show evident heterogeneity
(e.g. Pb) in BHVO-2 powder, the signal was calibrated using the certified concentrations of a
synthetic standard, which was also repeatedly measured. The external reproducibility (2σ
error) of the method is 6% or less for lithophile elements and 15% or less for chalcophile
elements.

## 3.5 Glass and crystal chemistry

Spot analyses of matrix glass and crystal composition (Table S5) were carried out using a Cameca SX100 electron microprobe (LMV), with a 15 kV acceleration voltage of 4 nA beam current and a beam of 5 μm diameter for glass analyses. However, for the spiny opaque scoria, characterized by abundant crystals with rapid growth textures, a voltage of 8 nA beam current and a beam of 10 μm diameter were used. For this latter sample, 10 analyses per sample were performed due to the heterogeneity within the highly crystallised glass (Fig. 8a), while for the other samples 6 analyses per sample were enough to characterize the clean homogeneous glass. For crystal analysis, a focused beam was used. For the characterization of the meso- and micro-crysts, due to their small size, only two to three measurements were performed, one at the edge, one in the middle and one at the core of the crystals, to check for possible zonation.

## 3.6 Melt inclusions

Melt inclusions (MIs; Table S6, Figs. 8b and 9) were characterized in the olivine mesocrysts from the three groups of scoriae (fluidal, spiny glassy and spiny opaque), but not in the pumice group, because crystals were too rare and small to be studied for MIs.

Olivine crystals were handpicked under a binocular microscope from the 100– 250 and 250– 600 μm grain size fractions of crushed tephra. Crystals with MIs were washed with acetone, embedded in epoxy and polished individually to generate adequate exposure of the MIs for *in situ* electron probe microanalysis. The MIs are spherical to oblate in shape and range in size from 10 to 200 μm. Some of the MIs contain shrinkage bubbles but all of those studied are totally deprived of daughter minerals. Major elements were measured on a Cameca SX-100 microprobe at LMV (Table S6). For major elements, the larger MIs were analyzed with a spot diameter of 10-20 μm and sample current of 8 nA, whereas the smaller MIs were analyzed with a beam of 5 μm and a sample current of 4 nA. The results are given in Table S6, and analytical details and uncertainties are listed in Óladóttir et al. (2011) and Moune et al. (2012).

## 4 Results

### 4.1 Deposit texture (grain size, componentry, morphology) and petrological description of the samples

The pyroclastic deposits at the WF and UF sites (Fig. 1a) are formed by scattered
homogeneous smooth fluidal (Figs. 3d) bombs and lapilli scoria. The average dimension of
the fragments is around 4 cm (maximum axis) with bombs up to 10 cm and scoria lapilli up to
2 cm in size (Fig. 3e).
At the MV, the reversely graded deposit (Fig. 3b) is made up of lapilli and bombs, with
only minor coarse ash (Fig. 3c). The lower 5 cm at the base are very well-sorted and show a
perfect Gaussian distribution with a mode at 4 mm (Fig. 3c). In contrast, the grain size
distribution of the upper 5 cm is asymmetrical with a main mode coarser than 22 cm and a
second mode at 8 mm (Fig. 3c). This upper deposit is negatively skewed due to the abundance
of coarse clasts. The dataset shows a similarity between the grain size distributions of the
basal tephra ejected from the 2014 MV and the ones for the lava fountaining of the 2010
summit event (Fig. 3f and Hibert et al., 2015). On the contrary, the top of the 2014 fall differs
from fountain deposits, being coarser and polymodal, and it is ascribed to dominantly
Strombolian activity (Fig. 3f).
In terms of componentry of the deposits, four types of clasts were distinguished (Fig.
4): (i) golden pumice, (ii) smooth or rough fluidal scoriae, (iii) spiny glassy scoria, (iv) spiny
opaque scoria. The pumices are vesicular, low-density fragments, characterized by a golden to
light brown color, sometimes with a shiny outer surface (Fig. 4a). They are usually rounded in
shape. Golden clasts studied for textures contain a few microcrysts of plagioclase (up to 0.1
mm in diameter), clinopyroxene up to 0.05-0.06 mm in diameter, and small olivine up to 0.03
mm in diameter (Fig. 4), together with large areas of clean, light brown glass. The fluidal
scoria fragments have dark, smooth or rough shiny surfaces (Fig. 4b). They can be more or
less elongated in shape and have spindle as well as flattened shapes. The fluidal fragments are
characterized by rare mesocrysts of plagioclase and clinopyroxene and microcrysts of
plagioclase, clinopyroxene and olivine (Fig. 4b). The spiny glassy fragments are dark, spiny
scoria that range in shape from subrounded to angular (Fig. 4c). These fragments contain
abundant glassy areas, while the spiny opaque fragments lack a glassy, iridescent surface.
Both groups of spiny clasts are characterized by the presence of dark and light brown glass.
The spiny opaque fragments are the densest fragments and have the largest amount of
crystals. They contain, as the most abundant phase, relatively large meso- and micro-crysts of
plagioclase, up to 3 mm long, together with meso- and micro-crysts of clinopyroxene and
olivine (Figs. 4c and 4d). In the dark portions of their matrix, tiny fibrous microcrysts of
olivine + clinopyroxene + plagioclase + Fe-Ti oxides occur. The spiny glassy fragments have
the same crystal populations as the spiny opaque ones, but their plagioclases are much smaller
and attain a maximum length of only 0.3 mm. Clusters of plagioclase and clinopyroxene are
present in both the spiny opaque and the spiny glassy fragments, as well as rare macrocrysts
of olivine. The olivine macrocrysts exhibit the typical compositional (Fo 84.2) and
petrographic features of olivine phenocrysts described in previous studies (Clocchiatti et al.,
1979; Albarede and Tamagnan, 1988; Bureau et al., 1998a and b; Famin et al., 2009; Welsch
et al., 2013). They are automorphic, fractured with oxides (mostly chromite) and melt
inclusions (Fig. 4c). Fluidal and pumice fragments studied for textures contain rare
macrocrysts and mesocrysts of olivine, and the crystals are essentially microcrysts. The
pumice and some fluidal fragments have lower contents of microcrysts than some fluidal and
spiny fragments, with the latter having the highest microcryst content (Table S4). For
comparison two fragments of lava have been analyzed as well (Table S3). The lava fragments
are poorly vesiculated and completely crystalline (Fig. 4e). The lava contains the same
paragenesis of crystals described in the spiny opaque fragments, with the main difference that
its matrix is completely crystallized and constituted mostly by well-formed plagioclase up to
800 microns and clinopyroxene up to 500 microns. Scarce, smaller olivines, are also present
Ubiquitous tiny rounded Fe-Ti oxides provide evidence of post emplacement crystallization.

The componentry results are reported in Figure 5 for the MV deposits; being the

deposits from the WF and UF characterized exclusively by fluidal clasts (Fig. 3). At the base
of the MV deposit, the coarse fraction of the deposit is rich in golden and fluidal components
that represent more than 60-70 vol. % (Figs. 5a and 5b). The proportion of the two groups is
similar. In contrast, in the upper, coarse grained fall deposit, the clasts bigger than 8 mm are
dominated by the spiny scoria fragments, while the fraction smaller than 8 mm show a
dramatic increase in the golden and fluidal fragments, with the fluidal ones always more
abundant than the golden ones (Figs. 5a and 5b). Abundant low-density, golden, coarse lapilli
pumice and bombs have been found scattered laterally up to 30 metres from the main axis and
were not found in the proximal deposit. On the basis of the high amount of pumice in the
lower part of the deposit, we correlate the large, low-density clasts with the base of the
proximal deposit, and consequently we interpret them as material emitted at the beginning of
the June 2014 eruptive event.
**4.2 Particle density, porosity, connectivity, permeability and micro-texture**
Density analyses performed on 200 coarse lapilli reveal a large variation in density values
from 390 kg/m$^3$ to 1700 kg/m$^3$ with a median value at 870 kg/m$^3$ (Table S3). The fragments
collected from the MV have a bimodal density distribution, with a main population of low-

density fragments having a mode at 800 kg/m$^3$, and a second and denser population centered at 1400 kg/m$^3$ (Fig. 6a). The golden and fluidal fragments form the lower-density population and the spiny fragments are dominant in the denser population (Fig. 6a). For these samples there is a marked correlation between porosity and morphology, so that the spiny-opaque clasts are the densest (up to 1600 kg m$^{-3}$, with a vesicularity of 45 vol. %) and the golden pumice are the least dense (minimum density of 390 kg m$^{-3}$ with a vesicularity of up to 86 vol. %; with a Dense Rock Equivalent density of 2880 kg m$^{-3}$). The fluidal fragments collected at the WF (Fig. 1b), have a density range from 700 to 1400 kg m$^{-3}$ and a mode at 1000 kg m$^{-3}$ (Fig. 6b). The five fragments from the only bomb collected at the UF are characterized by two distinct density values, the low density one (700-800 kg m$^{-3}$) refers to the core of the sample, while the high density one (1400-1500 kg m$^{-3}$) represents the quenched external rim of the bomb. Finally, the two fragments of lava show the highest density values at 1800 and 2150 kg m$^{-3}$. This last value is one of the highest found in the lava collected from 2014 up to 2017 (see Fig. 13 in Harris et al., 2017 and unpublished data).

In all these samples, the increase in vesicularity correlates with an increase in the amount of small (0.1 mm), medium (0.5-1 mm) and large (up to 4 mm) vesicles. In the fluidal clasts, these vesicles have a regular rounded or elliptical shape and are scattered throughout the sample. The low-density pumices are often characterized by the presence of a single, large central vesicle (10 – 15 mm) with the little vesicles and a few medium vesicles distributed all around it (Fig. 4). The spiny glass texture is characterized by a lower amount of small vesicles than in the pumice and by the presence of mostly medium sized vesicles, while the spiny opaque has more irregular shape, very large (up to 10 mm) vesicles with a small and a medium sized bubble population. In the spiny glass samples, the glass is more or less brown, with the dark brown portions being the ones with the lowest vesicle content and the highest microcrysts content. The opaque samples have a central, very dark glass portion, with low vesicle content, and a more vesicular glassy portion at the outer edges (Fig. 4). The two fragments of lava are poorly vesiculated (Fig. 6a) and characterized by large, irregular vesicles (up to 5 mm in diameter). Clusters of small vesicles (up to 0.1 mm) are scattered between the large ones.

The vesicle size distribution (VSD in Fig. 4) histograms are characterized by a decrease in percentage of vesicles from the golden to the lava as well as an increase in coalescence and or expansion signatures in the spiny fragments, marked by the increasing of the large vesicles population (Figs 4c and 4d). This trend is also marked by the decrease in number of vesicle per unit of volume ($N_v$, Fig. 4) from the golden to the lava. Finally, the

trend is also mirrored by the total percentage of crystals (calculated in 3D, Fig. 4 and reported
for each sample in Table S3) that increases with the increase of density of the clasts, from a
minimum of 8 vol. % for the golden up to 55 vol. % for the spiny opaque scoria, and 100 vol.
% for the lava (Fig. 4). Mesocrystals, formed mostly by the same proportion of plagioclase
and clinopyroxenes, are absent or very scarce in the golden and fluidal fragments, while they
reach their maximum values, up 21 vol. % in the spiny opaque fragment. The population of
microcrystals is mostly constituted by plagioclases that range from a minimum of 6 vol. % in
the golden, up to 23-25 vol. % in the spiny fragments and to 64 vol. % in the lava.

The connectivity data (Fig. 6c) also indicate that the fluidal and golden clasts have a

larger amount of isolated vesicles (up to 40 vol. %) with respect to the spiny products. The
fluidal clasts from the WF are the most homogeneous with an average percentage of isolated
vesicles around 30 vol. %. In contrast, both the pumice and the fluidal fragments from the
MV, characterized by higher values of porosity (> 75%), have a wide range in percentage of
isolated vesicles (between 20 and a few vol. %). The fragments of the bomb collected at the
UF are consistent with a vesiculated core characterized by scarce isolated vesicles and the
quenched rind that has 30 vol. % of isolated vesicles. Finally the spiny fragments have the
lowest content of isolated vesicles (0-5 vol. %). Despite the presence of these isolated
vesicles, all the samples shear high values of permeability, with the Darcian (viscous, $K_1$)
permeability values ranging from $10^{-11}$ to $10^{-10}$ m$^2$ (Fig. 6d and Table S3). The graph of
vesicularity versus $K_1$ shows a slightly increase in permeability with vesicularity, being the
golden pumice the most permeable among the samples and the spiny glassy fragment the least
permeable. The three samples collected from the February 2015 eruption fit this trend.
However, the densest spiny opaque scoria of the 2014 eruption shares the high permeability
value of the golden pumice.
**4.3 Chemistry of the products**
Major and trace element concentrations of whole-rock and hand-picked glass samples are
reported in Table S4. Whole rock major element composition is very uniform (e.g.,
6.5<MgO<6.7 wt%) and well within the range of Steady State Basalts (SSB), the most
common type of basalts erupted at PdF (Albarède et al., 1997). However, compatible trace
elements, such as Ni and Cr, are at the lower end of the concentration range for SSB
(<100ppm) indicating that the June 2014 eruption sampled relatively evolved melts. Ni and Cr
generally show higher concentrations in 2014 bulk rocks (79<Ni<92ppm and 71<Cr <87ppm)
compared to the 2014 glass chips (66<Ni<73ppm and 54<Cr <59ppm for all but two chips).
In the Cr vs Ni plot (Fig. 7a), whole rocks plot to the right of the main clinopyroxene +/-
plagioclase–controlled melt differentiation trend. This shift reflects the addition of Ni-rich
olivine (Albarède and Tamagnan, 1988). We estimate that the Ni excess results from the
occurrence of a low amount (0.7 to 1.3 wt%) of cumulative olivine in whole rocks, consistent
with thin section observations. The composition of olivine macrocrysts (ca. Fo84) is too
magnesian to be in equilibrium with the low-MgO evolved composition of the 2014 magma.
Using our estimate for the amount of cumulative olivine, we recalculate the olivine–corrected
MgO content of the 2014 magma at 6.2 wt%. The June 2014 melt is thus only moderately
depleted in compatible elements compared to the previous eruption of December 2010
(MgO~6.6 wt%, Ni~80 ppm, Cr~120 ppm). Conversely, the June 2014 melt is significantly
depleted in compatible elements compared to the earlier November 2009 eruption, which
sampled relatively primitive magmas (average MgO~7.7 wt%, Ni~135 ppm, Cr~350 ppm)
(Fig. 7a). The 2014 evolved composition plots at the low-Ni-Cr end of PdF historical
differentiation trend (Albarède and Tamagnan, 1988), near the composition of lavas erupted
on 9 March 1998 after 5.5 years of quiescence (1992-1998). Note that olivine accumulation at
PdF generally occurs in melt having ca.100 ppm Ni (Albarède and Tamagnan, 1988). Olivine
accumulation in evolved melts (Ni < 70 ppm) seems to be a distinctive feature of many small
post-2007 eruptions (e.g. this event and the three 2008 eruptions, see Di Muro et al., 2015).
A closer inspection of Ni-Cr variability in June 2014 whole rock samples (Fig. 7b)
reveals that scoria from the WF (140624-9b-6, Table S4) and early erupted lavas (1406-21-1,
Table S4) have the lowest amount of olivine (<0.9%) whereas scoria from the UF (140624-
13a) and late erupted lavas (140324-12) have a slightly higher amount of olivine (>1.2%).
This is consistent with the general trends observed at PdF of olivine increase from the start to
end of an eruption (Peltier et al., 2009).
The so called "olivine control trend" in Ni-Cr space cannot be explained either by
addition of pure olivine, which contains less than 500 ppm Cr (Welsch et al., 2009; Salaün et
al., 2010; Di Muro et al., 2015), or by the addition of olivine plus pyroxene (which would
require ca. 50% pyroxene with 970 ppm Ni and 4800 ppm Cr, see Fig. 7 caption). Instead,
addition of olivine hosting ca. 1% Cr-spinel (with 25 wt.% Cr) accounts for data and
observations, and is consistent with crystallization of olivine and Cr-spinel in cotectic
proportions (Roeder et al., 2006). The fact that some samples (golden pumice) plot off the
main, well-defined array, can be explained either by addition of more or less evolved olivine
crystals (within the range of Fo 80-85 measured in June 2014 samples) and/or slight
variations (± 0.02%) in the proportion of Cr-spinels (Fig. 7b).
The glass chemistry of the four clast types allows us to correlate porosity and oxide
contents and shows an increase in MgO from the spiny opaque to fluidal and golden
fragments (Fig. 8a). Consistent with petrological and textural observations, the spiny opaque
is the most heterogeneous type of clast in terms of glass composition (Fig. 8). The glassy
portion at the edge of the clast is similar to the spiny glass, while the interior, characterized by
dark areas rich in tiny fibrous microcrysts, shows scattered glass compositions with very low
MgO content as well as a decrease in CaO (Fig. 8). We attribute the significant variation in
glass composition within the different components to variable degrees of micro-crystallisation
as the bulk chemistry of all clasts is very similar and globally homogeneous.

## 4.4 Melt inclusions

MI analyses must be corrected for post-entrapment host crystallisation at the MI - crystal
interface. We used a Kd = $(FeO/MgO)_{ol}$ / $(FeO/MgO)_{melt}$ = 0.306 (Fisk et al., 1988; Brugier,
2016) and an average $Fe^{3+}/\Sigma Fe_{total}$ ratio of 0.11 (Bureau et al., 1998a; Di Muro et al., 2016 and
references therein) defined for PdF magmas. For the June 2014 melt inclusions, the post
entrapment crystallization (PEC) ranges from 2.9 to 10.5 wt%. Raw and corrected major and
volatile element concentrations of MIs are reported in Table S6.
Host olivines span a large compositional range from $Fo_{80}$ to $Fo_{86}$. Despite the evolved
bulk composition of the magma, most olivines are quite magnesian ($Fo_{83-85}$) and are not in
equilibrium with the evolved host magma. On the contrary, Mg-poor olivines ($Fo_{80-81}$) can be
considered as being in equilibrium with the bulk rock composition. The corrected
compositions of MIs in phenocrysts from the different samples partly overlap with the
evolved bulk rocks ($MgO_{wr}$: 6.1-7.2 wt%) and extend to higher MgO contents of up to 8.8
wt% (Table S6). MIs display a narrow range of transitional basaltic compositions ($K_2O$= 0.5-
0.9 wt%) and show no significant difference between the three types of scoriae. The major
element composition of melt inclusions correlates with that of the host olivines. Melt
inclusions in the high Fo-olivines have the highest MgO, CaO and $TiO_2$ and lowest $K_2O$
concentrations (Table S6). It is interesting to note that the June 2014 products contain two
populations of magnesian ($Fo_{>83}$) olivines hosting melt inclusions with two distinct Ca
contents. Most of the magnesian olivines contain MIs with unusually high CaO contents (11.6
– 12.9 wt%) and high $CaO/Al_2O_3$ ratios (0.8-0.9), higher than that of the bulk rocks (0.8) (Fig
8). The occurrence of olivines with "high Ca" melt inclusions has been observed in all three
different types of scoriae. A few magnesian olivines and all Mg-poor olivines ($Fo_{80.5-83.6}$) host
MIs with lower CaO contents (11.4 wt%). This latter composition overlaps with that of the
bulk rock (Fig 8). The "high Ca" population of inclusions is also enriched in $TiO_2$ and $Al_2O_3$
and depleted in MgO, $FeO_T$ and $Na_2O$ for a given olivine Fo content with respect to the "low
Ca" population. Both low- and high-Ca populations of melt inclusions have similar $K_2O$
contents and total alkali content increases from 3 wt% at 12.6 wt% CaO, to 3.5 wt% at 10.8
wt% CaO. However, we remark that high Ca melt inclusions from the June 2014 activity
record a significant scattering in $K_2O$ contents, which range from 0.55 to 0.9 wt%. These
anomalous compositions potentially track processes of crystal dissolution (e.g. pyroxene
dissolution).

MIs in olivines from June 2014 can best be compared with those of other recent small-

volume and short-lived eruptions which emitted basalts with low phenocryst contents, like
those in March 2007 ($0.6 \times 10^6$ $m^3$) and November 2009 ($0.1 \times 10^6$ $m^3$) (Roult et al., 2012).
March 2007 aphyric basalt has a bulk homogeneous composition with intermediate MgO
content ($MgO_{wr}$: 7.33 wt%; $K_2O$: 0.67 wt%). Their olivines (Fo 81) are in equilibrium with
the bulk rock and their composition is unimodal (Di Muro et al., 2014). November 2009
products are the most magnesian lavas emitted in the 2008-2014 period, slightly zoned
($MgO_{wr}$: 7.6-8.3 wt%; $K_2O$: 0.75 – 0.62 wt%) and contain a few percent of normally zoned
olivine macrocrysts with bimodal composition (Fo81 and Fo83.5, see Di Muro et al., 2016).
June 2014 bulk rocks ($MgO_{wr}$: 6.7 wt%; $K_2O$: 0.75 wt%) and melt inclusions in $Fo_{80-81}$
olivines are quite evolved. Their composition is close to that of products emitted by summit
intracaldera eruptions in 2008, ca. 1.5 years after the large 2007 caldera forming eruption (Di
Muro et al., 2015) (Fig. 8). As already reported for 2008 products, many olivine macrocrysts
of 2014 are clearly too magnesian to be in equilibrium with the relatively evolved host melts.
Overall, MgO content in 2007-2014 melt inclusions tends to decrease with decreasing Fo
content of the host olivines. MIs in olivines also exhibit a trend of linear decrease in MgO and
increase in FeO from April 2007 to 2009-2014 products (Fig. 9). Melt inclusions in March
2007, November 2009 and June 2014 follow the same trend of FeO enrichment (Fig. 9). In the
large-volume and olivine-rich April 2007 products, MIs in magnesian olivines with $Fo_{>82}$ have
distinctly higher MgO, FeO and lower $SiO_2$ and $Al_2O_3$ than MIs in 2009-2014 products. The
distinctive FeO enrichment of many of the MIs from the April 2007 oceanite has been
interpreted by Di Muro et al. (2014) as a result of post-entrapment modification related to new
magma inputs into long lasting magma storage.

Two populations of low- and high-Ca melt inclusions are also found in the November

2009 olivines. Low-Ca melt inclusions from the November 2009 and June 2014 eruptions
indicate a single trend of chemical evolution (Fig. 8), consistent with bulk rock compositions.
June 2014 products have lower MgO and CaO contents than those from November 2009.
Significant scattering in $K_2O$ content (0.6-0.9 wt%) is found in low-Ca inclusions from 2009,
as observed in high-Ca inclusions from the 2014 eruption, but they share similar $K_2O$
contents. In 2009 and 2014 products, $K_2O$ content of melt inclusions is partly anti-correlated
with the olivine Fo content. This observation has been attributed to moderate heterogeneity of
primary melts feeding the plumbing system of PdF. Rapid temporal changes of $K_2O$ content in
PdF basalts have been reported (Boivin and Bachelery, 2009).

**4.5 Mineral composition and glass – plagioclase equilibrium**

All 2014 scoriae (spiny, fluidal, golden) contain the same paragenesis of olivine,
clinopyroxene and plagioclase. The composition of minerals found in golden, fluidal and
spiny scoriae is indistinguishable.
In olivines, average MgO content decreases from macrocrysts ($Fo_{84.1}$) to mesocrysts
($Fo_{79.6}$) to microcrysts. Olivine microcrysts (Table S5) are normally zoned. Their composition
ranges from $Fo_{78.0-75.3}$ in the cores to $Fo_{74.3-70..5}$ in the rims. Overall, olivines in 2014 products
span the full range of typical Fo contents of recent PdF magmas (Boivin and Bachèlery, 2009;
Di Muro et al., 2014; 2015). Clinopyroxene composition (augites) ranges from $En_{53}Fs_{15}Wo_{32}$
to $En_{41}Fs_{14}Wo_{45}$. Their average composition ($En_{45}Fs_{14}Wo_{41}$) is consistent with that found in
other recent evolved melts like those emitted by the 2008 eruptions (Di Muro et al., 2015) and
more generally in recent PdF products (Boivin and Bachèlery, 2009). Clinopyroxenes are
unzoned, the composition of cores and rims is very similar and close to that found in
microcrysts and mesocrysts. Plagioclase composition ranges from $An_{79.5}Ab_{19.9}Or_{0.6}$ to
$An_{63.1}Ab_{35.7}Or_{1.2}$ with a bimodal distribution ($An_{76.5-79.5}$ and $An_{63.1-72.9}$, Fig. 10a). Similar
bimodal distributions were observed in many other products at PdF (Di Muro et al., 2015).
Mesocrysts ($An_{75.5}Ab_{23.8}Or_{0.7}$ on average) are more calcic with respect to microcrysts
($An_{65.7}Ab_{33.1}Or_{1.2}$ on average). Normal zoning is found from plagioclase cores to rims (Fig.
10a). The composition and zonation of 2014 plagioclases clearly contrast with the complex
and often reverse zoning patterns and intermediate composition of the 2008 PdF products that
were attributed to pre-eruptive magma heating (Di Muro et al., 2015).
Plagioclase-melt equilibrium and melt composition in pyroclastic rocks and water-
quenched lavas were used to estimate both temperature and water content dissolved within the
melt (Fig. 10b and Table S5). Temperature estimates are based on the (dry) equation of Helz
and Thornber (1987) recalibrated by Putirka (2008). Dissolved water content was calculated
from the plagioclase hygrometer of Lange et al. (2009) at 50 MPa. This pressure corresponds
to the average $CO_2$-$H_2O$ saturation pressure (recalculated with Papale et al., 2006) typically
recorded in melt inclusions from central products at PdF (e.g. 1931 eruption in Di Muro et al.,
(2016) and references therein). This pressure roughly corresponds to the sea level depth,
which is inferred to be the location of the potential main shallow magmatic reservoir (Peltier
et al., 2009; Lengliné et al., 2016; Coppola et al., 2017). The application of the plagioclase
hygrometer of Lange et al. (2009) makes it possible to estimate the dissolved water content in
the melt with a nominal uncertainty of 0.15 wt% and is only slightly dependent on pressure.
Plagioclase compositions not in equilibrium with the melt (glass or bulk rock) are those of
mesocryst cores with the highest ($An_{>76.5}$) anorthite content (Fig. 10a and Table S5). Such
compositions are more in equilibrium with CaO-richer magnesian melts than those measured
in matrix glasses and bulk rocks of 2014 eruption and likely formed during early stages of
shallow magma differentiation (Fig. 10a).
In order to determine pre-eruptive conditions, calculations were performed only on
paired plagioclase rims and matrix glasses in equilibrium, using the plagioclase-melt
equilibrium constant of Putirka (2008) calibrated for melts whose temperature exceeds
1050°C ($Kd_{An-Ab}$ = 0.27±0.05). Our review of published and unpublished data shows that melt
temperature progressively decreases from April 2007 (1188+/-16 °C) to January-October
2010 (1147+/-9°C) and positively correlates with $K_2O$ content in melts which increases from
0.70 to 0.96 wt%  (Fig. 10b). The melts from the June 2014 eruption record the lowest
temperatures in post-2007 eruptions (1131±15 °C) together with the highest $K_2O$-enrichment
($K_2O$: 0.90±0.12 wt%). The lowest temperatures are recorded by spiny scoriae, while the
temperature of golden scoriae overlaps with that of 2010 products emitted before the 2010-
2014 phase of quiescence. In spite of the large variability in melt composition and
temperature, average pre-eruptive water content dissolved in the melts (0.5 +/- 0.2 wt%) is
quite homogeneous for the whole 2008-2014 period. In 2014, the lowest estimated dissolved
water content (down to 0.38 wt%) is for the golden and some fluidal scoriae, while the
maximum amount (0.68 wt%) is for the spiny opaque scoriae. However, water content
estimated from core-bulk rock equilibrium (0.3±0.1 wt%) is slightly lower than that estimated
from rim and microlite-matrix glass equilibrium (0.5±0.2 wt%), but the difference broadly
overlaps the nominal uncertainty related to calculations. Dissolved water contents in melts of
the pyroclasts are thus intermediate between those measured in 2007 melt inclusions ($H_2O$:
0.8 +/- 0.15 wt% and up to 1.1 wt%) and those typically found in degassed matrices of lava
and Pele's hairs of 2007 (Fig. 10b; 0.2 wt%; see Di Muro et al., 2015; 2016).

## 5 Discussions

### 5.1 Eruptive dynamics

The activity fed by the uppermost WF and UF (Fig. 1) was very short-lived, as shown by the presence of only scattered bombs and coarse lapilli (Figs 3d and 3e). The homogeneity of these clasts, their coarse grained nature and the fluidal smooth texture are in agreement with very short-lived fire-fountaining/magma jets. Glassy outer surfaces of clasts have been interpreted as a late-stage product of fusion by hot gases streaming past the ejecta within the jet/fountain (Thordarson et al., 1996; Stovall et al., 2011). However, the occurrence of this process is not supported by the homogeneous glass composition in our fluidal clasts. Therefore, we interpret these features here just as rapid quenching and not re-melting. Vlastélic et al. (2011) documented the mobility of alkalis and other elements on PdF clasts that experienced long exposures to acid gases. In the 2014 eruption pyroclasts, the mobility of elements was prevented by the short duration of the events.

At lower altitude and close to the MV (Fig. 1), the 5 cm layer at the base of the fall deposit is fine-grained (Figs. 3b and 3c), rich in fluidal and golden fragments (Fig. 5), with a perfect Gaussian grain size curve (Fig. 5), and similar to that reported from the weak 2010 fountaining event (Fig. 3f and Hibert et al., 2015). Therefore, we interpret this deposit as being due to weak Hawaiian like fountaining (sustained, but short-lived) activity. We want to remark here that this activity happened during the night and was not observed. The top of the same deposit is coarse grained (Figs 3b and 3c), bimodal, has a lower content in coarse ash (Table S2) and is rich in spiny opaque and spiny glass fragments (Fig. 5). The reverse grain size likely records the transition from early continuous fountaining to late discrete Strombolian activity (observed and recorded on the 21 of June 2014, Fig. 2). This transition in activity is typical of many eruptions at PdF (Hibert et al., 2015). The reverse grading of the whole deposit (Figs. 3b and 3c) is thus not correlated with an increase in energy of the event, but with two different eruptive dynamics and fragmentation processes. The decrease in coarse ash, which correlates with the decrease in energy of the event, highlights the most efficient fragmentation process within the Hawaiian fountaining with respect to the slow gas ascent and explosion of the Strombolian activity. These conclusions are consistent with (i) the continuous and progressive decrease in intensity of Real time Seismic Amplitude Measurement recorded by the OVPF seismic network (unpublished data), and (ii) satellite derived TADR which suggest continuous decay of magma output rate after an initial short-lived intense phase (Coppola et al., 2017).

**5.2 Interpretation of the different textural signatures and the meaning of the 4 typologies of clasts.**

1) *Background on the texture of clasts from Hawaiian and Strombolian activities*

The first microtextural analysis of Hawaiian ejecta was performed by Cashman and Mangan (1994) and Mangan and Cashman (1996) on pyroclasts from 1984 to 1986 Puʻu ʻŌʻō fountainings. The authors defined two clast types: 1) 'scoria' consisting of closed-cell foam of ≤85% vesicularity, with round, undeformed, broadly-sized vesicles, and 2) 'reticulite', an open-cell polyhedral foam with ~1 μm thick vesicle walls with >95% vesicularity. They stated that the scoria to reticulite transition is a consequence of Ostwald ripening, where larger bubbles grow at the expense of smaller bubbles due to post-fragmentation expansion of clasts within the fountain. According to this model, scoria preserves textures closer to conditions at fragmentation, whereas continued vesiculation and clast expansion in the thermally-insulated core of the fountain results in reticulate. This model was confirmed at lava fountains at Etna (Polacci et al., 2006), Villarrica (Gurioli et al., 2008), Kīlauea Iki, (Stovall et al., 2011 and 2012), Mauna Ulu (Parcheta et al., 2013) and Al Madinah (Kawabata et al., 2015). These last authors also measured the connected and isolated porosity in the AD1256 Al-Madinah Hawaiian fountaining eruptions. They found that the reticulite-like textures from the central part of these very high fountains showed isolated vesicles in agreement with low shear rates and low viscosity melts, where bubbles may grow spherically and remain isolated. In contrast, at margins of the fountains, high shear may lead to stretching and mechanical coalescence of bubbles, forming the common, fluidal types of particles seen also in the deposits. They also stated that lower vesicularity and greater isolated porosity were found in some tephra interpreted as resulting from violent Strombolian eruptive phases.

The data that we found in our study of the typical activity of PdF agree only partially with all these interpretations. The reason is that we sampled and measured products of very weak Hawaiian to Strombolian activities. If we plot the approximate durations and masses of these events on the Houghton et al. (2016) diagram, the 2014 activity of PdF falls into the two fields for transient and fountaining activity, but at the base of the diagram. We here show for the first time that short lived and weak fountaining can preserve pyroclast textures that record magma ascent and fragmentation conditions before the explosions and also provide some information about the pre-eruptive storage conditions. The occurrence of time-variable ascent conditions is also reflected in the time evolution of eruptive dynamics, with the golden and

fluidal scoriae emitted from the low Hawaiian fountaining episodes and the spiny fragments
from the Strombolian-like explosions
*2) The four typologies of clasts and their distribution in space and in time in the 2014*
*eruption at PdF*
So, as described in 5.1, longitudinal variation in eruptive style along the fracture system
produces a spatial variability in the proportions of the four typologies of clasts. The
uppermost fractures (WF and UF, Fig. 1a) are characterized solely by fluidal fragments (Fig.
4b); they lack both the spiny and the golden components. In addition, these fluidal clasts are
the ones showing the smoothest surfaces (indicative of rapid quenching in a very hot
environment), low porosity values (between 50 to 77 vol. %, Fig. 6b), the highest content in
isolated vesicles (~ 30 vol. % Fig. 4c), and low vesicle numbers (3 to 5 x $10^6$, Fig. 4b),
comparable to the spiny fragments. They have scarce mesocrysts (1-2 vol. % Table S3) and
very low amount of microcrysts of plagioclase and clinopyroxene (3 to 11 vol. %, Table S3).
These fluidal scoria fragments were emitted by short lived jets of magma, therefore they
underwent rapid quenching in a very hot environment that prevented any expansion or further
vesiculation and preserved a very high number of isolated vesicles (Fig. 6d). Syn-eruptive
crystallization was hindered by high ascent velocities in the dyke, due to the sudden release of
over-pressure in the shallow magma reservoir.
The four typology of clasts, golden pumice, fluidal scoria and the spiny fragments
(Fig. 4) were found associated only at the MV. The relative proportions of these four
typologies of clasts correlate with the eruptive dynamics. The golden lapilli and fluidal clasts
were in fact dominant in the Hawaiian, more energetic activity at the beginning of the
eruption (during the night between the 20 and the 21 of June 2014). In contrast, the spiny
fragments were dominant during the Strombolian activity, coinciding with the decreasing in
Mass Discharge Rate (MDR, early in the morning of the 21, Fig. 2 and Coppola et al., 2017).
The golden and fluidal fragments from the MV show the highest porosity (86 %, Fig. 6a),
variable proportions of isolated vesicles (Fig. 6c) and high, but variable, $N_V$ numbers (Figs.
4a). They are also characterized by a uniform vesicle size population with clear evidence of
incipient expansion, especially in the fluidal fragments (Figs. 4a and 4b). From the
connectivity graph, there is a clear decrease in isolated vesicles with the increase in
vesicularity (Fig. 6c). The content in crystal, mostly formed by microcrysts of sodic
plagioclase (Fig. 10a) due to magma degassing during its ascent and decompression in the
conduit (Di Muro et al., 2015), is very low, especially in the golden pumice (up to 15 vol. %),
and slightly higher for the fluidal clasts (up to 23 vol. %). We interpret the golden fragments,
at the MV, to be the fastest (low amount of microcrysts) and less degassed magma (high
vesicularity coupled with high $N_V$), which experienced only a very short residence time in the
magma transport system (dyke+vent), followed by the fluidal fragments. In contrast the spiny
fragments, characterized by higher percentage of microcrysts and mesocrysts, by the lack of
isolated vesicles, by the presence of coalescence signature and low $N_V$ values (Figs. 4c and
4d), are indicative of an extensively degassed and cooled magma. The presence of the
mesocrysts (that formed in the shallow reservoir) in the spiny fragments, and their slightly
cooler temperature (Fig. 10b), strongly support this interpretation. The spiny fragments likely
record the slowest ascent velocity and the longest residence time in the reservoir+dyke+vent
system compared to the golden/fluidal counterpart. Therefore these fragments are associated
with Strombolian events, and decreasing MDR, in agreement with their slower ascent that
allows extensive syneruptive crystallization.
Among spiny fragments, the opaque ones are the densest, they lack a uniform glassy
surface, and they are characterized by i) very high microcrysts content, ii) strong coalescence
signature (Fig. 4d), iii) heterogeneous glass chemistry, and iv) mingling with hotter magma at
the clast edges (Fig. 8a). All these features reveal the composite nature of these clasts. We
interpret the spiny opaque as spiny glass fragments recycled inside the eruptive vent during
the explosions, being the densest portion of the magma prone to fall back in the vent/fracture
(Fig. 2b).

*3) Degassing-driven versus cooling-driven crystallization*

Syn-eruptive degassing is favoured by bubble connectivity/permeability (Figs. 6c and 6d) in
the ascending magma, enhanced by syn-eruptive crystallisation in the conduit (especially
microcrysts of plagioclase, Fig. 10a), even for magmas at low vesicularity. However, our
dataset also supports the occurrence of magma stratification in the reservoir. Textural and
petrological data demonstrate that the initial activity emitted a small volume of melt
(represented by golden and large part of the fluidal fragments) with very scarce crystals. This
crystal-poor melt was followed in time by the main volume of magma that contains a larger
amount of mesocrysts (spiny clasts and lava). Lava flows represent the main volume emitted
in the 2014 eruption. Mesocrysts are absent in the golden, scarce in the fluidal and more
abundant in the spiny (Figs 4b, 4c and 4d) and lava (Fig. 4e) fragments and consist in an equal
percentage of plagioclase and clinopyroxene and minor olivine. Their composition indicates
that they formed in the reservoir, as shown by their different composition in respect to the
microcrysts counterparts that formed during melt degassing in the conduit (Fig. 10a). Most
important, a large amount of microcrysts in lava formed in the reservoir as well during
magma cooling (Figure 10a). So, we have a range of crystallization conditions. The fact that
the lighter plagioclase are not concentrated in the upper and early erupted portion of the
reservoir can be due either to the fact that often they are locked in clusters with the
clinopyroxene or that this melt was expelled from the crystal-rich portion of the reservoir (see
Figure 10b). Water exsolution from the melt can result from its extensive crystallization,
which induces an increase in dissolved volatile content, up to saturation (second boiling) and
can drive melt-crystal separation.

In conclusion, the crystals in the 2014 fragments do reflect the shallow reservoir

conditions and the ascent degassing processes.

*4) Textural syn-eruptive versus post fragmentation modifications*

To prove that the 2014 vesiculation of the clasts have been not modified by post
fragmentation expansion process, following Stovall et al. (2011), we use a plot of vesicle-to-
melt ratio ($V_G/V_L$, after Gardner et al., 1996) and vesicle number density ($N_V$, Fig. 11). As
demonstrated by Stovall et al. (2011), addition of small bubbles leads to an increase in $N_V$ and
only a slight increase in $V_G/V_L$. Bubble growth by some combination of diffusion and
decompression leads to an increase in $V_G/V_L$ at constant $N_V$. $N_V$ decreases while $V_G/V_L$
increases during bubble coalescence, whereas loss of bubbles via collapse or buoyant rise
leads to a reduction in both parameters. Intermediate trends on the diagram reflect
combinations of more than one of these processes. The pumice and the scoria from the MV of
PdF show the highest $V_G/V_L$, but also the highest Nv, suggesting preservation of small
vesicles and growth by some combination of diffusion and decompression. The presence of
the small vesicles and the lack of a strong coalescence/expansion signature confirm that the
weak PdF activity leads to only limited post-fragmentation expansion inside the hot portions
of the short-lived fountains. These data contrast with the data from the more energetic
fountaining events observed at Kilauea or elsewhere, where pre-eruptive information is
basically erased because pumice textures are dominated by expansion effects due to their
longer residence within the long-lived energetic fountaining. In contrast, the densest, spiny
scoriae and the scoria from the Fractures activity show the lowest values of Nv and $V_G/V_L$,
due to incipient coalescence and/or loose/lack of small bubbles.
According to previous works (listed above), the golden pumice of PdF should be
derived from the central part of the fountains, but they do not show the strong post expansion
signatures reported by other samples collected from more energetic Hawaiian fountainings
(Fig. 11). It is interesting to note that the fluidal fragments at the MV are less smooth (Fig. 4),
more vesiculated, and have a lower content of isolated vesicles than the fluidal scoria from the
uppermost Fractures (Fig. 6). Therefore fluidal fragments at the 2014 MV could indeed
represent clasts that have been partly modified during their residence in the external part of
the fountains, while the golden samples could come from the central part (Stovall et al., 2011
and 2012). However, the slight differences in crystallinity and glass chemistry between the
fluidal and golden fragments support the idea that each of these fragments has an imprint from
the pre-fragmentation setting. In contrast, the spiny fragments from the MV and the fluidal
fragments from the Fractures show low $N_V$ and low $V_G/V_L$ in agreement with loss of vesicles
and coalescence. However, the presence of large numbers of isolated vesicles within the
fluidal scoria from the Fractures agrees with their provenance from a fast hot ejection of
relatively degassed magma (low $N_V$). In contrast the spiny fragments, especially because of
the presence of abundant mesocrysts and increase in syneruptive microcrysts, are indicative of
the slowest ascent velocity and extensively degassing and cooled magma. The spiny
fragments are the most degassed, densest and the most crystal rich magma that was emitted
during low-energy activity by Strombolian explosion, where recycling phenomena were also
very frequent (Fig. 2f).
Our vesicle connectivity results are in full agreement with the recent review of
Colombier et al. (2017b). According to these authors, connectivity values can be used as a
useful tool to discriminate between the basaltic scoria from Hawaiian (fire fountaining) and
Strombolian activity. The broad range in connectivity for pumice and scoria from fire
fountaining is interpreted simply as being due to variations in the time available before
quenching due to differences in location and residence time inside the fountain. The fluidal
fragments from the WF are the richest in isolated vesicles because they are transported by
very short lived hot lava jets. In contrast, the higher connectivity observed in scoria from
Strombolian activity is probably related to their higher average crystallinity, and more
extensive degassing prior to the eruption (Colombier et al., 2017b). The spiny surface of these
Strombolian fragments is due to the fact that these weak explosions emit only a small solid
mass fraction and the partially quenched dense clasts land quickly after a short cooling path
through the surrounding atmosphere (e.g. Bombrun et al., 2015).

All the clast, from golden to spiny, are very permeable, independent on their vesicularity, crystal content and/or of the presence of isolated vesicles. This is in agreement with our interpretation that magma degasses during its ascent in the conduit and that promotes microlite nucleation (see the sodic plagioclase, Fig. 10a) before magma fragmentation (see also Di Muro et al. 2015 with the Pele's hairs and tears samples for the three 2008 eruptions). Moreover, we always find that some of the spiny clasts (especially the opaque ones) are slightly less permeable than the golden and fluidal ones, but not with a low permeability as we would expect by their low vesicularity.

In conclusion, we can state that i) the crystals lower the percolation threshold and stabilize permeable pathways and ii) this is true for the syn-eruptive sodic plagioclase that favor an efficient degassing in the relatively crystal-rich magma, because of their low wet angles that favor degassing against nucleation (Shea, 2017) and their aspect ratio (e.g. Spina et al. 2016) iii) therefore permeability develops during vesiculation through bubble coalescence, which allows efficient volatile transport through connected pathways and relieves overpressure (Lindoo et al., 2017). Pervasive crystal networks also deform bubbles and therefore enhance outgassing (Oppenheimer et al., 2015). Based on Saar et al. (2001) crystals should start to affect the behavior of the exsolved volatile phase when they approach 20 vol. % (Lindoo et al., 2017). In our dataset, apart from the golden and part of fluidal, all the other clasts do have microcrysts >20%. Our data completely support that slow decompression rate allows more time for degassing-induced crystallization, which lowers the vesicularity threshold at which bubbles start to connect.

Rapid re-annealing of pore throats between connected bubbles can happen due to short melt relaxation times (Lindoo et al; 2016). This phenomenology could explain the high amount of isolated vesicles in the fountaining samples. However, vesicle distributions of the golden and fluidal fragments are almost perfect Gaussian curves, so it seems that if the relaxation process happens it just merged perfectly with the expected vesicle distribution. In contrast, coalescence and/or expansion (as we observe in the spiny fragments) do not fit the curves (Fig. 4). In addition, we should expect that in crystal-poor fragments, due to melt relaxing and pathway closure, the clasts became almost impermeable after quenching, as revealed by some petrological experiments performed on crystal-poor basaltic magma (Lindoo et al., 2016). In contrast, in high crystalline magmas, the presence of micro-crystals increases viscosity thus preserving the coalesced textures (see Moitra et al., 2013). The isolated vesicle-rich fragments of the 2014 PdF eruption are highly permeable, and are characterized by variable ranges of porosity and numbers of vesicles (Fig.4 and Fig. 6d) that

seem more related to the pre-eruptive conditions than to the post relaxation of low-viscosity
melts. In the 2014 crystal-poor samples, the permeability increases rapidly once the
percolation threshold has been reached, and efficient degassing prevents bubble volumes from
expanding past the percolation threshold (Rust and Cashman 2011).
In conclusion, also the vesicles in the 2014 fragments do partly reflect the shallow
reservoir conditions and mostly the ascent degassing processes.

**5.4 Integration between the physical and textural characteristics of the products and**
**their geochemical signature: insight into the feeding system**
According to Peltier et al. (2016), the June 2014 eruption emitted magma from a shallow
pressurized source located only 1.4-1.7 km below the volcano summit. Coppola et al. (2017)
suggest that the 2014 event was fed by a single shallow and small volume magma pocket
stored in the uppermost part of the PdF central plumbing system. All 2014 clasts show
homogeneous and evolved bulk compositions, irrespective of their textural features. June
2014 products are among the most evolved products erupted since at least 1998 and are
moderately evolved with respect to those emitted in 2010, just before the 2010-2014
quiescence. Bulk rock and melt inclusion data suggest that the 2014 evolved magma can be
produced by crystal fractionation during the long lasting (4.6 years) storage and cooling of the
magma injected and partly erupted in November 2009. The different types of scoria and
pumice emitted in 2014 show significant variations in glass composition (Fig. 8b) due to
variable degrees of micro-crystallization. In theory, microcrysts can reflect late stage (during
magma ascent and post-fragmentation) crystallization. In this case, their variable amount
within, for instance, the glassy and opaque parts of the spiny scoria might reflect slower
ascent velocity or longer residence time in the system (e.g. Hammer et al., 1999, Stovall et al.,
2012; Gurioli et al., 2014) in agreement also with the vesicle signature. However, the four
typologies of clasts differ also in terms of mesocryst content (from rare to 5 vol. % for the
golden and fluidal and 14-23 vol. % for the glassy spiny and spiny opaque, respectively).
Equilibrium plagioclase-melt pairs record an almost constant and moderate dissolved water
content, intermediate between that expected for melts sitting in the main shallow reservoir
(located close to sea level) and the degassed matrix of lavas. Dissolved water contents are
thus consistent with pre-eruptive magma water degassing during its storage at shallow level,
as suggested by geophysical data, and suggest that the plagioclase mesocrysts and some of the
microcrysts in the spiny scoria and in the lava grew during magma storage (Fig. 10a). Melt

composition records a potential pre-eruptive thermal gradient of ~30 °C between the hotter (pumice and fluidal) and the cooler (spiny) magma (Fig. 10b).

Tait et al. (1989) suggest that magma evolution can lead to oversaturation of volatile species within a shallow reservoir and trigger a volcanic eruption. At PdF, the golden and the fluidal clasts might represent the portion of magma located at the top of the shallow reservoir and enriched in bubbles of water rich fluids, released by the cooler, more crystallized and more degassed "spiny-lava" magma (Fig. 10b). The small volume of magma, its constant bulk composition and the very small inflation recorded prior to the eruption (Fig. 1d) could be consistent with an internal source of over-pressure related to volatile exsolution. Larger inflation rates over a broader area are expected when shallow reservoir pressurization is related to a new magma input from a deeper source. Slight baseline extensions both on distal and proximal sites suggest that magma transfer towards shallower crustal levels started short before (11 days) the final magma eruption. Geochemical data do not support the occurrence of a new magma input in the degassed and cooled 2014 reservoir. We can thus speculate that stress field change related to progressive deep magma transfer has promoted volatile exsolution, melt-crystal separation and melt expansion in the shallow reservoir. Textural heterogeneity of the 2014 products partly reflects a pre-eruptive physical gradient recorded by the variability in crystal and bubble contents in the shallow reservoir feeding this eruption. The golden and fluidal fragments are the bubble richer and hotter portion of the melt. The spiny fragments are the degassed and cooler portion of the reservoir, whose progressive tapping led to a decrease in explosive intensity (from fountaining to Strombolian activity). Our results are also consistent with processes of mechanical reservoirs/dyke stratification, as observed by Menand and Phillips (2007). As explained earlier, magma ascent promoted syneruptive degassing induced crystallization. The spiny opaque clasts can be considered as being recycled material that fell back into the system. Accumulation of olivine crystals out of equilibrium with the host magma produces minor variations in mesocryst contents as observed within the same type of clasts sampled at different times/locations during the eruption, with the scoria from the WF and early erupted lava being the ones with the lowest amount of olivine (Table S4 and Fig. 7b). Again, this temporal variation supports an increase in large heavy crystals within the most degassed magma emitted toward the end of activity, further suggesting that it corresponds to the lower part of the reservoir.

Our dataset permits us to propose that the 2014 eruption was fed by a physically zoned magma reservoir. The low-density, crystal-poor, bubble-rich magma located in the upper part of the storage system, ascended first, rapidly and fed the early, more energetic phase, the

Hawaiian fountaining. This low-density magma is not more evolved than the spiny one (same
bulk compositions) and it is not necessarily richer in dissolved volatile amounts; it is just
poorer in crystal and richer in bubbles. Second boiling, possibly triggered a few days before
the eruption by stress field change, is responsible of the extraction of bubble rich melt from a
crystal-rich network. This last one will represent the main volume of the erupted lava. Fast
ascent of the foam hinders its crystallization and preserves high number of vesicles, high
vesicularity and it is only little modified by post-fragmentation expansion. Decrease in initial
overpressure translates in a progressive decrease in magma ascent rate and output rate (e.g.
Coppola et al., 2017 and references therein). Nucleation of microcrysts is enhanced in melt
ascending with lower speed and is mostly related to syneruptive degassing (for the spiny). The
larger volume (dense lava) corresponds to crystallized and less vesiculated magma which
experiences a slow ascent in the dyke and even further micro-crystallisation during its
subaerial emplacement.
Melt inclusion results allow us to confirm the involvement of a single and only slightly
heterogeneous magma source in 2014, related to cooling and fractional crystallisation of an
older magma batch (November 2009). Interestingly, this latter short lived summit eruption
was also characterized by the same large textural range of pyroclastic products found in 2014
in spite of its more mafic composition.
This suggests that bubble accumulation and source pressurisation is highly dependent
on the shallow storage depth, which facilitates rapid water exsolution (Di Muro et al., 2016),
and it is not necessarily the outcome of slow magma cooling and differentiation (Tait et al.,

979 1989).

**6. Proposed model for the 2014 eruption and conclusions**
In this paper we show that textural and petro-chemical study of the eruptive products can be
used to characterize the on-going activity at PdF and to constrain both the trigger and the
evolution of short-lived and small-volume eruptions. This approach is extremely valuable in i)
understanding processes that lead to an eruption which was preceded by short-lived and
elusive precursors, and ii) in reconstructing the time evolution of eruptive dynamics in an
eruption with poor direct observations.
Following the sketch in Figure 12, we infer that residual magma from the 2009
eruption ponding at shallow levels experienced long-lasting cooling and crystallization (Fig.
12a). Between 2010 and 2014 the volcano progressively deflated (Fig. 12b) possibly because
of magma degassing and cooling, facilitated by the shallow depth of the reservoir. During this
phase mesocrysts and some microcrysts formed (Figs. 4e and 10a).
The occurrence of deep (>10 km bsl) lateral magma transfer since March-April 2014
has been inferred by Boudoire et al., (2017) on the basis of deep (mantle level) seismic
swarms and increase in soil $CO_2$ emissions on the distal western volcano flank. The incipit of
magma transfer towards shallower crustal levels is potentially recorded by subtle volcano
inflation about 11 days before the June 2014 eruptions (Figs. 1d and 12c). We suspect that
these deep processes can have progressively modified the shallow crustal stress field and
favoured magma vesiculation and melt-crystal separation. Second boiling could thus have
over-pressured the shallow seated reservoir and triggered magma ascent (Fig. 12c).
Without this deep magma transfers we believe that the small reservoir activated in
2014 would have cooled down completely to form an intrusion (as suggested by the pervasive
crystallization of the lava, one of the densest emitted from 2014 to 2017, Harris et al. 2017).
The 2014 event represented instead the first of a long series of eruptions, whose magmas
became progressively less evolved in time (Coppola et al., 2017). In this scenario the trigger
mechanisms of 2014 activity are both internal and external in the sense that the small shallow
reservoir hosting cooled magma permitted to create the conditions favourable to a second
boiling (Fig. 12c, and Tait et al., 1989). The second boiling was likely triggered by an almost
undetectable stress field change, and was favoured by the shallow storage pressure of the
magma (Fig. 12c) that promoted fast water exsolution and rapid magma response to external
triggers. The second boiling possibly contributed to the inflation registered 11 days before the
eruption at 1.4-1.7 km (Fig. 12c) caused both by magma expansion and transfer of hot fluids
to the hydrothermal system (Lénat et al., 2011).
Our data permit to exclude (i) new magma input and/or fluid inputs (CO2-rich fluids)
from deep magmatic levels to trigger the June 2014 eruption. We also exclude (ii) heating and
enhanced convection of the shallow magma reservoir (due to heat diffusion without fluid or
mass transfer), because this process is very slow. Furthermore, the 2014 minerals do not
record evidences of magma heating. We can exclude equally (iii) deformation of the volcanic
edifice and decompression of the magma reservoir and/or hydrothermal system due to flank
sliding because geodetic data show no evidence of flank sliding able to produce stress change
in the hydrothermal and magmatic system. Geophysical and geochemical data have permitted
to track vertical magma and fluid transfer below the volcano summit in April 2015, that is
about one year after the early deep lateral magma transfer (Peltier et al., 2016). Deep
processes are difficult to detect for any monitoring network.
We conclude that the overpressure, caused by the second boiling, triggered the
eruption. The occurrence of a hydrous almost pure melt at shallow depth permitted its fast
vesiculation upon ascent towards the surface. In turn, fast ascent of the foam (Fig. 12d)
hindered its crystallization and preserved high number of vesicles. Decrease in initial
overpressure translated in a progressive decrease in magma ascent rate and output rate (e.g.
Coppola et al., 2017 and references therein) and a temporal transition from Hawaiian activity
to Strombolian activity (Fig. 12 d). Nucleation of microcrysts was enhanced in melt ascending
with lower speed and in turn this syn-eruptive crystallization favoured bubble
connectivity/permeability in the ascending magma, even for magma at low vesicularity. The
largest volume (dense lava) corresponds to highly-crystallized and degassed magma already
in the reservoir, that experienced a slower ascent in the dyke and even further micro-
crystallisation during its subaerial emplacement.
The texture of the products allowed us to follow the dynamic evolution of the system
in space, from smooth fluidal scoria emitted from rapid jet of lava at the fractures, to a more
stable activity at the MV, and in time. At the MV, in fact, we observed the transition from the
golden and fluidal fragments emitted from Hawaiian fountaining, at the peak of the intensity
of the eruption, to the spiny fragments, emitted from a declining Strombolian activity at the
end of the eruption.
Therefore we here show for the first time that short lived and weak Hawaiian
fountaining and Strombolian events can preserve pyroclast textures that can be considered a
valid approximation to shallow reservoir conditions and ascent degassing processes before the
explosions and correlate to the eruptive dynamics as well.
To conclude, these results highlight the importance of petrological monitoring, which
can provide complementary information regarding the ongoing volcanic activity to other
geophysical and geochemical monitoring tools commonly used on volcanoes.
**Acknowledgements**
OVPF team and T. Lecocq for monitoring and fieldwork. F. van Wyk de Vries provided an
English revision for the proof. We thank the STRAP project funded by the Agence Nationale
de la Recherche (ANR-14-CE03-0004-04). This research was financed by the French
Government Laboratory of Excellence initiative no. ANR-10-LABX-0006, the Région
Auvergne, and the European Regional Development Fund. This is Laboratory of Excellence
Clervolc contribution number XXXX

**References list**


Albarède, F., and V. Tamagnan (1988), Modelling the recent geochemical evolution of the
Piton de la Fournaise volcano, Réunion island, 1931-1986, *J. Petrol., 29*, 997-1030.
Albarède, F., B. Luais, G. Fitton, M.P. Semet, E. Kaminski, B.G.J Upton, P. Bachèlery, and
J.L. Cheminée (1997), The geo-chemical regimes of Piton de la Fournaise Volcano Réunion.
during the last 530,000 years, *J. Petrol., 38*, 171–201.
Andronico, D., M.D. Lo Castro, M. Sciotto, and L. Spina (2013a), The 2010 ash emissions at
the summit craters of Mt Etna: relationship with seismo-acoustic signals, *J. Geophys. Res.,*
*118*, 51–70, doi:10.1029/2012JB009895.
Andronico, D., J. Taddeucci, A. Cristaldi, L. Miraglia, P. Scarlato, and M. Gaeta (2013b), The
15 March 2007 paroxysm of Stromboli: video-image analysis, and textural and compositional
features of the erupted deposit, Bull. Volcanol., 75, 733, doi:10.1007/s00445-013-0733-2.
Andronico, D., S. Scollo, M.D. Lo Castro, A. Cristaldi, L. Lodato, and J. Taddeucci (2014),
Eruption dynamics and tephra dispersal from the 24 November 2006 paroxysm at South-East
Crater,    Mt    Etna,    Italy,    *J.    Volcanol.    Geotherm.    Res.,    274*,    78–91,
doi:10.1016/j.jvolgeores.2014.01.009.
Bachèlery, P., J.F. Lénat, A. Di Muro, and L. Michon (2016), Active Volcanoes of the
Southwest Indian Ocean: Piton de la Fournaise and Karthala. Active Volcanoes of the World.
Springer-Verlag, Berlin and Heidelberg, 1-428, DOI 10.1007/978-3-642-31395-0_12.
Boivin, P., and P. Bachèlery (2009), Petrology of 1977 to 1998 eruptions of Piton de la
Fournaise, La Réunion Island, *J. Volcanol. Geotherm. Res., 184*, 109–125.
Bombrun, M., A. Harris, L. Gurioli, J. Battaglia and V. Barra (2015), Anatomy of a
strombolian eruption: inferences from particle data recorded with thermal video, *J. Geophys.*
*Res., 120*(4):2367-2387. DOI.10.1002/2014BO11556.
Boudoire, G., M. Liuzzo, A. Di Muro, V. Ferrazzini, L. Michon, F. Grassa, A. Derrien, N.
Villeneuve, A. Bourdeu, C. Brunet, G. Giudice, and S. Gurrieri (2017), Investigating the
deepest part of a volcano plumbing system: evidence for an active magma path below the
western flank of Piton de la Fournaise (La Réunion Island), *J. Volcanol. Geotherm. Res.*, doi:
10.1016/j.jvolgeores.2017.05.026.
Brenguier, F., P. Kowalski, T. Staudacher, V. Ferrazzini, F. Lauret, P. Boissier, A. Lemarchand,
C. Pequegnat, O. Meric, C. Pardo, A. Peltier, S. Tait, N.M. Shapiro, M. Campillo, and A. Di
Muro (2012), First Results from the UnderVolc High Resolution Seismic and GPS network
deployed on Piton de la Fournaise Volcano, *Seismo. Res. Lett. 83*(7),
doi:10.1785/gssrl.83.1.97.
Brugier, Y.A. (2016), Magmatologie du Piton de la Fournaise (Ile de la Réunion): approche
volcanologique, pétrologique et expérimentale. Sciences de la Terre. Université d'Orléans,
NNT: 2016ORLE2007, pp. 251.
Bureau, H., F. Pineau, N. Métrich, P.M. Semet, and M. Javoy (1998a), A melt and fluid
inclusion study of the gas phase at Piton de la Fournaise volcano (Reunion Island), *Chem.*
*Geol. 147*, 115–130.
Bureau, H., N. Métrich, F. Pineau, and M.P. Semet (1998b), Magma-conduit interaction at
Piton de la Fournaise volcano (Réunion Island): a melt and fluid inclusion study, *J. Volcanol.*
*Geotherm. Res. 84*, 39–60.
Carey, R.J., M. Manga, W. Degruyter, D. Swanson, B. Houghton, T. Orr, and M. Patrick
(2012), Externally triggered renewed bubble nucleation in basaltic magma: the 12 October
2008 eruption at Halemaʻumaʻu Overlook vent, Kīlauea, Hawaiʼi, USA, *J. Geophys. Res.,*
*117*, B11202. doi:10.1029/2012JB009496.
Carey, R.J., M. Manga, W. Degruyter, H. Gonnermann, D. Swanson D, B. Houghton, T. Orr,
and M. Patrick (2013), Convection in a volcanic conduit recorded by bubbles, *Geology, 41*(4),
395–398.
Cashman, K.V., and M.T. Mangan (1994) Physical aspects of magmatic degassing II:
constraints on vesiculation processes from textural studies of eruptive products, In: Carroll
MR, Holloway JR (eds) Volatiles in magmas, Reviews in mineralogy. *Miner. Soc. Am.,*
Fredricksberg, pp 447–478.
Clocchiatti, R., A. Havette, and P. Nativel (1979), Relations pétrogénétiques entre les basaltes
transitionnels et les océanites du Piton de la Fournaise (Ile de La Réunion, océan Indien) à
partir e la composition chimique des inclusions vitreuses des olivines et des spinelles, *Bull.*
*Minér., 102*, 511–525.
Colombier, M., L. Gurioli, T.H. Druitt, T. Shea, P. Boivin, D. Miallier, and N. Cluzel (2017a),
Textural evolution of magma during the 9.4-ka trachytic explosive eruption at Kilian Volcano,
Chaîne des Puys, France, *Bull. Volcanol., 79*(2), 1-24. doi:10.1007/s00445-017-1099-7.
Colombier, M., F.B. Wadsworth, L. Gurioli, B. Scheu, U. Kueppers, A. Di Muro, and D.B.
Dingwel (2017b), The evolution of pore connectivity in volcanic rocks, *Earth Planet. Sci.*
*Lett., 462*, 99-109. DOI: 10.1016/j.epsl.2017.01.011.
Colò, L., M. Ripepe, D.R. Baker, and M. Polacci (2010), Magma vesiculation and infrasonic
activity at Stromboli open conduit volcano, *Earth Planet. Sc. Lett. 292*(3–4):274–280.
Coppola, D., N. Villeneuve, A. Di Muro, V. Ferrazzini, A. Peltier, M. Favalli, P. Bachèlery, L.
Gurioli, A. Harris, S. Moune, I. Vlastélic, B. Galle, S. Arellano, and A. Aiuppa (2017), A
Shallow system rejuvenation and magma discharge trends at Piton de la Fournaise volcano
(La Réunion Island), *Earth Planet. Sci. Lett. 463*, 13-24.
Corsaro, R., and L. Miraglia (2014), The transition from summit to flank activity at Mt. Etna,
Sicily (Italy): Inferences from the petrology of products erupted in 2007–2009, *J. Volcanol.*
*Geother. Res., 275*, 51– 60.
Darcy, H. (1856) Les Fontaines Publiques de la Ville de Dijon, Dalmont, Paris.
Di Muro, A., Métrich, N., Vergani, D., Rosi, M., Armienti, P., Fougeroux, T., Deloule, E.,
Arienzo, I., Civetta, L. (2014), The shallow plumbing system of Piton de la Fournaise Volcano
(La Réunion Island, Indian Ocean) revealed by the major 2007 caldera forming eruption, *J.*
*Petrol., 55*, 1287-1315.
Di Muro, A., T. Staudacher, V. Ferrazzini, N. Métrich, P. Besson, C. Garofalo, and B.
Villemant (2015), Shallow magma storage at Piton de la Fournaise volcano after 2007 summit
caldera collapse tracked in Pele's hairs, chap 9 of Carey, R. J., V. Cayol, M. P. Poland, and D.
Weis (eds.), Hawaiian Volcanoes: From Source to Surface, *American Geophysical Union*
*Monograph 208*, pp 189–212, doi:10.1002/9781118872079.ch9.
Di Muro, A., N. Métrich, P.Allard, A. Aiuppa, M. Burton, B. Galle, and T. Staudacher (2016),
Magma degassing at Piton de la Fournaise volcano, Active Volcanoes of the World, series,
Springer, Bachelery, P., Lenat, J.F, Di Muro, A., Michon L., Editors. Pg. 203-222.
DYNVOLC Database (2017) Observatoire de Physique du Globe de Clermont-Ferrand,
Aubière, France. DOI:10.25519/DYNVOLC-Database. Online access:
http://dx.doi.org/10.25519/DYNVOLC-Database
Eychenne, J., B.F. Houghton, D.A. Swanson, R.J. Carey, and L; Swavely (2015), Dynamics of
an open basaltic magma system: the 2008 activity of the Halemaʻumaʻu Overlook vent,
Kīlauea Caldera. *Earth Planet. Sci. Lett., 409*, 49–60.
Famin, V., B. Welsch, S. Okumura, P. Bachèlery, and S. Nakashima (2009), Three
differentiation stages of a single magma at Piton de la Fournaise (Réunion hotspot). *Geoch.*
*Geoph. Geos. 10*, Q01007. doi:10. 1029/2008GC002015.
Fisk, M.R., B.G.J Upton, C.E. Ford, and W.M. White (1988), Geochemical and experimental
study of the genesis of magmas of Reunion island, Indian Ocean, *J. Geophys. Res., 93*, 4933-

1154 4950.

Forchheimer, P. (1901) Wasserbewegung durch Boden, Z. Ver. Dtsch. Ing. 45:1781–1788.
Formenti, Y, and T.H. Druitt (2003), Vesicle connectivity in pyroclasts and implications for
the fluidisation of fountain-collapse pyroclastic flows, Montserrat (West Indies), *Earth Planet.*
*Sci. Lett., 214*, 561–574.
Gardner, J.E., R.M.E. Thomas, C. Jaupart, and S. Tait (1996), Fragmentation of magma
during Plinian volcanic eruptions, *Bull. Volcanol., 58*, 144–162.
Giachetti, T., T.H. Druitt, A. Burgisser, L. Arbaret, and C. Galven (2010), Bubble nucleation
and growth during the 1997 Vulcanian explosions of Soufrière Hills Volcano, Montserrat, *J.*
*Volcanol. Geotherm. Res., 193*(3–4):215–231. doi:10.1016/j.jvolgeores.2010.04.001.
Gonnermann, H.M., and M. Manga (2013) Dynamics of magma ascent in the volcanic
conduit. In: Fagents, S.A., Gregg, T.K.P., Lopes, R.M.C. (Eds.), Modeling Volcanic Processes:
The Physics and Mathematics of Volcanism. Cambridge University Press, Cambridge.
Gurioli, L., A.J.L. Harris, B.F. Houghton, M. Polacci, and M. Ripepe (2008) Textural and
geophysical characterization of explosive basaltic activity at Villarrica volcano, *J. Geophys.*
*Res., 113*, B08206. doi:10.1029/2007JB005328
Gurioli, L., A.J.L. Harris, L. Colo, J. Bernard, M. Favalli, M. Ripepe, and D. Andronico
(2013), Classification, landing distribution and associated flight parameters for a bomb field
emplaced during a single major explosion at Stromboli, Italy, *Geology, 41*, 559-562, DOI
10.1130/G33967.1.
Gurioli, L., L. Colo', A.J. Bollasina, A.J.L. Harris, A. Whittington, and M. Ripepe (2014),
Dynamics of strombolian explosions: inferences from inferences from field and laboratory
studies of erupted bombs from Stromboli volcano, *J. Geophys. Res., 119*(1),
DOI:10.1002/2013JB010355.
Gurioli, L., D. Andronico, P. Bachelery, H. Balcone-Boissard, J. Battaglia, G. Boudon, A.
Burgisser, S.B. M.R. Burton, K. Cashman, S. Cichy, R. Cioni, A. Di Muro, L. Dominguez, C.
D'Oriano, T. Druitt, A.J.L Harris, M. Hort, K. Kelfoun, J.C. Komorowski, U. Kueppers, J.L.
Le Pennec, T. Menand, R. Paris, L. Pioli, M. Pistolesi, M. Polacci, M. Pompilio, M. Ripepe,
O. Roche, E. Rose-Koga, A. Rust, L. Scharff, F. Schiavi, R. Sulpizio, J. Taddeucci, and T.
Thordarson (2015), MeMoVolc consensual document: a review of cross-disciplinary
approaches to characterizing small explosive magmatic eruptions, *Bull. Volcanol., 77*, 49.
DOI: 10.1007/s00445-015-0935-x.
Hammer, J.E., K.V. Cashman, R.P. Hoblitt, and S. Newman (1999) Degassing and microlite
crystallization during pre-climactic events of the 1991 eruption of Mt. Pinatubo, Philippines,
*Bull. Volcanol. 60*, 355–380.
Harris, A.J.L., N. Villeneuve, A. Di Muro, V. Ferrazzini, A. Peltier, D. Coppola, M. Favalli, P.
Bachèlery, J.-L. Foger, L. Gurioli, S. Moune, I. Vlastélic, B. Galle, and S. Arellano (2017),
Effusive Crises at Piton de la Fournaise 2014-2015: A Review of a Multi-National Response
Model, *Applied Volcanology, 6,* 11, DOI 10.1186/s13617-017-0062-9
Helz, R.T., and C.R. Thornber (1987), Geothermometry of Kilauea Iki lava lake, Hawaii, *Bull.*
*Volcanol., 49*, 651–668.
Hibert, C, A. Mangeney, M. Polacci, A. Di Muro, S. Vergniolle, V. Ferrazzini, B. Taisne, M.
Burton, T. Dewez, G. Grandjean, A. Dupont, T. Staudacher, F. Brenguier, N.M. Shapiro, P.
Kowalski, P. Boissier, P. Catherine, and F. Lauret (2015), Multidisciplinary monitoring of the
January 2010 eruption of Piton de la Fournaise volcano, La Réunion island, *J. Geophis. Res.,*
*120*(5), 3026-3047
Higgins M.-D. (2000). Measurement of crystal size distributions, *American Mineralogist*, 85,

1201    1105-1116.

Houghton, B.F., and C.J.N. Wilson (1989), A vesicularity index for pyroclastic deposits, *Bull.*
*Volcanol., 51*, 451–462. doi:10.1007/BF01078811
Houghton, B.F., D.A. Swanson, R.J. Carey, J Rausch., and A.J Sutton (2011), Pigeonholing
pyroclasts, insights from the 19 March 2008 explosive eruption of Kīlauea volcano,
*Geology,39*, 263–266, doi:10.1130/G31509.1.
Houghton, B.F., D.A. Swanson, J. Rausch, R.J. Carey, Fagents S.A., and T.R. Orr (2013),
Pushing the volcanic explosivity index to its limit and beyond: constraints from exceptionally
weak explosive eruptions at Kılauea in 2008, *Geology, 41*(6):627–630
Houghton, B.F., J. Taddeucci, D. Andronico, H.M. Gonnermann, M. Pistolesi, M.R. Patrick,
T.R. Orr, D.A. Swanson, M.Edmonds, D. Gaudin, R.J. Carey and P. Scarlato (2016), Stronger
or longer: Discriminating between Hawaiian and Strombolian eruption styles, *Geology* doi:
10.1130/G37423.1
Inman, D.L. (1952) Measures for describing the size distribution of sediments, *J. Sed. Petrol.,*
*22*, 125–145.
Kahl, M., S.Chakraborty, M. Pompilio, and F. Costa (2015), Constraints on the nature and
evolution of the magma plumbing system of Mt. Etna Volcano (1991–2008) from a combined
thermodynamic and kinetic modelling of the compositional record of minerals, *J.Petrol., 56*,
2025–2068, doi:10.1093/petrology/egv063.
Kawabata, E., S.J. Cronin, M.S. Bebbington, M.R.H. Moufti, N. El-Masry, and T. Wang
(2015), Identifying multiple eruption phases from a compound tephra blanket: an example of
the AD1256 Al-Madinah eruption, Saudi Arabia, *Bull. Volcanol., 77*, 6 DOI 10.1007/s00445-
014-0890-y.
Lange, R.A., H.M. Frey, and J. Hector (2009), A thermodynamic model for the plagioclase-
liquid hygrometer/thermometer, *Am. Mineral., 94*, 494–506.
Lautze, N., J. Taddeucci, D. Andronico, C. Cannata, L. Tornetta, P. Scarlato, B. Houghton, and
D. Lo Castro (2012), SEM-based methods for the analysis of basaltic ash from weak
explosive activity at Etna in 2006 and the 2007 eruptive crisis at Stromboli, *Phys. Chem.*
*Earth 45*,46, 113–127, doi:10.1016/j.pce.2011.02.001.
Leduc, L., L. Gurioli, A.J.L. Harris, L. Colo', and E. Rose-Koga (2015), Types and
mechanisms of strombolian explosions: characterization of a gas-dominated explosion at
Stromboli, *Bull. Volcanol., 77*, 8, DOI: 10.1007/s00445-014-0888-5
Lénat, J.-F., P. Bachèlery, and A. Peltier (2011), The interplay between collapse structures,
hydrothermal systems and magma intrusions: the case of the central area of Piton de la
Fournaise volcano, *Bull. Volc.* doi:10.1007/s00445-011-0535-3
Lénat, E.F., P.B. Bachelery, and O. Merle (2012), Anatomy of Piton de la Fournaise volcano
(La Réunion, Indian Ocean), *Bull. Volcanol. 74*, 1945–1961.
Lengliné, O, Z. Duputel, and V. Ferrazzini (2016), Uncovering the hidden signature of a
magmatic recharge at Piton de la Fournaise volcano using small earthquakes, *Geophys. Res.*
*Lett., 43*, doi: 10.1002/2016GL068383
Lindoo, A., J.F. Larsen, K.V. Cashman, A.L. Dunn, and O.K Neill (2016), An experimental
study of permeability development as a function of crystal-free melt viscosity, *Earth Planet.*
*Sci. Lett., 435*, 45–54, doi: 10 .1016 /j .epsl .2015 .11 .035.
Lindoo, A., J.F. Larsen, K.V. Cashman, and J. Oppenheimer (2017), Crystal controls on
permeability development and degassing in basaltic andesite magma, *Geology, 45*(9), p. 831-
1246  834.

Liuzzo, M., Di Muro, A., Giudice, G., Michon, L., Ferrazzini, V., and Gurrieri, S. (2015),
New evidence of $CO_2$ degassing anomalies on the Piton de la Fournaise volcano and the link
with volcano tectonic structures Geochemistry, Geophysics, Geosystems, 16, doi:10.1002/
2015GC006032.
Mangan, M.T., and K.V. Cashman (1996), The structure of basaltic scoria and reticulite and
inferences for vesiculation, foam formation, and fragmentation in lava fountains. *J. Volcanol.*
*Geotherm. Res., 73*, 1–18.

Menand, T., and J.C. Phillips (2007), Gas segregation in dykes and sills. *J. Volcanol. Geother. Res.*, *159*(4), 393–408. https://doi.org/10.1016/j.jvolgeores.2006.08.003.

Michon, L., A. Di Muro, N. Villeneuve, C. Saint-Marc, P. Fadda, and F. Manta (2013), Explosive activity of the summit cone of Piton de la Fournaise volcano (La Réunion Island): a historical and geological review, *J. Volcanol. Geotherm. Res. 263*, 117-133.

Moitra, P., H.M. Gonnermann, B.F. Houghton, and T. Giachetti (2013), Relating vesicle shapes in pyroclasts to eruption styles, *Bull. Volcanol.* 75, 691. doi:10.1007/s00445-013-0691-8

Morgan D.J., and D.A. Jerram (2006), On estimating crystal shape for crystal size distribution analysis, *J. Volc. Geotherm. Res., 154*, 1–7.

Moune, S., O. Sigmarsson, P. Schiano, T. Thordarson, and J.K. Keiding (2012), Melt inclusion constraints on the magma source of Eyjafjallajökull 2010 flank eruption, *J. Geophys. Res., 117*, B00C07, doi:10.1029/2011jb008718.

Morandi, A., C. Principe, A. Di Muro, G. Leroi, L. Michon, and P. Bachèlery (2016), Pre-historic explosive activity at Piton de la Fournaise volcano. In: Bachèlery P, Lénat JF, Di Muro A, Michon L (eds) Active Volcanoes of the Southwest Indian Ocean: Piton de la Fournaise and Karthala. Active Volcanoes of the World. Springer-Verlag, Berlin and Heidelberg, pp 107–138

Óladóttir, B., O. Sigmarsson, G. Larsen, and J.-L. Devidal (2011), Provenance of basaltic tephra from Vatnajökull subglacial volcanoes, Iceland, as determined by major- and trace-element analyses, *Holocene, 21*, 1037–1048, doi:10.1177/0959683611400456.

Oppenheimer, J., A.C. Rust, K.V. Cashman, and B. Sandnes (2015), Gas migration regimes and outgassing in particle-rich suspensions, *Frontiers in Physics, 3*, 1–13, doi: 10 .3389 /fphy .2015 .00060.

Ort, M.H., A. Di Muro, L. Michon, and P. Bachèlery (2016), Explosive eruptions from the interaction of magmatic and hydrothermal systems during flank extension: the Bellecombe Tephra of Piton de La Fournaise (La Réunion Island), *Bull. Volcanol. 78*, 5, doi:10.1007/s00445-015-0998-8.

Papale P., R. Moretti, and D. Barbato (2006), The compositional dependence of the saturation
surface of H2O + CO2 fluids in silicate melts, *Chemical Geology*, 229, 1/3, 78-95,
doi:10.1016/j.chemgeo.2006.01.013.
Parcheta, C.E., B.F. Houghton, and D.A. Swanson (2013), Contrasting patterns of vesiculation
in low, intermediate, and high Hawaiian fountains: a case study of the 1969 Mauna Ulu
eruption, *J. Volcanol. Geotherm. Res., 255*, 79–89
Peltier, A., P. Bachèlery, and T. Staudacher (2009), Magma transport and storage at Piton de la
Fournaise (La Réunion) between 1972 and 2007: A review of geophysical and geochemical
data. *J. Volcanol.Geother. Res., 184,* 93-108.
Peltier, A., F. Beauducel, N. Villeneuve, V. Ferrazzini, A. Di Muro, A. Aiuppa, A. Derrien, K.
Jourde, and B. Taisne (2016), Deep fluid transfer evidenced by surface deformation during the
2014–2015 unrest at Piton de la Fournaise volcano, *J. Volcanol. Geotherm. Res., 321*, 140–
148. http://dx.doi.org/10.1016/j.jvolgeores.2016.04.031.
Polacci, M., R. Corsaro, and D. Andronico (2006), Coupled textural and compositional
characterization of basaltic scoria: insights into the transition from Strombolian to fire
fountain activity at Mount Etna, Italy, *Geology, 34*(3), 201–204. doi:10.1130/G223181.1.
Polacci, M., C. Bouvet de Maisonneuve, D. Giordano, M. Piochi, L. Mancini L., W.
Degruyter, and O. Bachmanng (2014), Permeability measurements of Campi Flegrei
pyroclastic products: an example from the Campanian Ignimbrite and Monte Nuovo
eruptions. *J. Volcanol. Geotherm. Res. 272*, 16–22.
Putirka, K.D. (2008), Thermometers and barometers for volcanic systems, *Rev. Mineral.*
*Geochem. 69*, 61-120.
Reynolds, O. (1900) Papers on Mechanical and Physical Subjects, Cambridge University
Press.
Roeder, P., E. Gofton, and C. Thornber (2006), Cotectic proportions of olivine and spinel in
olivine-tholeiitic basalt and evaluation of pre-eruptive processes, *J. Petrol., 47*, 883-900.
Roult, G., A. Peltier, T. Staudacher, V. Ferrazzini, B. Taisne, A. Di Muro, and The OVPF
Team (2012), A comprehensive classification of the Piton de la Fournaise eruptions (La
Réunion Island) spanning the 1986–2010 period. Search for eruption precursors from the

broad-band GEOSCOPE RER station analysis and interpretation in terms of volcanic processes, *J. Volcanol. Geotherm. Res., 241*, 78–104.

Rust, A.C., and K.V. Cashman (2011), Permeability controls on expansion and size distributions of pyroclasts, *J. Geophys. Res., 116*, B11202.

Saar, M.O., M. Manga, K.V. Cashman, and S. Fremouw (2001) Numerical models of the onset of yield strength in crystal-melt suspensions: *Earth Planet. Sci. Lett., 187*, 367–379, doi: 10 .1016 /S0012 -821X (01)00289 -8.

Salaün, A., Villemant, B., Semet, M.P., and T. Staudacher (2010), Cannibalism of olivine-rich cumulate xenoliths during the 1998 eruption of Piton de la Fournaise (La Réunion hotspot): Implications for the generation of magma diversity. *J.Volcanol. Geother. Res., 198*, 187-204.

Schiano, P., K. David, I. Vlastélic, A. Gannoun, M. Klein, F. Nauret, and Bonnand P. (2012), Osmium isotope systematics of historical lavas from Piton de la Fournaise (Réunion Island, Indian Ocean), *Contrib. Mineral. Petrol.,* http://dx.doi.org/10.1007/s00410-012-0774-0.

Shea, T., (2017) Bubble nucleation in magmas: a dominantly heterogeneous process? *J. Volcanol. Geotherm. Res. 343,* 155–170.

Shea, T., B.F. Houghton, L. Gurioli, K.V. Cashman, J.E. Hammer, and B. Hobden (2010), Textural studies of vesicles in volcanic rocks: an integrated methodology, *J. Volcanol. Geotherm. Res., 190*, 271–289.

Shea, T., L. Gurioli, and B.F. Houghton (2012), Transitions between fall phases and pyroclastic density currents during the AD 79 eruption at Vesuvius: building a transient conduit model from the textural and volatile record, *Bull. Volcanol., 74*, 2363–2381, doi:10.1007/s00445-012-0668-z.

Sparks, R.S.J. (1978), The dynamics of bubble formation and growth in magmas: a review and analysis, *J. Volcanol. Geotherm. Res., 3*, 1–37.

Sparks, R.S.J. (2003). Forecasting volcanic eruptions, *Earth Planet. Sci. Lett., 210*, 1–15.

Spina, L., C. Cimarelli, B. Scheu, D. Di Genova, and D. B. Dingwell (2016) On the slow decompressive response of volatile- and crystal-bearing magmas: An analogue experimental investigation, *Earth Planet. Sci. Lett., 433*, 44-53.

Staudacher, T., and A. Peltier (2015), Ground deformation at Piton de la Fournaise (La
Réunion Island), a review from 20 years of GNSS monitoring, In: Bachèlery P, Lénat, JF, Di
Muro A, Michon L (ed) Active volcanoes of the Southwest Indian Ocean: Piton de la
Fournaise and Karthala. Active volcanoes of the world. Springer, Berlin, 139-170.
doi:10.1007/978-3-642-31395-0_9
Staudacher, T., V. Ferrazzini, A. Peltier, P. Kowalski, P. Boissier, P. Catherine, F. Lauret, and
F. Massin (2009), The April 2007 eruption and the Dolomieu crater collapse, two major
events at Piton de la Fournaise (La Re¤union Island, Indian Ocean). *J. Volcanol. Geother. Res.*
*184*, 126-137, doi:10.1016/j.jvolgeores.2008.11.005.
Stovall, W.K., B.F. Houghton, H.M. Gonnermann, S.A. Fagents, and D.A. Swanson (2011),
Eruption dynamics of Hawaiian-style fountains: the case study of episode 1 of the Kīlauea Iki
1959 eruption, *Bull. Volcanol. 73*, 511–529. doi:10.1007/s00445-010-0426-z.
Stovall, W.K., B.F. Houghton, J.E. Hammer, S.A. Fagents, and D.A. Swanson (2012),
Vesiculation of high fountaining Hawaiian eruptions: episodes 15 and 16 of 1959 Kīlauea Iki,
*Bull. Volcanol., 74*, 441–455, doi:10.1007/s00445-011-0531-7.
Swanson, D.A., K. Wooten, and T. Orr (2009), Buckets of ash track tephra flux from
Halemaʻumaʻu crater, Hawaiʻi. *Eos Trans. AGU, 90*, 427–428. doi:10.1029/2009EO460003.
Taddeucci, J., M. Pompilio, and P. Scarlato (2002), Monitoring the explosive activity of the
July–August 2001 eruption of Mt. Etna (Italy) by ash characterization, *Geophys. Res. Lett.,*
*29*(8), 1029–1032. doi:10. 1029/2001GL014372.
Tait, S., C. Jaupart, and S. Vergniolle (1989), Pressure, gas content and eruption periodicity of
a shallow, crystallising magma chamber, *Earth Planet. Sci. Lett., 92*, 107-123.
Takeuchi, S., S. Nakashima, and A. Akihiko Tomiya (2008) Permeability measurements of
natural and experimental volcanic materials with a simple permeameter: toward an
understanding of magmatic degassing processes, *J. Volcanol. Geotherm. Res.*, 177:329–339.
Thornber, C.R., K. Hon, C. Heliker, and D.A. Sherrod (2003), A Compilation of Whole-Rock
and Glass Major-Element geochemistry of Kīlauea Volcano, Hawaiʻi, near-vent eruptive
products: January 1983 through September 2001: *U.S.G.S. Open File Report*, *03*-477.

Thordarson, T, S Self, N Óskarsson, and T Hulsebosch (1996), Sulfur, chlorine and fluorine degassing and atmospheric loading by the 1783–1784 AD Laki (Skaftár Fires) eruption in Iceland, *Bull. Volcanol. 58*, 205–225.

Villemant, B., A. Salaün, and T. Staudacher (2009), Evidence for a homogeneous primary magma at Piton de la Fournaise (La Réunion): A geochemical study of matrix glass, melt inclusions and Pélé's hairs of the 1998–2008 eruptive activity, *J. Volcanol. Geotherm. Res., 184*, 79–92.

Vlastélic, I., and A.J. Pietruszka (2016), A review of the recent geochemical evolution of Piton de la Fournaise Volcano (1927–2010). In: Bachèlery, P., Lénat, J.F., Di Muro, A., Michon, L. (Eds.), Active Volcanoes of the Southwest Indian Ocean. In: Active Volcanoes of the World, pp.185–201.

Vlastélic, I., A. Peltier, and T. Staudacher (2007), Short-term (1998-2006) fluctuations of Pb isotopes at Piton de la Fournaise volcano (Réunion Island): origins and constraints on the size and shape of the magma reservoir, *Chem. Geology, 244*, 202-220.

Vlastélic, I., C. Deniel, C. Bosq, P. Telouk, P. Boivin, P. Bachèlery, V. Famin,. and T. Staudacher (2009), Pb isotope geochemistry of Piton de la Fournaise historical lavas, *J. Volcanol. Geother. Res., 184*, 63-78.

Vlastélic, I., T. Staudacher, P. Bachèlery, P. Télouk, D. Neuville., and M. Benbakkar (2011) Lithium isotope fractionation during magma degassing: constraints from silicic differentiates and natural gas condensates from Piton de la Fournaise volcano (Réunion Island), *Chemical Geology, 284*, 26–34.

Vlastélic, I., G. Menard, M. Gannoun, J.-L Piro., T Staudacher, and V. Famin (2013), Magma degassing during the April 2007 collapse of Piton de la Fournaise: the record of semi-volatile trace elements (Li, B, Cu, In, Sn, Cd, Re, Tl, Bi), *J. Volcanol. Geother. Res., 254*, 94-107.

Vlastélic, I., A. Gannoun, A. Di Muro, L. Gurioli, P. Bachèlery, and J.M. Henot (2016), Origin and fate of sulfide liquids in hotspot volcanism (La Réunion): Pb isotope constraints from residual Fe–Cu oxides, *Geochim. Cosmochim. Acta, 194*, 179-192.

Welsch, B., F. Faure, P. Bachèlery, and V. Famin (2009), Microcrysts record transient convection at Piton de la Fournaise volcano (La Réunion Hotspot), *J. Petrol., 50*, 2287-2305.

Welsch, B., V. Famin, A. Baronnet, and P. Bachèlery (2013), Dendritic crystallization: a single
process for all textures of olivine in basalts? *J. Petrol., 54*, 539-574.
White, J.D.L., and B.F. Houghton (2006), Primary volcaniclastic rocks, *Geology, 34*, 677–
680, doi:10.1130/G22346.1.

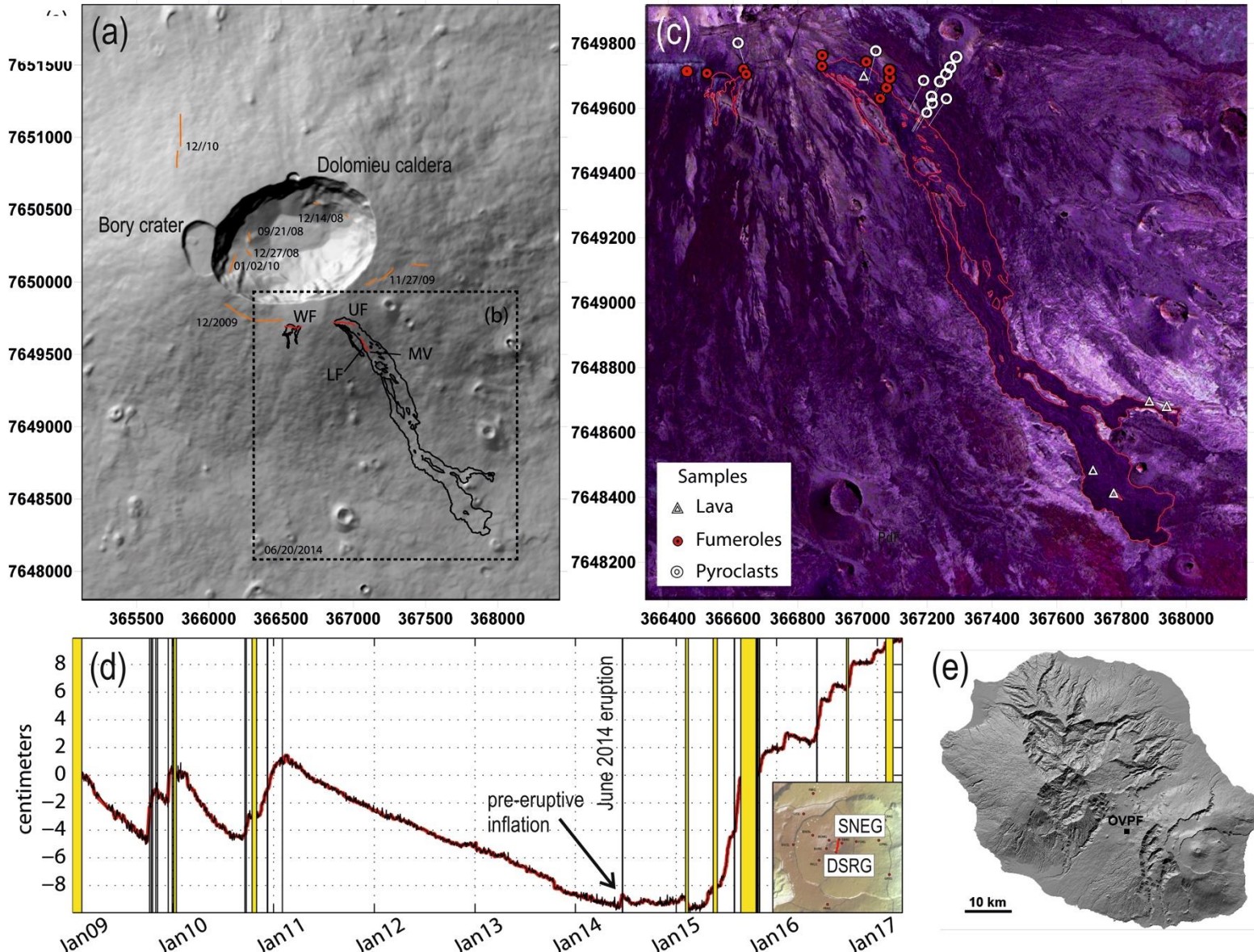

**Figure 1** a) Digital elevation model of the summit crater area at Piton de la Fournaise, La Réunion, France; orange = fractures generated by pre-
2014 eruptions (reported are the dates of their activities); b) red = fractures active during the 2014 eruption: WF (Western Fracture), UF (Upper
Fracture), LF (Lower Fracture), MV (Main Vent). Black= outline of the 2014 lava field; c) locations of sample collection points. The coordinates
are in UTM, zone 40 south. (d) Distance change (baseline) in centimetres between two GNSS summit stations: DSRG and SNEG (see location in
the inset). Increase and decrease of the signal mean a summit inflation and deflation, respectively. The yellow areas represent eruptive and
intrusive periods. In Figure 1d, the rapid and strong variations linked to dike injections preceding intrusions and eruptions by a few tens of
minutes have been removed; (e) Digital Elevation Model of La Réunion island.


# June 2014 eruption at PdF

Early morning, June 21

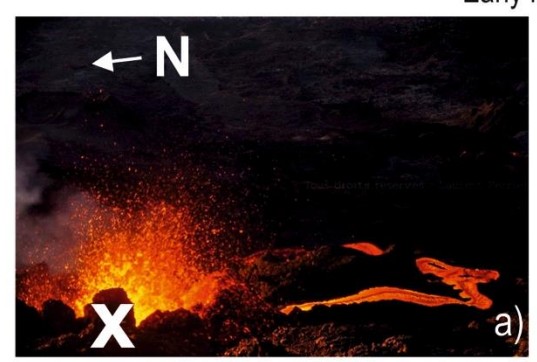

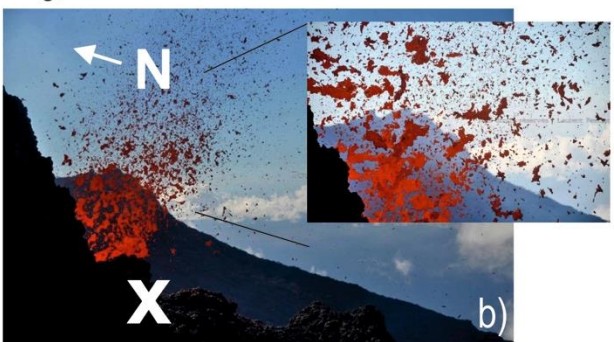

June 21 ~ 7h00

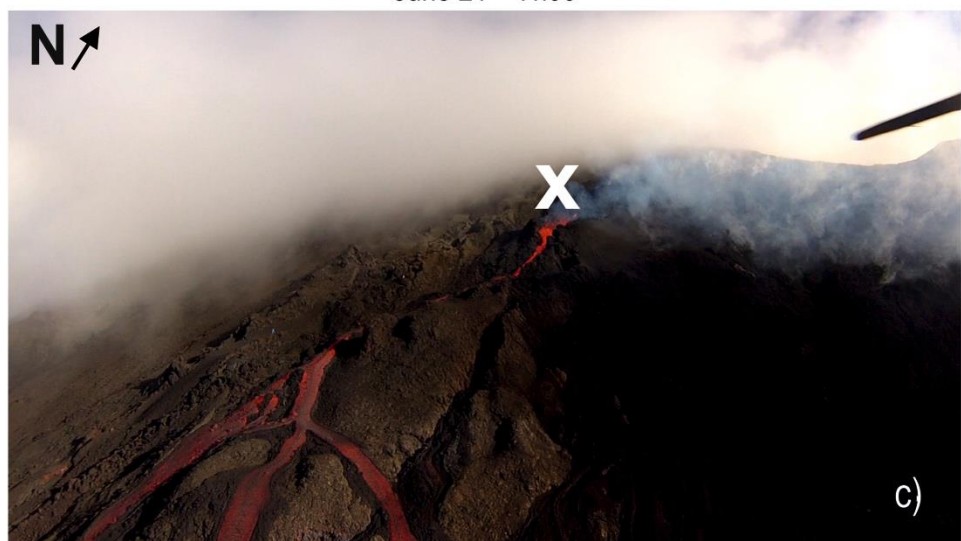

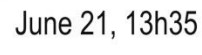

June 21, 7h38

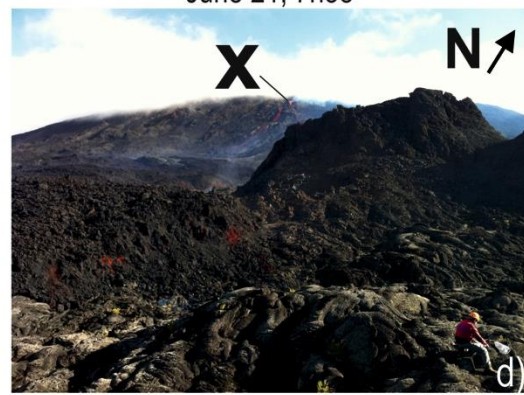

June 21, 13h35

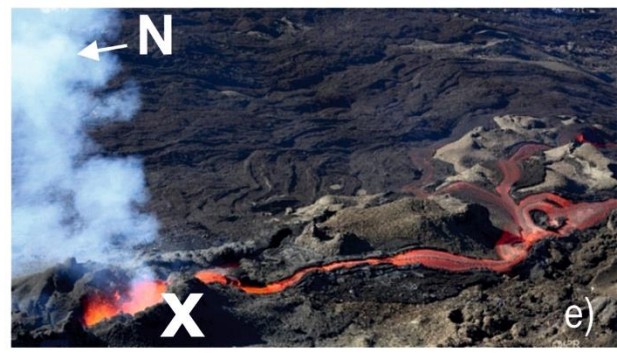

June 21,17h00

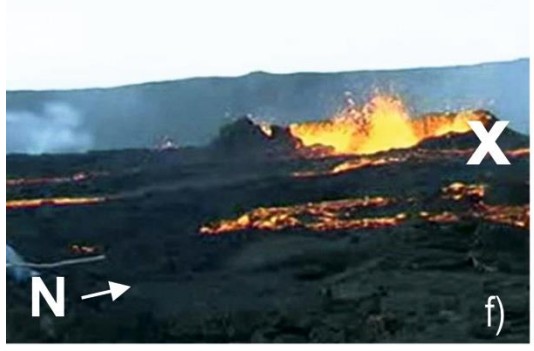

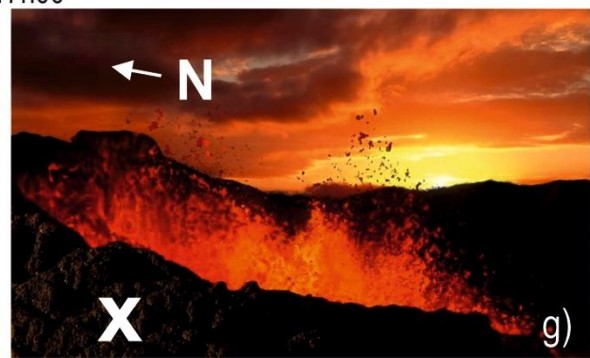


**Figure 2** Photos collection from the 2014 eruption at the MV, highlighted with a white cross (see location in Fig. 1). From a to g: evolution of the Strombolian activity from early morning to evening, June 21 that shows a decline in the activity with time. Unfortunately, the relatively more energetic Hawaiian fountaining events that happened during the night were not documented. a) Strombolian activity at the MV and associated lava flow; b) zoom view of the Strombolian activity at the MV. The images in a, b and the inset in b are from Laurent Perrier; c) aerial view of the SE flank of the PdF, taken by the OVPF team from the helicopter of the gendarmerie of La Réunion; d) Eastern front of the lava where the OVPF team collected a quenched lava sample; e) low Strombolian activity at the MV and the associated lava flow, photo from: http://www.ipreunion.com/volcan/reportage/2014/06/21/eruption-du-piton-de-la-fournaise-actualise-a-17h-la-lave-coule-sur-1-5-kilometre,26023.html; f) and g) decline of the Strombolian activity at the MV, the photo in e) is from http://www.zinfos974.com/L-eruption-du-Piton-de-la-Fournaise-Le-point-de 17h_a72981.html; and the photo if f) is from: f) http://nancyroc.com/eruption-a-la-reunion

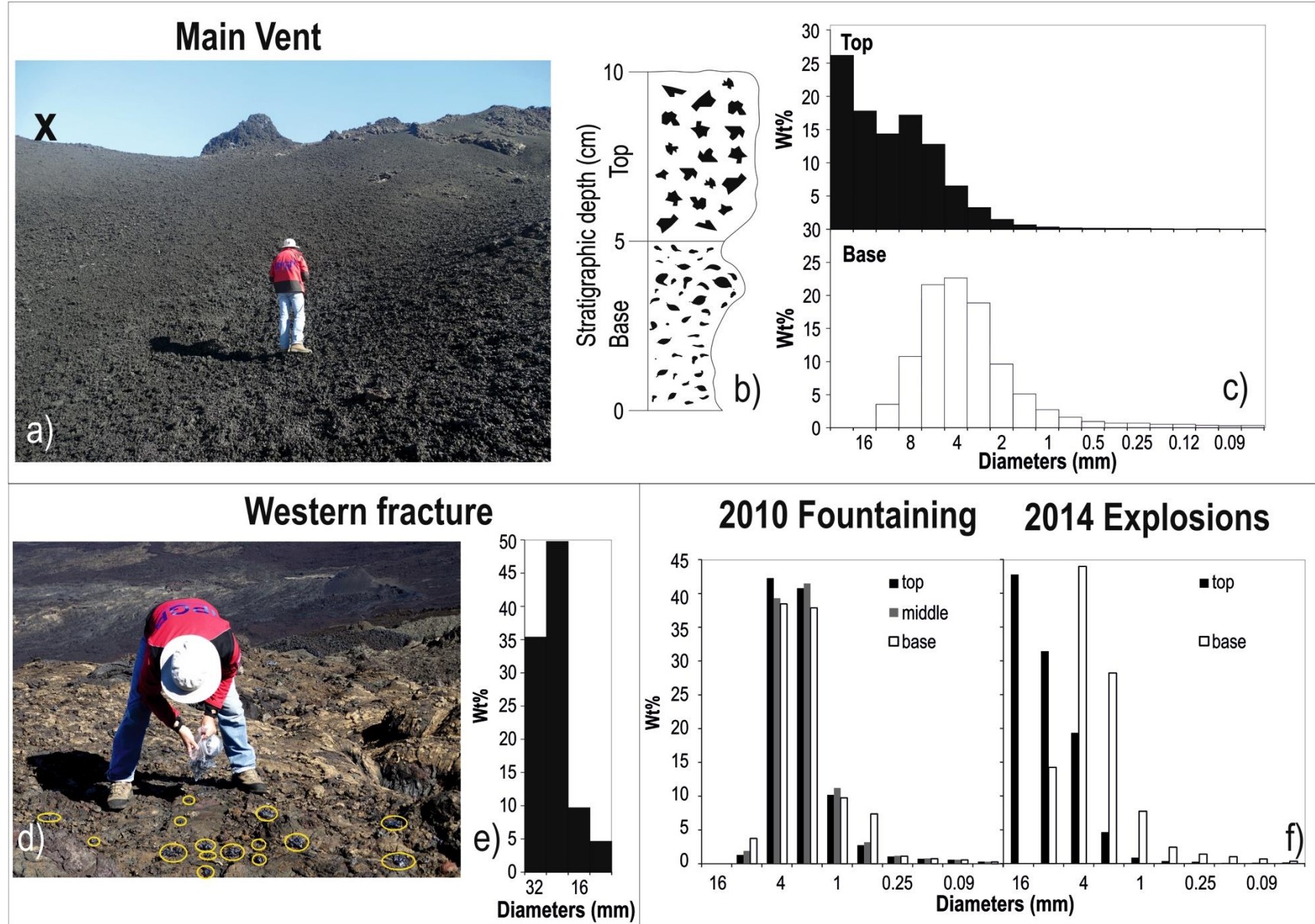

**Figure 3** a) Continuous blanket of scoria fall out deposit emitted from the MV (Fig. 1 for location) during June 2014 eruption at PdF. The black
cross locates the position of the MV (see Fig. 1 for the location); b) schematic stratigraphic log of the scoria fall out deposit emplaced during
June 2014 eruption at the MV. c) grain size histograms of the base and the top of the deposit of the MV, the particle diameters are at half phi; d)
scattered scoria (outlined in yellow) from the WF (see Fig. 1 for the location); e) grain size histogram of the scoria deposit at the WF, the particle
diameters are at half phi; f) comparison between the grain size histograms for the 2010 Hawaiian fountaining and the 2014 MV activity, both the
particle axes are reported in full phi for comparison.










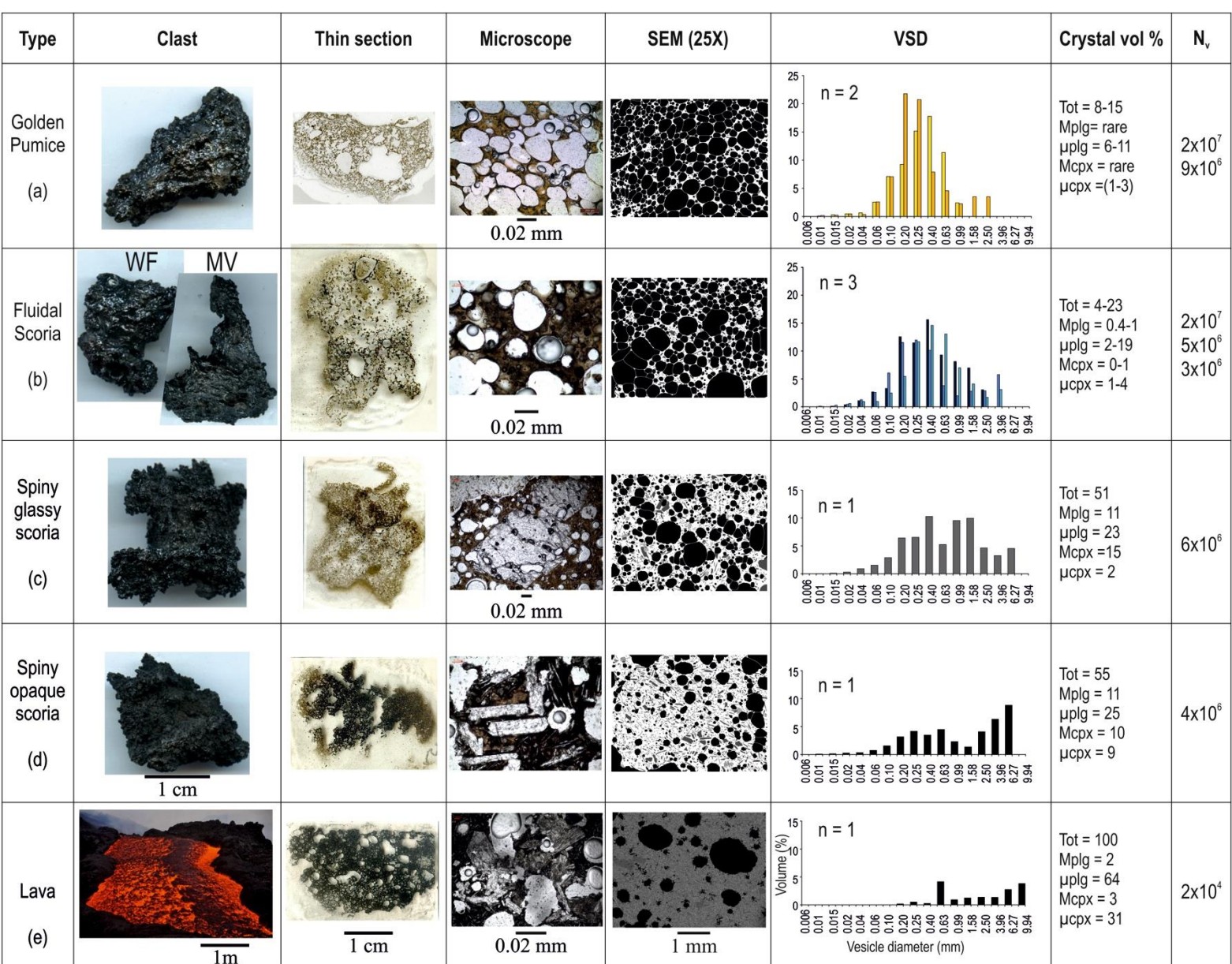

| Type | Clast | Thin section | Microscope | SEM (25X) | VSD | Crystal vol % | $N_v$ |
|------|-------|--------------|------------|-----------|-----|---------------|-------|
| Golden Pumice (a) | | | 0.02 mm | | n = 2 | Tot = 8-15<br>Mplg= rare<br>µplg = 6-11<br>Mcpx = rare<br>µcpx =(1-3) | $2 \times 10^7$<br>$9 \times 10^6$ |
| Fluidal Scoria (b) | WF  MV | | 0.02 mm | | n = 3 | Tot = 4-23<br>Mplg = 0.4-1<br>µplg = 2-19<br>Mcpx = 0-1<br>µcpx = 1-4 | $2 \times 10^7$<br>$5 \times 10^6$<br>$3 \times 10^6$ |
| Spiny glassy scoria (c) | | | 0.02 mm | | n = 1 | Tot = 51<br>Mplg = 11<br>µplg = 23<br>Mcpx =15<br>µcpx = 2 | $6 \times 10^6$ |
| Spiny opaque scoria (d) | 1 cm | | | | n = 1 | Tot = 55<br>Mplg = 11<br>µplg = 25<br>Mcpx = 10<br>µcpx = 9 | $4 \times 10^6$ |
| Lava (e) | 1m | 1 cm | 0.02 mm | 1 mm | n = 1<br>Vesicle diameter (mm) | Tot = 100<br>Mplg = 2<br>µplg = 64<br>Mcpx = 3<br>µcpx = 31 | $2 \times 10^4$ |


**Figure 4** Textural features of June 2014 pyroclasts and lava. Clast = photo of the different types of juvenile pyroclasts and lava channel. The
photo of the lava channel is from Laurent Perrier. WF = Western Fracture (smooth fluidal scoria), MV = Main Vent (fluidal scoria, less smooth
than the ones at the WF). Thin section = thin section imaged with a desktop scanner. Microscope = picture taken with an optical microscope
using natural light; SEM (25X) = image captured using a scanning electron microscopy (SEM), in BSE mode at 25x magnification: black are
vesicles, white is glass, grey are crystals. VSD = vesicle size distribution histograms, where the diameter, in mm, is plotted versus the volume
percentage, n = number of measured clasts; Crystal vol. % : Tot =  total percentage of crystals corrected for the vesicularity; Mplg = percentage
of mesocrysts of plagioclase; μplg = percentage of microcrysts of plagioclase; Mcpx = percentage of mesocrysts of pyroxene; μcpx = percentage
of microcrysts of pyroxene; Nv = number density corrected for the vesicularity.



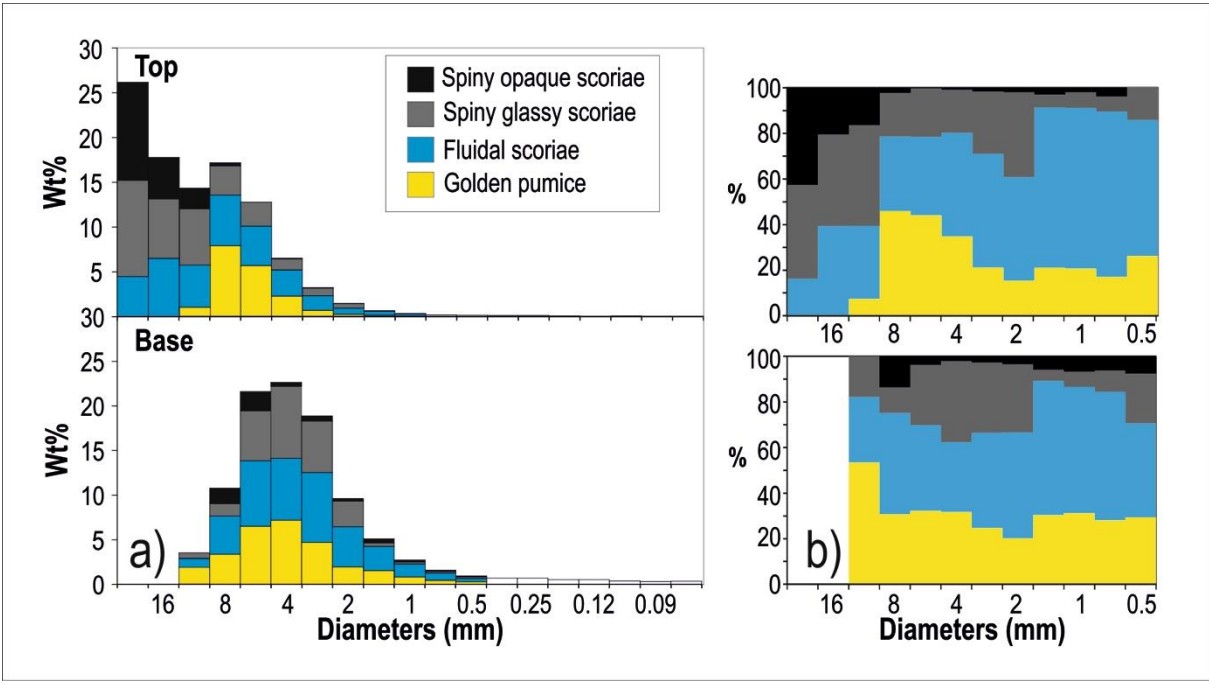


**Figure 5** Proportion of each type of clast measured from the base to the top of the 10 cm thick
deposit emplaced during the eruption, at the MV site. The deposit is dominated by Hawaiian-
like lapilli fragments at the base (golden pumice and fluidal scoria) and Strombolian-like
bombs and lapilli at the top (spiny scoria): (a) componentry within the different grain size
classes; b) normalized componentry composition from the base to the top of the deposit.









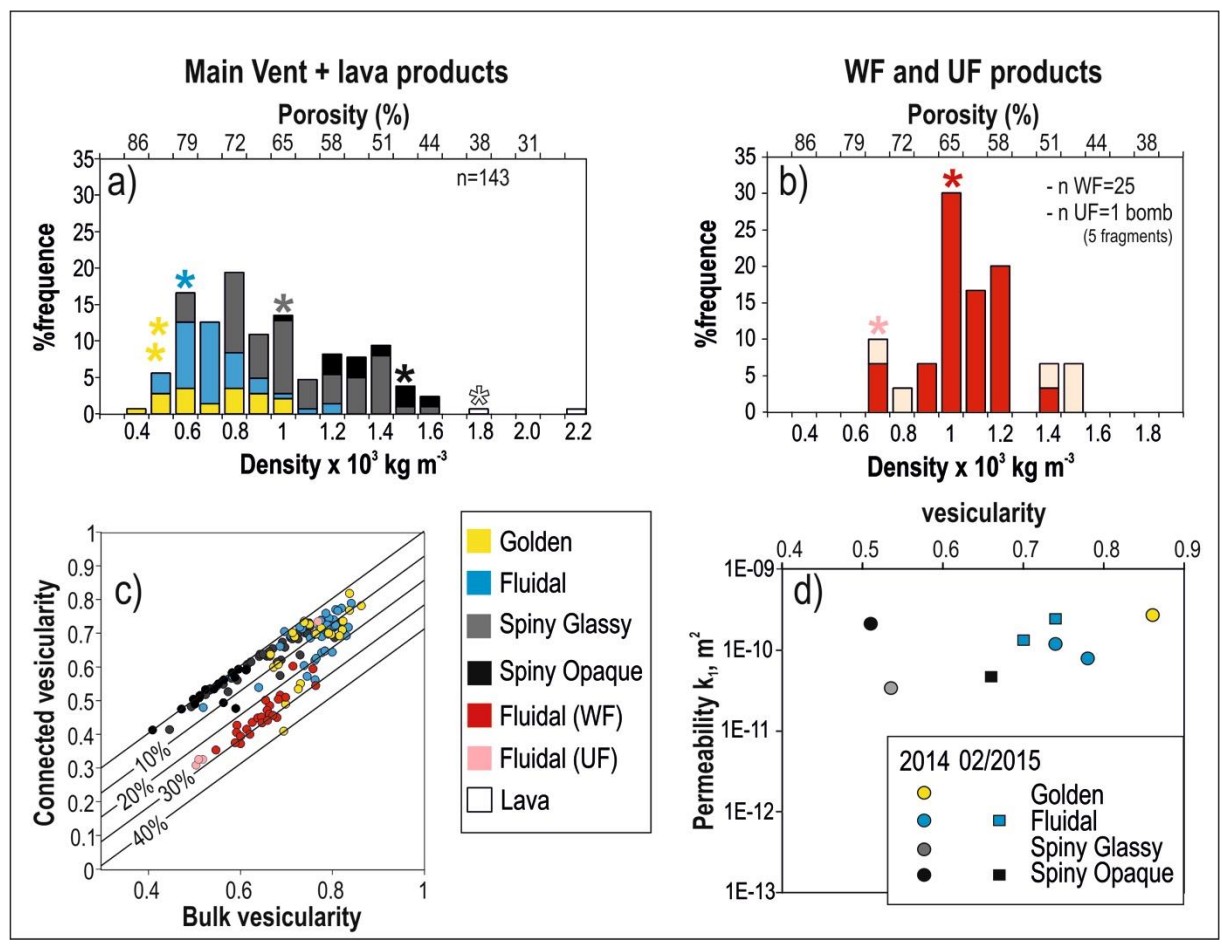

**Figure 6** Density, connectivity and permeability data of June 2014 pyroclast and lava fragments: a) density distribution histogram for all the pyroclast fragments measured at the MV + two lava fragments collected from the Eastern front of the lava flow (see Fig. 1 for location). n = number of measured clasts; b) density distribution histogram for the pyroclasts sampled at the WF and the bomb sampled at the UF. The bomb broke in five fragments (2 fragments from the core, the least dense, and three fragments from the quenched edges, the densest). In both the density histograms the stars represent the density intervals from which we picked the clasts for the textural measurements; c) graph of the connected vesicularity versus total vesicularity. The diagonal line represents equality between the connectivity and vesicularity, beneath this line the samples have isolated vesicles and the straight lines represent lines of equal fraction of isolated vesicles. To note that the bomb from the UF has the high vesicular core with less than 5% of isolated vesicles, while the three low vesicular fragments from the quenched edge have more than 25% of isolated vesicles (see pink spots); d) Darcian viscous permeability ($k_1$) versus vesicularity fraction for the four typologies of clasts collected at the MV. For comparison, two fluidal fragments and one spiny opaque fragments from February 2015 eruption are reported.

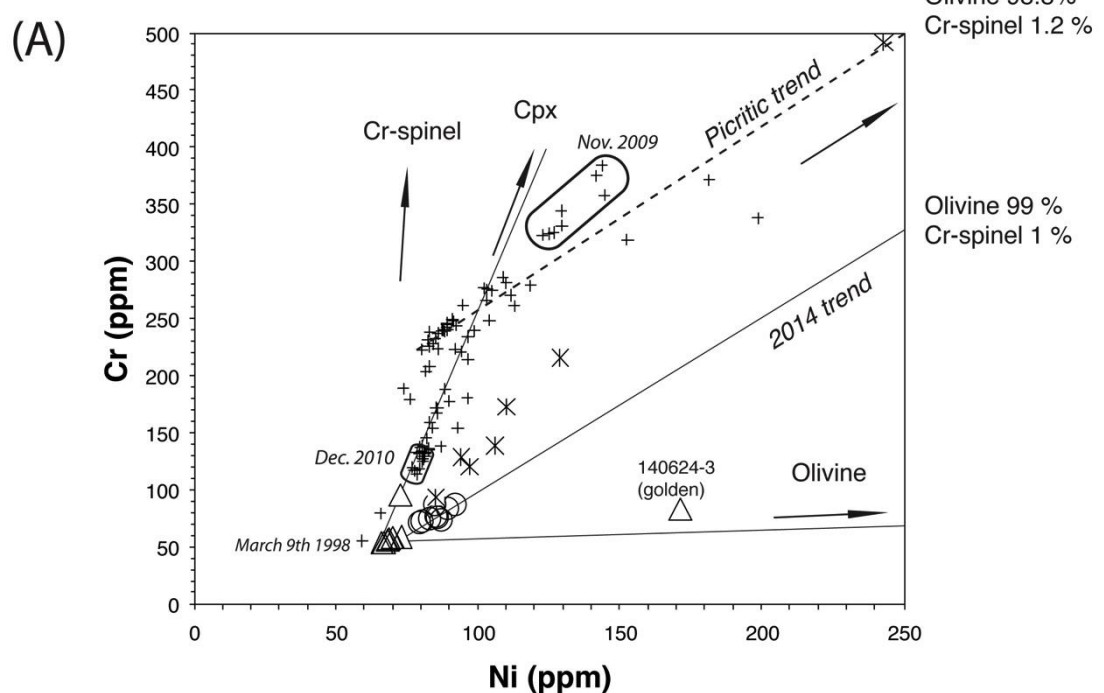

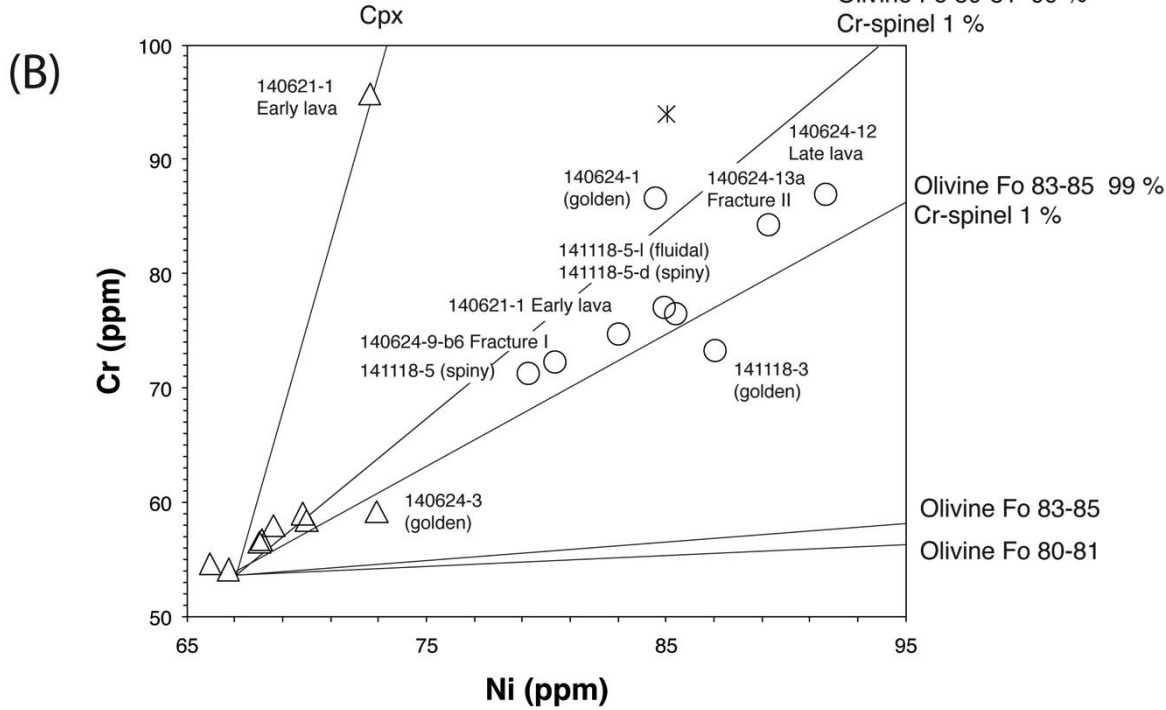

**Figure 7** Ni-Cr concentration plot. (a) Ni-Cr signature of the June 2014 lavas compared to that of recent eruptions (Di Muro et al. (2015) and unpublished data). Whole-rock (circles) and glass (triangles) compositions are shown for the June 2014 eruption. Olivine controlled lines are indicated for olivine hosting 1.2 and 0.6 wt.% Cr-spinel. Compositions used for olivine (Ni=1900 ppm, Cr=300ppm), clinopyroxene (Ni=970 ppm, Cr=4800 ppm), and Cr spinel (Ni=1500 ppm, Cr=25%) are inferred from Welsch et al. (2009), Salaün et al. (2010), and Di Muro et al. (2015). (b) Zoom of the Ni-Cr relationship between glass (triangles) and whole-rock (circles) samples from the June 2014 eruption. Fracture I = Western Fracture, Fracture II = Upper Fracture. Careful sample selection has permitted to obtain a set of virtually olivine-cpx free crystals. Any addition of mafic crystals translates into enrichment in Ni-Cr; those samples that contain a few % of crystals (consistent with textural and petrological observation) are slightly enriched in compatible elements.

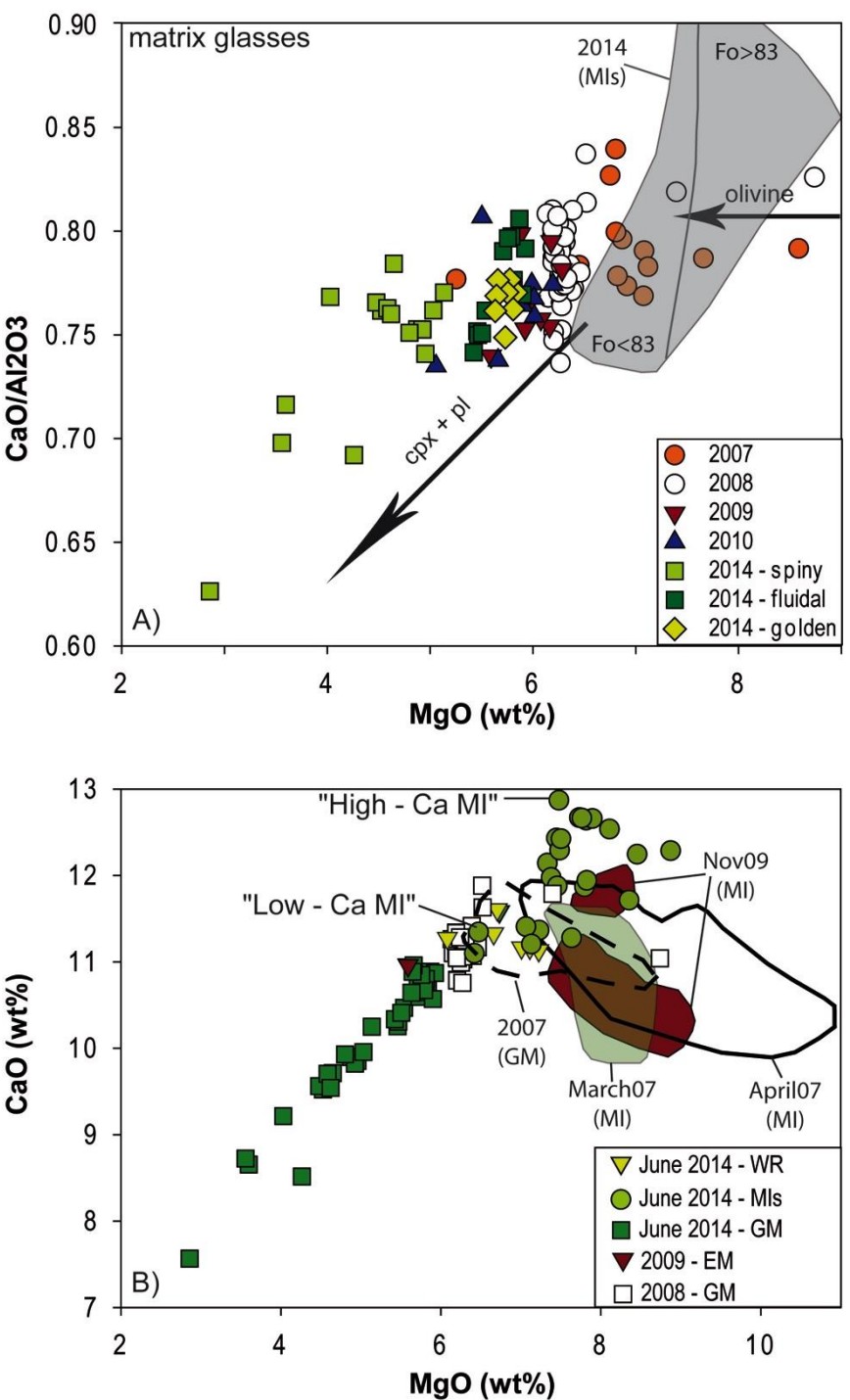

1515

**Fig. 8** (a) Evolution of CaO/Al$_2$O$_3$ ratio in the matrix glasses of recent eruptions at Piton de la Fournaise as a function of MgO content (directly proportional to melt temperature). MI = Melt inclusions (grey area for the 2014 samples). (b) CaO versus MgO content for Piton de la Fournaise products. WR = whole rock, GM = ground mass; MI = melt inclusion, EM = embayment glass


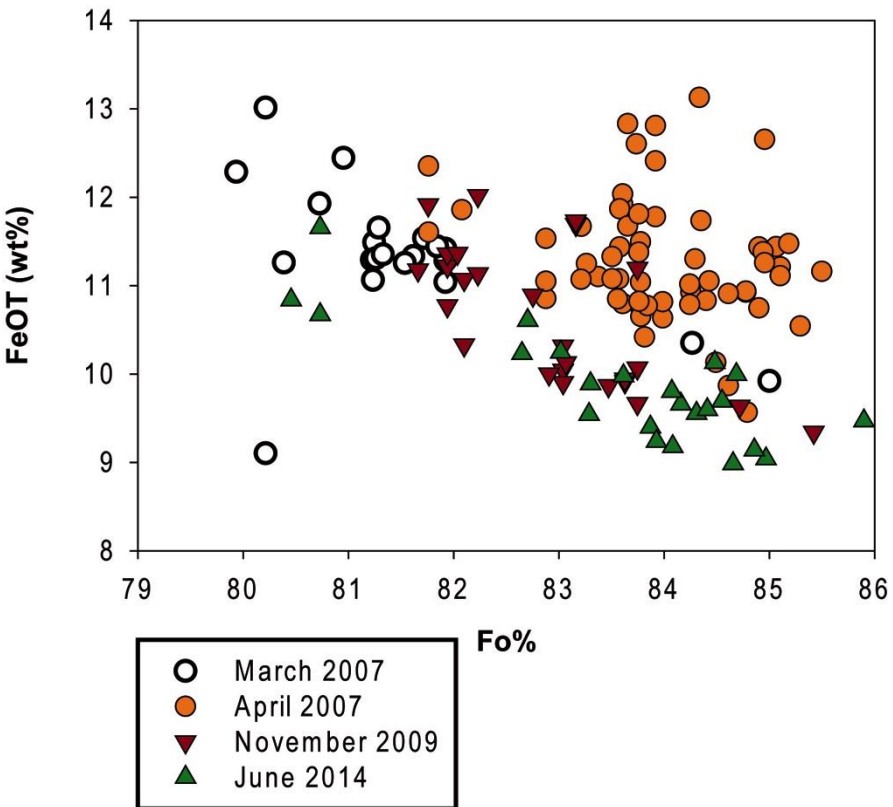


**Figure 9** FeO$_T$ in melt inclusions as function of Fo content of the olivine host for recent
eruptions at Piton de la Fournaise











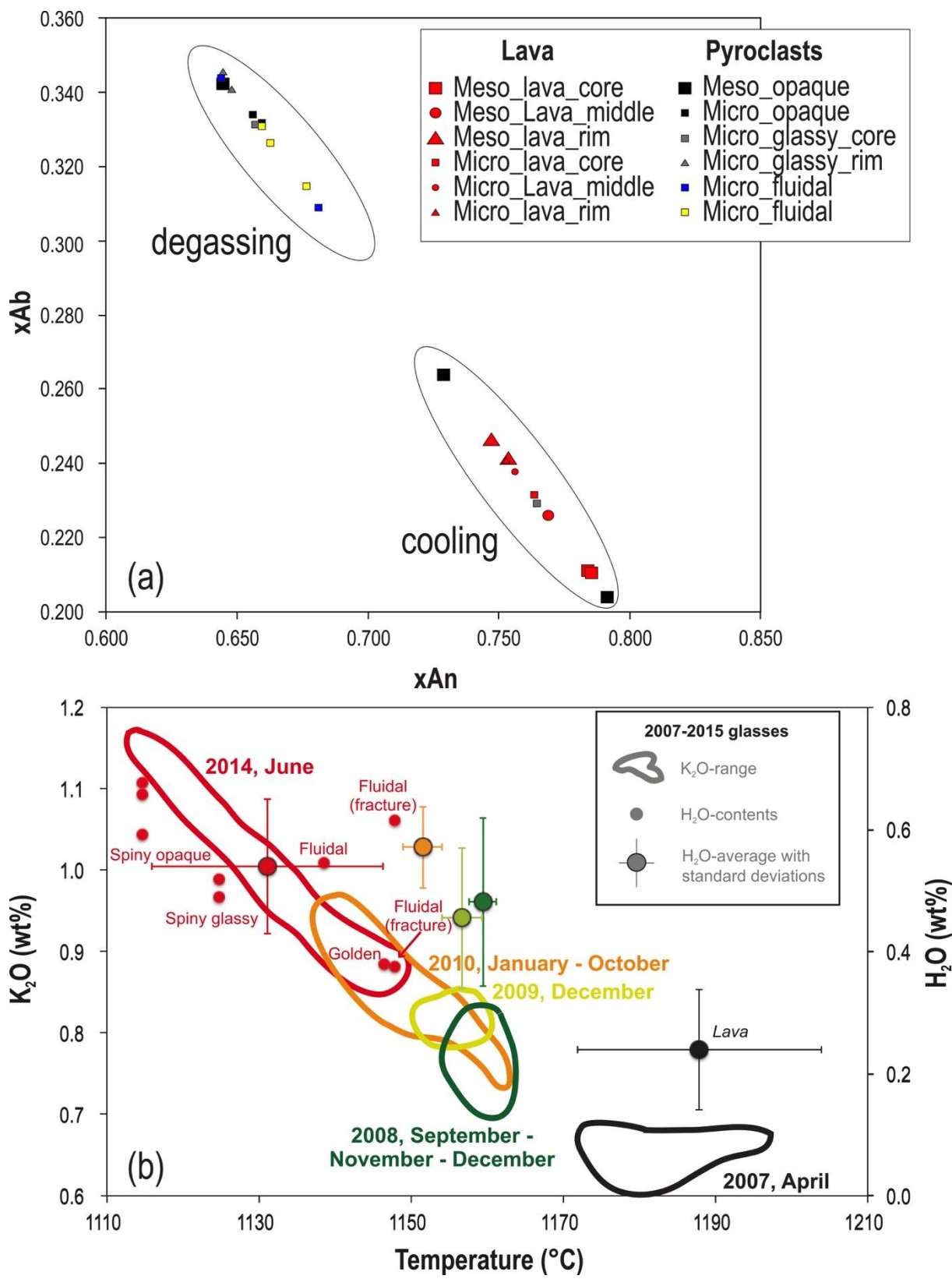

**Figure 10** a) Anorthite versus Albite compositions for the plagioclase crystals measured for June 2014 eruption of PdF; b) Temperature, composition ($K_2O$) and dissolved water content ($H_2O$) for the evolution of 2007-2014 melts from glasses. The data have been obtained by

studying the glass-plagioclase equilibrium or on the basis of matrix glass analyses. Temperature estimation based on the MgO-thermometer of Helz and Thornber (1987) modified by Putirka (2008). Water content is from the plagioclase hygrometer of Lange et al. (2009). Only plagioclases in equilibrium with melts are considered, following the procedure described by Putirka (2008) for >1050°C melts (Kd = 0.27±0.05). Error bars reported in Figure 10b correspond to the standard deviation of the plagioclase dataset, whose range is larger than error of the method. We stress that the reported temperatures are obtained using Helz dry model. Further uncertainty arises from the dependence of the method on dissolved water content as shown recently by Putirka (2008). In order to minimize the number of assumptions and perform a comparison between distinct eruptions, we preferred to adopt the dry model.







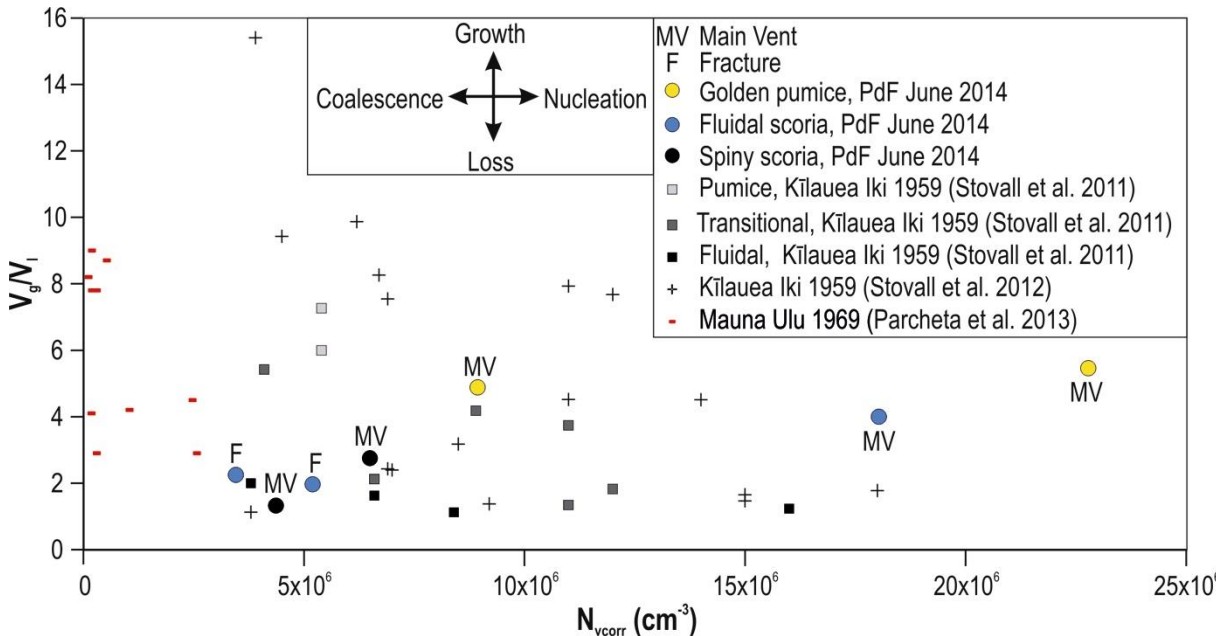


**Figure 11** Volumetric ratio of vesicles to melt ($V_G/V_L$) versus vesicle number density

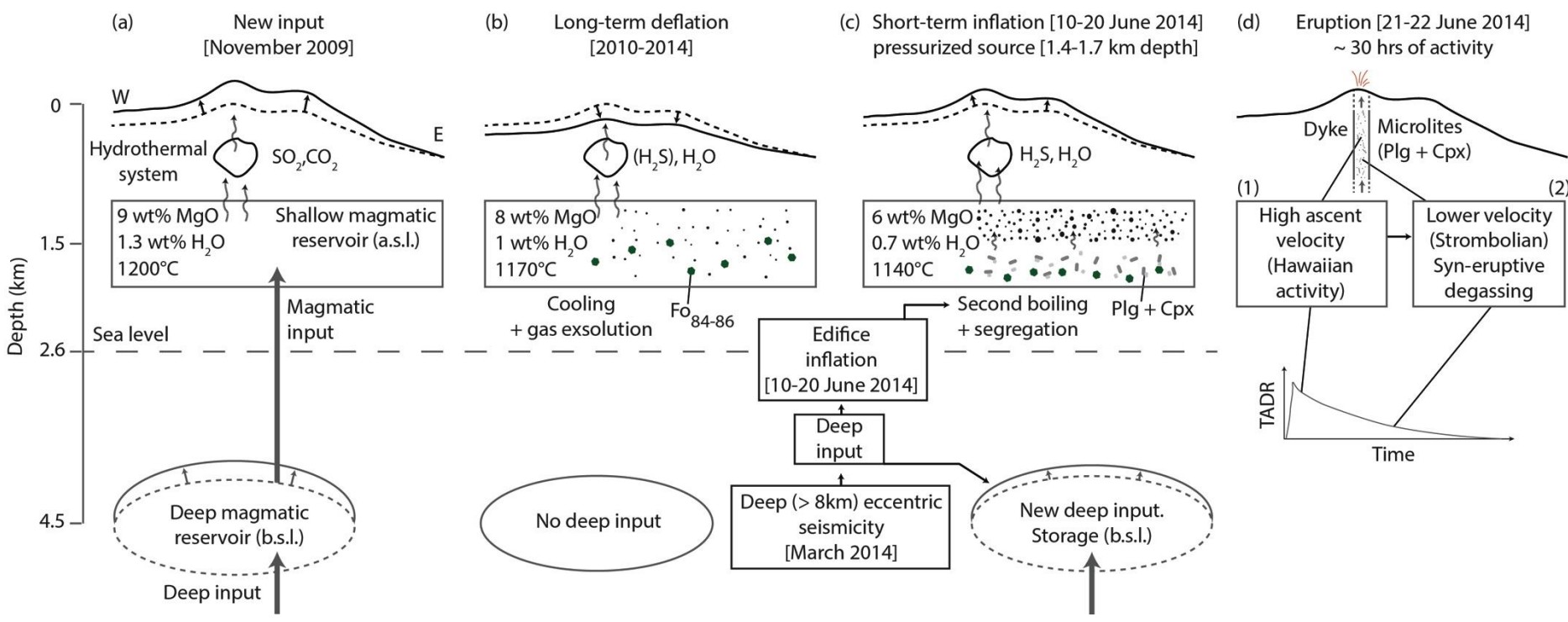


**Figure 12** Schematic model of the evolution of PdF volcanic system from the new deep magmatic input of November 2009 up to June 2014
eruption. See explanation in the text