# Peer review of "Supporting Information for"

_Solid Earth, 2017_

## Referee Comment (RC1) · A. Lindoo (Referee) · 16 Oct 2017

**Summary:**

This paper highlights the importance of combining textural, chemical and petrological data to describe volcanic eruptions. Employing this multidisciplinary approach, the authors provide a narrative for transition in eruption activity during the 2014 eruption at Piton de La Fournaise, la Reunion. The bulk of interpretations made by the authors come from four pyroclast typologies that were produced over the course of the eruption from the Main Vent and Western Fracture. Melt inclusion data indicates a single, heterogeneous magma source. The clasts record a cooler, relatively crystal-rich, de-

gassed, portion of magma along with a hotter, volatile-rich portion. Gas accumulation in the shallow reservoir, released from the cooler, crystal-rich magma, contributed to the fast ascent rate or hotter magma. This resulted in the hawaiian fire-fountain activity. The authors propose successive tapping of the reservoir caused the decrease in explosive intensity. The conclusions are reasonable given the observational and analytical data. This work provides a wealth of data that will no doubt be useful for future study and comparison. However, I think the paper could better exemplify the multi-disciplinary approach and strengthen its conclusions by incorporating more equal weight/description to each component. I recommend this paper to be accepted after minor/moderate revision which I have detailed in the following text.

**General Comments:**

I was intrigued when I saw permeability data would also be included, (pg.7 lines 208-210, "more than 200 clasts of similar size (maximum diameter between 16 and 32 mm, see Gurioli et al. 2015) were collected, both close to the Main Vent and in the 'distal' area (30 metres away from the vent) for density, connectivity, permeability, petrological and geochemical analysis") however, the paper does not present any permeability data. In Supplementary table S1, I see that it was performed on some samples. It would be interesting to see how that data adds to the interpretation of eruptive activity. Also, I think it would also be a more intuitive measurement than isolated/connected vesicularity.

I would like to see a more detailed discussion of the crystallinity data given the large impact of crystals on bubble deformation, connectivity/permeability (Spina et al. 2016, Lindoo et al. 2017), volatile distribution in the conduit (Parmigiani et al. 2011, Parmigiani et al. 2016), and ultimately eruptive style.

**Does the paper address relevant scientific questions within the scope of SE?**

The authors present observational and analytical data that contribute to furthering our understanding of transitions in eruption dynamics. The manuscript includes information of interest to a range of sub-disciplines including seismology, physical volcanology, geochemistry, and igneous petrology.

**Does the paper present novel concepts, ideas, tools, or data?**

The paper presents conventional methods of describing eruption products. The multidisciplinary approach in combing bulk textural and geochemical data into one paper sets a standard for future papers in describing volcanic deposits. However, some of the components could use more detail.

**Are substantial conclusions reached?**

The authors address most of the research questions posed in the introduction, which are best summarized in the conclusion section. I did not come away with a clear understanding of their first (i: why was such a small volume of magma erupted instead of forming an intrusion) or fifth (v: What was the time and space evolution of the eruptive event) objectives.

**Are the scientific methods and assumptions valid and clearly outlined?**

The methods employed in the study are generally well described, with the exception of a few points:

The authors employ circularity to characterize different clast types. First, how many particles of each typeology were measured using the Morphologi G3? I would also suggest the use of at least three shape descriptors, as recommended by Liu et al. 2015, to fully describe particle morphologies. Currently, the use of a shape descriptor in the interpretation of the eruptive products comes across as an afterthought. Because circularity is not really utilized in the description/interpretation of the products,

the section could just be removed. I do think it would be interesting to see if other shape descriptors (such as solidity and convexity) may better describe the relationships between particle shapes and eruption styles.

Why were only 25 clasts from the Western Fracture analyzed versus 146 from the Main Vent? I'm speculative whether the number of clasts accurately samples the Western Fracture explosion. Also, I do not find it clear how clasts were picked for analyzing vesicle size distributions. I find the Spiny Glass and Golden pumice density distributions to be slightly bimodal (Fig. 6c). Do the stars in Fig. 6 denote the mode determined for each component? This should be noted in the figure caption as well.

I do not see a table that includes all of the crystallinity data (vol.Crystallinity data could be inserted into Table 3 in the connected vesicle or isolated vesicle column, as it's not necessary to have both (connected/isolated) listed. There is some description in the results (phases present), but I find it difficult to follow without a table to reference/compare. I would also be interested to see the phase abundances and aspect ratios The amount of crystals (specifically high aspect ratio plagioclase) coupled with the vesicularity data, may give more insight into efficient vs. inefficient degassing in the different typeologies (see Shea et al. 2017). The amount of crystals (depending on the aspect ratio) will influence degassing as well (Lindoo et al. 2017).

**Are the results sufficient to support the interpretations and conclusions?**

The results support the interpretations and conclusions made by the authors. I would ask the authors to also consider the effect of crystals on the permeability of the "degassed, cooler reservoir" along with their interpretation of reservoir tapping. Crystals increasing bubble connectivity/permeability of the reservoir alone may contribute to extensive degassing and shifts in eruptive style.

**Is the description of experiments and calculations sufficiently complete and pre-**

**cise to allow their reproduction by fellow scientists (traceability of results)?**

Yes

**Do the authors give proper credit to related work and clearly indicate their own new/original contribution?**

Yes

**Does the title clearly reflect the contents of the paper?**

Yes

**Does the abstract provide a concise and complete summary?**

Yes

**Is the overall presentation well structured and clear?**

I find the manuscript to be fairly well structured. Section 5.2 might benefit from sub-sections or reorganization, perhaps divided by the different typologies, sampling area, or interpretation and comparison to other studies. There is a lot of information presented and comparison to other studies.

**Is the language fluent and precise?**

Yes

**Are mathematical formulae, symbols, abbreviations, and units correctly defined and used?**

Yes

**Should any parts of the paper (text, formulae, figures, tables) be clarified, reduced, combined, or eliminated?**

Lines 99-105 – Reassess/reorganize the questions posed here. There are 5 questions listed with (iv) and (v) attached to (iii). I suggest separating each question with a paragraph or do not separate them. Also, I do not think questions (i) or (v) were addressed in the discussion/conclusions section.

Table 3 does not need both connected vesicularity and isolated vesicularity listed.

Figure 5c needs a more descriptive caption. I'm not sure what I and II refer to or the arrows (the clasts pictured?). I think the caption only describes one of the two graphs?

Figure 6c – please clarify the meaning of the star symbols

Figure 11 could be redrafted to provide more clarity to the reader. I would move the references to the figure caption to make room for an inset similar to Stovall et al. 2011 to help the reader interpret trends.

**Are the number and quality of references appropriate?**
 89 references - I think the number of the references could be reduced.

**Is the amount and quality of supplementary material appropriate?**
Yes. Some formatting issues with supplementary tables.
**Minor corrections:**

Line 111 – I would recommend removing this final sentence. The authors make it clear earlier in the introduction the importance of the multi-disciplinary approach.

Line 218 – Should reference Fig. 3e not 3f?

Line 310 – Combine the two sentences with the rest of the paragraph.

Line 331 – Should reference Figure 3b?

Line 510 – subscript "wr" in MgOwr.

Line 645 – reference numbers for comparison to Houghton et al. 2016.

General - Vg/Vl should be Vg/Vl. Subscript "v" in Nv.

Figure 1 – An inset map of Reunion Island would be helpful. (1c) is very dark/difficult to see.

Figure 3c – The pictures are so small it is difficult to see.

Figure 3e – 2010, Fountaining is spelled wrong.

Figure 10 – Inconsistent figure formatting. Thick axes lines and bold axes values

**Missing or incorrect references:**

Bombrun et al. 2015 (line 703)

Di Muro et al. 2012 (line 126)

Gurioli et al. 2008 (line 633)

Hammer et al. 1999 (line 750)

Inman 1952 (line 223)

Liuzzo et al. 2015 (line 134)

Morandi 2015 (line 72)

Line 58 – Taddeucci misspelled

Line 60 – Extra "and"

Line 60 – Eychenne misspelled.

Line 61 – Should read "Leibrandt and Le Pennec, 2015".

Line 600 – references in italics.

Line 971 – Should read "Lange, R.A.,. . ."

Line 1016/1020 – reference chronology inconsistent.

Line 1023 – delete "a" from reference.

**References cited:**

Lindoo, A., Larsen, J. F., Cashman, K. V., and Oppenheimer, J., 2017, Crystal controls on permeability development and degassing in basaltic andesite magma: Geology, 45(9), p. 831-834.

Liu, E. J., Cashman, K. V., Rust, A. C., 2015, Optimising shape analysis to quantify volcanic ash morphology: GeoResJ, 8, p. 14-30.

Parmigiani, A., Huber, C., Bachmann, O., and Chopard, B., 2011, Pore-scale mass and reactant transport in multiphase porous media flows: Journal of Fluid Mechanics, v. 686, p. 40-76.

Parmigiani, A., Faroughi, S., Huber, C., Bachmann, O., Su, Y, 2016, Bubble accumulation and its role in the evolution of magma reservoirs in the upper crust: Nature, 532,

p. 492-494.

Spina, L., Cimarelli, C., Scheu, B., Di Genova, D., and Dingwell, D. B., 2016, On the slow decompressive response of volatile- and crystal-bearing magmas: An analogue experimental investigation: Earth and Planetary Science Letters, v. 433, p. 44-53.
* * *

---

## Referee Comment (RC2) · M. Myers (Referee) · 21 Oct 2017

Overview

The authors present a detailed study of the first, rather minor eruption of Piton de la Fournaise that occurred on June 20-21st 2014 after a 5-year period of dormancy. The ultimate goal of the work is to reconstruct the processes responsible for the eruption of these less-voluminous events. They proceed to answer this question through combining textural information for the diverse erupted products with petrologic and geochemical data. The approach is well thought out and the dataset large enough to allow for decoupling of magmatic, ascent, and surficial processes. Although the manuscript

is well written and the methods and results are quite detailed, the connections between sample locations, type of products collected, and ultimate textural results could be presented in a clearer fashion, which would only serve to strengthen the results and importance of the study. After addressing and clarifying the comments presented below, I recommend that this paper be accepted for publication.

Scientific Comments

L 35-37: This comment also concerns the end of the discussion. Although second boiling is a plausible triggering mechanism, I have two issues with this statement. First, the idea of second boiling, i.e. water exsolution, is directly the result of cooling and chemical evolution of a separate body, and cannot be decoupled. And second, there is some evidence for a mafic recharge event months before the June 20th eruption. Although I agree that there is no evidence for heat or chemical recharge in the erupted products from this minor eruption, ultimately I feel as if a potential recharge event two months before ending 5 years of dormancy is an important observation and should be at least comment on.

L108: What are the typical heights for Strombolian activity?

L133: This deeper seismicity and increase in soil CO2 seems to suggestion that some sort of magma movement/recharge is associated with the beginning of activity. Although decoupled in terms of months from the eruption on June 20th, a comment on how this fits into the plumbing system and inner working for PdF would make a nice addition for the reader.

L152-156: The inflation and deformation trends mentioned would be great to see as a figure (supplemental?), for integration of the information provided here, with the larger story of the PdF system.

Section 2.2: The detail of the samples collected is excellent, however it was challenging as a reader to understand how many samples were collected at each location, and

then how many of these samples were then focused on in the methodology. Perhaps a general sentence on this could help to transition the reader.

L214: Two bulk samples from the Main Vent. Does that mean the base and the top?

L245: How many sample sites were there? From the Figures it seems as only three samples are being presented: the top and bottom of the Main Vent, and then a sample from the Western Fracture.

L370: How many deposits from the Fractures were studied? It seems that the figures only have the Western Fracture; does that include multiple samples?

L411-415: The first line states that the fluidal and golden clasts have a larger amount of isolate vesicles, but then on 413 it states that these two types with high vesicularity are characterized by fewer amounts of isolate vesicles? Fewer, but still the largest amount compared to the other clast types? Some clarification required.

L422: How much of the lower Ni and Cr concentrations whole rock geochemistry could just be due to crystal content?

L524/L549: Some of the data (MIs and Plagioclase, specifically) point to having a bi-modal population. However, this point doesn't seem to come back up in the discussion.

L553: How detailed (in terms of spacing) were these transects compared to the Di Muro et al. dataset? Were BSE images taken? Seems hard to believe that both the 2008 and 2014 have bimodal plag populations, and that the 2014 eruption is a more evolved upper portion of the system, but doesn't contain complex zonation in the plag? I am not trying to discredit the observation if it is valid, but rather more information could help to support this statement.

L559: This is really shallow. How were the H2O/CO2 concentrations measured in Di Muro et al. 2016, and in what phase (plagioclase or olivine?)?

L575-581: Are these temperature +/- associated with the error in the thermometer, or

the standard deviation of the plagioclase dataset? Although it does appear to show a decrease in temperature, I wouldn't refer to this range (50 C) as large variability in temperature, especially considering I believe this thermometer has an error bar that will help to overlap the dataset.

L600: What would you expect to see as a geochemical signature of hot gases streaming past ejecta? Do people see evidence for this as a geochemical signature in other systems?

L611-612: Very neat observation!

Section 5.2: A strength to this section is starting with background information on the textural information observed in other systems.

L648-650: I think this is a key point for the community to come out of your paper that should be highlighted more in the conclusions.

L691-696: The information presented here may be more useful earlier in this section so the reader has it for guidance when reading through the results of this study. Just a suggestion.

L711-712: This manuscript has a rich amount of information. One of the weaknesses at the end, however, is the challenge of visualizing how the textural information fits into the eruption/sampling information. Perhaps a schematic depicting the statement that eruption style and thus eruptive products, vary along the length of the fracture system would help guide the reader and bring everything together.

L764: In this presentation, the cooler, crystallizing magma is below the shallow chamber that is being replenished with volatiles? Is this a stable configuration?

L772: This reference to Menand and Phillips seems random. Observed experimentally how?

L772-773: The golden and fluidal fragments vs. spiny fragment lines are a repeat of

Lines 762-765.

L790-792: I don't understand how to call on cooling, crystallization and water release as a pressurization mechanism, and then state that magma cooling and evolution is not helping to pressurize the source. I think from the MI sentence before I understand that the idea is there is no evidence for evolution controlling what types of products are erupted out, but I don't see how that can translate into the lack of evidence for cooling and evolution driving pressurization.

Figure (Caption) Comments

Figure 1: In many ways this figure is the most important, as it frames where the samples used in this study were taken. However, it is challenging to read and not fully explained. Including: (A). I can't tell the difference between red in orange at this scale. What are the dates? Eruptions or samples collected? Also the units for lat/long should be described. (C). Adding the sample locations to the blow up of C would be useful. Also C needs to be lighter as it is hard to read. Where were the gases collected that are listed as sampled in C? And, were they commented on in this study?

Figure 2: Photo collection is not just from 'the website', but rather several sources. Although I appreciate that the sources are provided, it would be nice to explain what the photo depicts, and why that is important for the study. How do these pictures fit into sample locations/clasts described?

Figure 3: It appears the thermal scale bars for the two images in a) are different. Are they still comparable? Why is the diameter scale different for the Western Fracture, shown in d), compared to a) and e)? Main vent should be capitalized to Main Vent.

Figure 4: I really like this figure. I found myself wondering the distribution of these 4 types. It might be nice to direct the reader to Figure 5 for that information.

Figure 5: Main Cone should be Main Vent for consistency. One thing I found confusing in this paper was keeping track of the different sampling locations and what was being

compared. Does this figure show data for the base and top (not through stratigraphy) from one sample location? If so, it might be nice to clearly state this.

Figure 6: Shouldn't a) and c) be the same if they are both for the Main Vent, where c) is broken down by clast type? What do the stars in c) represent? The diagonal lines in d) look the same, although the caption just refers to a single line. Perhaps explain what the % refer to (I assume the % vesicularity accommodated by isolated vesicles?)

Figure 10: Need to specify if the glasses are melt inclusions or matrix.

Technical Corrections

L119: The last previous sounds awkward. Perhaps just 'The last'?

L327: 'smooth fluidal (Figs. 3d) bombs and lapilli'. Refers to multiple figures, and also reads oddly. Are the bombs and lapilli fludial?

L225, L445, L451: Lines where paragraph indents are needed

L690: Need another parentheses at the end.

---

## Referee Comment (RC3) · Anonymous Referee #3 · 25 Oct 2017

Review of "Integrating field, textural and geochemical monitoring to track eruption triggers and dynamics: a case-study from Piton de la Fournaise" by Gurioli et al., submitted to Solid Earth Discussions

General comments:

This manuscript presents new data on the textural and geochemical analyses of pyroclasts from the 2014 eruption at Piton de la Fournaise (PdF), la Reunion. Using various sampling and measuring techniques, the authors determined a chronology of the eruption events, morphology, grain size and microtexture of pyroclasts, and petrology and geochemistry of the bulk rock, glass, crystal and the melt inclusions. Based on small precursory activity and the analyses in this study, the authors suggest that the eruption was triggered by pressurization due to bubble accumulation in a shallow magma reservoir, as opposed to magma chamber cooling or a new batch of magma flux into the reservoir

This study includes a very thorough physicochemical analysis of pyroclasts from PdF, which is worth publishing after revising the comments mentioned here, which are mainly related to the discussion or implication sections of the manuscript. In general, the outcomes of this study are not transparent with regards to the questions addressed in Lines 99-105. It seems that the paper includes a number of hypotheses while the validity of those are inadequately presented. I suggest either rephrasing parts of the manuscript as applicable or provide some quantitative analysis in support of some of the conclusions. Also, I find a number of parameters in the figures are not defined properly in the text or in figure captions, making it difficult to follow at places. I hope the authors will find the following specific comments useful for further improvements.

Specific comments:

Lines 801-807: Following my general remark, several possible scenarios have been proposed here without a reasonable justification. For example, "we found that this kind of eruption can be triggered solely by bubble accumulation and source pressurization" – The relationship of bubbles, pressure build-up and its extent for eruption triggering have not been demonstrated in this study.

Lines 798-799: It seems like the hypotheses of a shallow magma reservoir and its pressurization are mostly driven by the weak and short geophysical precursors, which is not the focus of this study. In other words, the contribution of geochemical/petrological monitoring independent of geophysical signals – for tracking eruption triggers and dynamics are not transparent.

Title: The title is too broad. Although it is catchy, but based on the previous two comments, neither the trigger nor the dynamics are adequately addressed in this study.

Lines 636-640 and 683-689: Isolated vesicles, also mentioned in some other parts of the manuscript, could simply be a result of post-coalescence surface tension forces, especially for low viscosity magmas due to relatively smaller viscous forces. Therefore it may not represent the low rate of deformation, and can even get overprinted during cooling of the pyroclasts. On the other hand, the presence of micro-crystals increase viscosity preserving the coalesced textures (see Moitra et al. 2013, Relating vesicle shapes in pyroclasts to eruption styles, Bull Volc, for a discussion), and therefore if syneruptive, it may not represent cooled magma and longer residence times. Therefore the implications/conclusions need to be more convincing, or a discussion on the various possibilities is required, also insightful, at the least.

Figure 5c: There is no discussion on circularity? What about any other shape factor? What do they mean?

Figure 6d: There are a number of solid lines drawn without a proper caption. Which diagonal line (and therefore the samples) represents equality and what are those various percentages?

Technical corrections:

Line 75: space between grain and size
Line 81: weird spacing
Line 189: $Mm^3$ could be defined in line 188, where million $m^3$ is first introduced, for better clarity.
Figure 1c caption: locations instead of location
Figure 4 caption: %cry and not %Cry to be consistent
Figure 9 – 'T' in FeOT should be in subscript
- The name/expression "Piton de la Fournaise" is not consistent in the manuscript: 'La' is often used instead of 'la'
- Figure subplots are sometimes labeled by capital letters, sometimes by small letters

---

## Editor Comment (EC1) · M. J. Heap (Editor) · 27 Oct 2017

Dear Lucia Gurioli,

As you have no doubt seen, I have now received three reviews of your manuscript "Integrating field, textural and geochemical monitoring to track eruption triggers and dynamics: a case-study from Piton de la Fournaise". All three reviewers were positive, but all recommended that the paper is slightly reconstructed/rejigged to improve clarity and therefore the impact of your manuscript. This is always a challenge for data-rich contributions. Please pay particular attention to these comments when preparing your revised manuscript.

[Figure]

Please now prepare a revised manuscript and point-by-point rebuttal letter. Addressing the comments of the reviewers will no doubt help improve your manuscript. I look forward to reading your revised manuscript.

Thanks,

Mike Heap (Topical Editor of Solid Earth)

---

## Author Comment (AC1) · 28 Nov 2017

Dear Madison,

Thanks a lot for your comments and corrections. Please find here our detailed list of responses and the manuscript attached with all the corrections and the new figures; A few explanations are reported on Amanda responses.

"The connections between sample locations, type of products collected, and ultimate textural results could be presented in a clearer fashion, which would only serve to strengthen the results and importance of the study. "

[Figure]

We made it clearer moving the sample strategy in the methodology and adding the corresponding samples to the sample sites and improving the figures.

"L 35-37: This comment also concerns the end of the discussion. Although second boiling is a plausible triggering mechanism, I have two issues with this statement. First, the idea of second boiling, i.e. water exsolution, is directly the result of cooling and chemical evolution of a separate body, and cannot be decoupled. And second, there is some evidence for a mafic recharge event months before the June 20th eruption. Although I agree that there is no evidence for heat or chemical recharge in the erupted products from this minor eruption, ultimately I feel as if a potential recharge event two months before ending 5 years of dormancy is an important observation and should be at least comment on."

In the discussion we clarified that deep magma transfer (mantle level depth) has been identified by Boudoire et al., 2017 (GRL) months before the June 2014 eruption. We speculate that deep magma transfer can have modified the stress field at crustal level and promoted/facilitated volatile exsolution in the shallow reservoir. Vertical magma transfer at crustal level has been identified only in 2015 by Peltier et al., 2016 and resulted in progressive change of magma composition (Coppola et al., 2017).

"L108: What are the typical heights for Strombolian activity?"

Average height of PdF fountains is 20 meters (we added in the text); larger fountains (tens-hundreds of meters only occur during large and intense eruptions, like 2007. Strombolian are usually less than 10 meters high

"L133: This deeper seismicity and increase in soil CO2 seems to suggestion that some sort of magma movement/recharge is associated with the beginning of activity. Although decoupled in terms of months from the eruption on June 20th, a comment on how this fits into the plumbing system and inner working for PdF would make a nice addition for the reader."

Please see previous remarks/answers on this point

"L152-156: The inflation and deformation trends mentioned would be great to see as a figure (supplemental?), for integration of the information provided here, with the larger story of the PdF system."

We added this information in Figures 1 and 12

"Section 2.2: The detail of the samples collected is excellent, however it was challenging as a reader to understand how many samples were collected at each location, and then how many of these samples were then focused on in the methodology. Perhaps a general sentence on this could help to transition the reader."

We moved the sampling strategy and specify the samples

"L214: Two bulk samples from the Main Vent. Does that mean the base and the top?"

Yes, we explained it better (see also new Figure 3)

"L245: How many sample sites were there? From the Figures it seems as only three samples are being presented: the top and bottom of the Main Vent, and then a sample from the Western Fracture."

Yes, we explain that the sample sites for the texture were three: Western Fracture, Upper Fracture and Main Vent and we specify the number of samples and clasts for each site ( from line 208)

"L370: How many deposits from the Fractures were studied? It seems that the figures only have the Western Fracture; does that include multiple samples?"

We studied one deposit from the Western Fracture (for a total of 25 scoriae) and only one big bomb at Upper Fracture that broke in five fragments (see 208). Actually, we stressed in the text the description of this bomb, because we could measure the core and the quenched rind and find interesting results, see new Figure 6 + caption + (from line 435)

"L411-415: The first line states that the fluidal and golden clasts have a larger amount of isolate vesicles, but then on 413 it states that these two types with high vesicularity are characterized by fewer amounts of isolate vesicles? Fewer, but still the largest amount compared to the other clast types? Some clarification required."

We rewrote it (from line 471).

"L422: How much of the lower Ni and Cr concentrations whole rock geochemistry could just be due to crystal content?"

Careful sample selection has permitted to obtain a set of virtually olivine-cpx free crystals. Any addition of mafic crystals translates into an enrichment in Ni-Cr; those samples that contain a few % of crystals, (consistent with textural and petrological observation) are slightly enriched in compatible elements. We added this explanation in caption of Figure 7

"L524/L549: Some of the data (MIs and Plagioclase, specifically) point to having a bimodal population. However, this point doesn't seem to come back up in the discussion."

Bimodal MI composition has been used as i) further evidence (beside geochemical modeling) to link the November 2009 and June 2014 magmas. Discussion to constrain the duration of cooling 2009-2014 vs the timing of foaming (11 days before the eruption as constrained by inflation) and ii) to support processes of crystal recycling. Recall here that i) bimodal composition of plagioclase is common at PdF and ii) it tracks two environments: calcic plagioclase formed in depth during cooling (before degassing) and sodic plagioclase formed during magma ascent and degassing in the dyke before magma fragmentation and extrusion (see new Figure 10a).

"L553: How detailed (in terms of spacing) were these transects compared to the DiMuro et al. dataset? Were BSE images taken? Seems hard to believe that both the 2008 and 2014 have bimodal plag populations, and that the 2014 eruption is a more evolved upper portion of the system, but doesn't contain complex zonation in the plag? I am not trying to discredit the observation if it is valid, but rather more information could help to support this statement."

The 2008 eruptive products contained plagioclase with complex zoning and unusual composition. Their intermediate composition, in fact, filled the gap typically observed between calcic and sodic composition usually observed in many PdF eruption. The composition of 2014 plagioclase is bimodal and does not show the occurrence of intermediate compositions (Fig. 10a). Plagioclase analyses were performed on spots representative of core, mantle and rim portions of the crystals.

"L559: This is really shallow. How were the H2O/CO2 concentrations measured in Di Muro et al. 2016, and in what phase (plagioclase or olivine?)? Di Muro et al., 2016 performed a review of all analyses on melt inclusions performed at PdF. Most analyses of volatiles were obtained on melt inclusions host in olivines and pyroxene. The shallow pressure has been confirmed by the study of several PdF eruptions and is attributed to shallow magma emplacement (consistently with geophysical data; see Di Muro et al., 2014 for a review). A few melt inclusions have been also identified recording late stage water and CO2 leakage and diffusion. This last process, however, does not modify significantly the average shallow saturation pressure recorded by most melt inclusions at PdF."

Besides that, it is important to recall that the vast majority of volcano-tectonic earthquakes recorded at PdF are located in the uppermost 2 km of the volcano edifice, at shallow depth below the summit caldera.

"L575-581: Are these temperatures +/- associated with the error in the thermometer, or the standard deviation of the plagioclase dataset? Although it does appear to show a decrease in temperature, I wouldn't refer to this range (50 C) as large variability in temperature, especially considering I believe this thermometer has an error bar that will help to overlap the dataset."

Error bars reported in Figure 10b correspond to the standard deviation of the plagioclase dataset, whose range is larger than error of the method. We stress that reported temperatures are obtained using Helz dry model; further uncertainty arises from the dependence of the method on dissolved water content as shown recently by Putirka; in order to minimize the number of assumptions and perform a comparison between distinct eruptions, we preferred to adopt the dry model. We added this explanation in caption of figure 10b

"L600: What would you expect to see as a geochemical signature of hot gases streaming past ejecta? Do people see evidence for this as a geochemical signature in other systems?"

Vlastélic et al. have documented the mobility of alkalis and other elements on PdF clasts having experiences long exposures to acid gases. This is a well-known process potentially affecting samples with a high glass contents (e.g. Pele's hairs, golden pumices etc). Our aim was to show that our samples, collected rapidly after eruption, do not show any evidence of post-emplacement modification by acid attack.see explanation added at line 671.

"L611-612: Very neat observation!"

Thanks. I stressed this point especially for past basaltic deposits, where we need to be careful when we interpret them.

"Section 5.2: A strength to this section is starting with background information on the textural information observed in other systems."

Amanda asked to reorganize this section and in part we did it, but we agree with Madison to leave the background first

"L648-650: I think this is a key point for the community to come out of your paper that should be highlighted more in the conclusions."

Thanks Madison, we agree with you and we will stress this point, but we also have to convince review 3 that we are right; according to him/her everything happen after the explosion

"L691-696: The information presented here may be more useful earlier in this section so the reader has it for guidance when reading through the results of this study. Just a suggestion."

Yes, we moved it up

"L711-712: This manuscript has a rich amount of information. One of the weaknesses at the end, however, is the challenge of visualizing how the textural information fits into the eruption/sampling information. Perhaps a schematic depicting the statement that eruption style and thus eruptive products, vary along the length of the fracture system would help guide the reader and bring everything together."

We added a new Figure (Fig. 12) to show the eruptive style variation in time and link it with the reservoir-dyke system and deep system

"L764: In this presentation, the cooler, crystallizing magma is below the shallow chamber that is being replenished with volatiles? Is this a stable configuration?"

We explained the configuration earlier with a zoned shallow reservoir and we added Figure 12

"L772: This reference to Menand and Phillips seems random. Observed experimentally how?"

We just cited them and we deleted the experimental side, that doesn't concern the paper

"L772-773: The golden and fluidal fragments vs. spiny fragment lines are a repeat of Lines 762-765."

Removed

"L790-792: I don't understand how to call on cooling, crystallization and water release as a pressurization mechanism, and then state that magma cooling and evolution is not helping to pressurize the source. I think from the MI sentence before I understand that the idea is there is no evidence for evolution controlling what types of products are erupted out, but I don't see how that can translate into the lack of evidence for cooling and evolution driving pressurization."

See new interpretation and Figure 12

"Figure 1: In many ways this figure is the most important, as it frames where the samples used in this study were taken. However, it is challenging to read and not fully explained. Including: (A). I can't tell the difference between red in orange at this scale. "

We enlarged the figure

"What are the dates? " The dates when the fractures were active. We added in the caption of Figure 1.

"Eruptions or samples collected? " Eruptions

"Also the units for lat/long should be described". Added

"(C). Adding the sample locations to the blow up of C would be useful." We enlarged the Figure

"Also C needs to be lighter as it is hard to read. " Done

"Where were the gases collected that are listed as sampled in C? And, were they commented on in this study?" We just mention them in the sample strategy (see line 221) but we also state that we do not discuss these data in the paper

"Figure 2: Photo collection is not just from 'the website', but rather several sources. Corrected

"Although I appreciate that the sources are provided, it would be nice to explain what the photo depicts, and why that is important for the study. " We added more explanations in the captions, in the photos and also we added more useful photos.

"How do these pictures fit into sample locations/clasts described? " We added all the geographical symbols to locate the area

"Figure 3: It appears the thermal scale bars for the two images in a) are different. Are they still comparable? The setting range used for the acquisition of the data was the same; the occurrence of slightly different maxima in the two fields of view results in distinct scale bars; however, the two figures can still be combined to qualitatively illustrate the sampling field soon after the eruption. The temperature of the deposit were instead measured using a thermocouple." We removed the thermal photo and we added a photo of the deposit

"Why is the diameter scale different for the Western Fracture, shown in d), compared to a) and e)?" Fig 3b is in half phi, while in c and d the diameter is in full phi, we added in the caption.

"Main vent should be capitalized to Main Vent." Done

"Figure 4: I really like this figure. I found myself wondering the distribution of these 4 types. It might be nice to direct the reader to Figure 5 for that information." We added it in the caption and we added the lava as well and the crystals properly

"Figure 5: Main Cone should be Main Vent for consistency." Corrected

"One thing I found confusing in this paper was keeping track of the different sampling locations and what was being compared. " We added explanation in the methodology

"Does this figure show data for the base and top (not through stratigraphy) from one sample location? If so, it might be nice to clearly state this." We added explanation in the figure caption

"Figure 6: Shouldn't a) and c) be the same if they are both for the Main Vent, where c) is broken down by clast type? " Yes, thanks a lot, we re-did the graphs, with the right normalization

"What do the stars in c) represent? " They represent the picked samples for the texture measurements. We added in the caption and we adjusted the histograms

"The diagonal lines in d) look the same, although the caption just refers to a single line. Perhaps explain what the % refer to (I assume the % vesicularity accommodated by isolated vesicles?) " yes, we added the explanation

"Figure 10: Need to specify if the glasses are melt inclusions or matrix." The data have been obtained by studying the glass-plagioclase equilibrium or on the basis of matrix glass analyses; we added this information in the caption

Technical Corrections

"L119: The last previous sounds awkward. Perhaps just 'The last'?" Done

"L327: 'smooth fluidal (Figs. 3d) bombs and lapilli'. Refers to multiple figures, and also reads oddly. Are the bombs and lapilli fludial? " yes

"L225, L445, L451: Lines where paragraph indents are needed" Added indents

"L690: Need another parentheses at the end". Added correction

Please also note the supplement to this comment:
https://www.solid-earth-discuss.net/se-2017-99/se-2017-99-AC1-supplement.pdf

[Figure]

[Figure]

**Fig. 1.**

**Supplement:**

[revised manuscript text omitted]
 (4.6 years) and was preceded by only a few days of weak and short geophysical seismic, geodetic and geochemical precursors (Peltier et al., 2016 and Fig. 1d). This multidisciplinary approach provides new constraints on the mechanisms triggering such short-lived, small but potentially hazardous eruptions (STIAMO RIPETENDO TRE VOLTE GLI STESSI CONCETTI. RIASSUMI E TAGLIA).supposewas sittingcollingDuring the inflation period (b,itself allows exsolution, and magma with crystallization in situ of thelites, as observed in the lava sampleFig.This process favours a physical zonation of the shallow reservoir. Therefore, magma storage at shallow depth favours volatile (mostly H₂O) exsolution at several steps during magma ponding, cooling and evolution (Fig. 12b)~~

The occurrence of deep (>10 km bsl) lateral magma transfer since March-April 2014 has been inferred by Boudoire et al., (2017) on the basis of deep (mantle level) seismic swarms and increase in soil CO2 emissions on the distal western volcano flank. The incipit of magma transfer towards shallower crustal levels is potentially recorded by subtle volcano inflation about 11 days before the June 2014 eruptions (Figs. 1d and 12c). We suspect that these deep processes can have progressively modified the shallow crustal stress field and favoured magma vesiculation and melt-crystal separation. Second boiling could thus have over-pressured the shallow seated reservoir and triggered magma ascent  (Fig. 12c).

Without this deep magma transfers  we believe that the small reservoir activated in 2014 would have cooled down completely to form  an intrusion (as suggested by the pervasive crystallization of the lava, one of the densest emitted from 2014 to 2017).  The 2014 event represented instead the first of a long series of eruptions, whose magmas became progressively less evolved in time (Coppola et al., 2017). In this scenario the trigger mechanisms of 2014 activity are both internal and external in the sense that the small shallow reservoir hosting cooled magmas permitted to create the conditions favourable to a second boiling (Fig. 12c, and Tait et al., 1989). The second boiling was likely trigger by an almost undetectable stress field change, and was favoured by the shallow storage pressure of the magma (Fig. 12c) that promotes fast water exsolution and rapid magma response to external triggers. The second boiling possibly contributed to  the inflation registered 11 days before the eruption at 1.4-1.7 km (Fig. 12c) both by magma expansion and  transfer of hot fluids to the hydrothermal system (Lénat et al., 2011).

Our data permit to exclude (i) new magma input and/or to fluid inputs (CO2-rich fluids) from deep magmatic levels to trigger the June 2014 eruption. We also exclude (ii) heating and enhanced convection of the shallow magma reservoir (due to heat diffusion without fluid or mass transfer), because this process is very slow . Furthermore,   2014 minerals do not record evidences of  magma heating. We can exclude equally  (iii) deformation of the volcanic edifice and decompression of the magma reservoir and/or hydrothermal system due to flank sliding because geodetic data show no evidence of flank sliding able to produce stress change in  the hydrothermal and magmatic system. ~~Geochemical (bulk rock) and petrological (mineral composition and zoning) data, permit to exclude this hypothesis. The magma erupted in 2014 results to be one of the most evolved and cold magmas ever erupted at Piton de la Fournaise (Figs 8 and 10b); it is very homogeneous (Fig. 7), minerals do not exhibit reverse zoning and their compositional evolution from phenocrysts to microlites record magma cooling and final degassing (new Figure 10a).cannot bedetected by the OVPF geodetical~~ detect for any monitoring network.

We conclude that the overpressure, caused by the second boiling, triggered the ~~The 11 days of weak summit volcano inflation, which preceded the 2014 eruption, possibly result from volatile exsolution and expansion of both the shallow magma reservoir and the hydrothermal system (Fig. 12c). We also exclude (ii) heating and enhanced convection of the shallow magma reservoir (due to energy transfer without fluid or mass transfer, because rocess is very slow because of slow heat diffusion and 2014 minerals do not record evidences of slow magma heating. We can exclude equally (iii) deformation of the volcanic edifice and decompression of the magma reservoir and/or hydrothermal system due to flank sliding because geodetic data show no evidence of flank sliding able to produce decompression of the hydrothermal and magmatic system. However, it is necessary to discuss the pressurisation (volcano inflation) and/or depressurization (volcano deflation) of the (iv) hydrothermal system located between the Dolomieu crater and the roof of the shallow magma reservoir (Fig. 12c) as a possible eruption trigger, as suggested by Lénat et al. (2011). Expansion of the hydrothermal system is due to inputs of heat and fluids from the magma reservoir or deeper and pressurization is favored by its sealing (because of mineral precipitation; lava accumulation at the volcano top). Related to this point, Froger et al., (2015) suggest that PdF hydrothermal system (and its potential sealing as well) was largely disrupted during the 2007 caldera collapse. In Lénat's model, thermal expansion of heated geothermal fluids induce rock~~

fracturing by pore pressure increase. Hydrothermal fracturing would cause transient decompression of the magma reservoir, thus triggering vesiculation and starting magma ascent process. However, we found no evidences of new inputs of magma or fluids in the 2014 reservoir, that would have induced the pressurization of the hydrothermal system. So, in our model the combination of change of stress field (deep input) and the physical zonation of the shallow magma reservoir promote the second boiling that enhanced the foam accumulation. Our dataset permits us to propose that the 2014 eruption was fed by a physically zoned magma reservoir with the lighter crystal poor magma erupted first (and possibly located in the upper part of the storage system) that ascends faster and feed the more energetic phase, the fountaining (Fig. 12d). This lighter magma is not more evolved than the spiny one (same bulk compositions) and it is not necessarily richer in dissolved volatile amounts; it is just poor in crystal. We conclude that the second boiling is responsible of the extraction of bubble rich melt from a crystal rich network. This last one will represent the main volume of erupted lava. eEruption. The occurrence of a hydrous almost pure melt at shallow depth permitted its fast vesiculation upon ascent towards the surface. In turn, fast ascent of the foam (Fig. 12d) hindered its crystallization and preserved high number of vesicles, high vesicularity and it is only little modified by post fragmentation expansion. Decrease in initial overpressure translated in a progressive decrease in magma ascent rate and output rate (e.g. Coppola et al., 2017 and references therein) and a temporal transition from Hawaiian activity to Strombolian activity (Fig. 12 d). Nucleation of microcrysts was enhanced in melt ascending with lower speed and in turn this syn-eruptive crystallization favoured bubble connectivity/permeability in the ascending magma, even for magma at low vesicularity and was mostly controlled by syneruptive degassing. The larger volume (dense lava) corresponds to highly-crystallized and degassed magma already in the reservoir, and experienced a slower ascent in the dyke and even further micro-crystallisation during its subaerial emplacement.

The texture of the products allowed us to follow the dynamic evolution of the system in space, (from smooth fluidal scoria emitted from rapid jet of lava fromat the fractures, to a more stable activity at the Main Vvent, and in time. At the Main Vent, in fact, we observed the transition ) and in time, at the Main vent (fromfrom 
[revised manuscript text omitted]

---

## Author Comment (AC2) · 28 Nov 2017

Dear Amanda,

Thanks a lot for your comments and corrections. Please find here our detailed list of responses and the manuscript attached with all the corrections and the new figures

"It would be interesting to see how that permeability data adds to the interpretation of eruptive activity. Also, I think it would also be a more intuitive measurement than isolated/connected vesicularity. "

We added now the permeability data in the Supplementary material (Table S3), as you suggested, and we added a new Figure 6d. We didn't add the permeability in the submitted version because we had a limited dataset of only 6 measurements. Moreover, we checked them again and we had to remove one, because we found some epoxy that infiltrated the sample during its preparation. The clasts are quite fragile for these measurements, so we lost a lot of samples. However, the measurements on 2014 clasts performed on one spiny opaque, one spiny glassy, one fluidal clast and two golden pumice (all collected at the Main Vent site) are consistent with the data that we obtained for other PdF eruptions (2015-2016 and 2017, that I am not inserting in this paper because they are part of another project of our PhD student). In the diagram in Figure 6d, we added also the data from February 2015, for comparison (that is: three samples, two fluidal and 1 spiny fragments). The raw data can be found also in the DynVolc database (2017). As you can see from Figure 6d, all the clasts, fluidal, golden or spiny scoria, are quite permeable, independent on their vesicularity, crystal content or of the presence of isolated vesicles. This is in agreement with our interpretation that magma degasses during its ascent in the conduit and that promotes microlite nucleation before magma fragmentation (see also Di Muro et al. 2015 with the Pele's hairs ad tears samples for the three 2008 eruptions). Moreover, we often find that the spiny clasts (especially the opaque ones) are slightly less permeable than the golden and fluidal ones, but not as impermeable as you would expect by their low vesicularity. In conclusion, we completely agree with the findings of the publications that you listed. We discuss these findings in the results and discussion sections and we added the references that you suggested. We can see that i) the crystals lower the percolation threshold and stabilize permeable pathways and ii) permeability develops during vesiculation through bubble coalescence, which allows efficient volatile transport through connected pathways and relieves overpressure (Lindoo et al. 2017). We also agree that pervasive crystal networks also deform bubbles and therefore enhance outgassing (Oppenheimer et al., 2015). Based on Saar et al., (2001) you suggest that crystals should start to affect the behavior of the exsolved volatile phase when they approach 20 vol% (Lindoo et al; 2017). In our dataset, apart from the golden and part of fluidal, all the other clasts do have microlites >20% (lines 845-854). Our data completely agree that slow decompression rate allows more time for degassing-induced crystallization, which lowers the vesicularity at which bubbles connect (lines854-857 and more). However, in the crystal-poor fragments we do NOT see a decrease in (i) vesicularity, (ii) number of vesicles, and (iii) permeability (see discussion from lines 858-874). We do not have evidence from the natural samples that the crystal-poor fragments remain impermeable after quenching, due to melt relaxes and pathways closure, as revealed by experiments (Lindoo et al., 2016). The only evidence of this relaxation process could be the high percentage of isolated vesicles in the fluidal and golden fragments due to rapid re-annealing of pore throats between connected bubbles due to short melt relaxation times (Lindoo et al. 2016). However, as explained later to the third review, we doubt about these relaxation process. It would be great to see these samples in 3D, because it is difficult in 2D to say which the isolated vesicles are. What we see in the crystal-poor samples is that permeability increases rapidly once the percolation threshold has been reached, and efficient degassing prevents bubble volumes from expanding past the percolation threshold (Rust and Cashman 2011). In our samples, in fact, we do not have strong evidences of expansions and coalescence.

"I would like to see a more detailed discussion of the crystallinity data given the large impact of crystals on bubble deformation, connectivity/permeability (Spina et al. 2016, Lindoo et al. 2017), volatile distribution in the conduit (Parmigiani et al. 2011, Parmigiani et al. 2016), and ultimately eruptive style."

We agree with these comments and we added all the crystal percentage expressed as total crystallinity in 3D, using the Higgins program and CSDcorrections. The corrected crystallinity for the porosity for mesocrysts and microcrysts percentage (found with Higgins), for plg and cpx are reported now as ranges in Figure 4 and we added the data in Table 3, for each sample, and we deleted the isolated-vesicle column. We expanded the methodology (lines 280-286), results (lines 460-468) and discussion (lines 878-897) paragraphs.

"I did not come away with a clear understanding of their first (i: why was such a small volume of magma erupted instead of forming an intrusion) or fifth (v: What was the time and space evolution of the eruptive event) objectives." From the comments of all reviewers (Amanda, Madison and the unknown reviewer), it is clear that we had to improve the discussion paragraph. We agree that these two points (and other as well, like the trigger mechanisms) needed to be reframed and expanded. In terms of small eruption versus intrusion and precursor intensity and duration, we summarize here below our reasoning. Let's start to speak about the trigger mechanisms of PdF eruption and the constraints provided by our dataset. An eruption of a shallow system like PdF can be triggered by either internal processes or external processes or a combination of both External processes: (i) Shallow magma reservoir pressurization because of volume changes related to either new magma input and/or to fluid inputs (CO2-rich fluids) from deeper magmatic levels. (ii) Heating and enhanced convection of the shallow magma reservoir (energy transfer without fluid or mass transfer). (iii) Pressurisation (volcano inflation) and/or depressurization (volcano deflation) of the hydrothermal system located between the Dolomieu crater and the roof of the shallow magma reservoir. Expansion of the hydrothermal system is due to inputs of heat and fluids from the magma reservoir or deeper and pressurization is favored by its sealing (because of mineral precipitation; lava accumulation at the volcano top). (iv) Deformation of the volcanic edifice and decompression of the magma reservoir and/or hydrothermal system due to flank sliding. (v) Deformation of the volcanic edifice due to deep magma transfers

Internal processes (vi) Accumulation of bubbles in magma recently emplaced in a shallow reservoir at low pressure. (vii) Rapid volatile exsolution (water-dominated fluids; second boiling) after slow magma cooling and extensive crystallization and evolution.

Process (i): Geochemical (bulk rock) and petrological (mineral composition and zoning) data permit to exclude the first hypothesis. The magma erupted in 2014 results to be one of the most evolved and cold magmas ever erupted at Piton de la Fournaise (Figs 8 and 10); it is very homogeneous (Fig. 7), minerals do not exhibit reverse zoning and their compositional evolution from phenocrysts to microlites record magma cooling and final degassing (new Figure 10b). Process (ii): is very slow because of slow heat diffusion and 2014 minerals do not record evidences of slow magma heating. About process (iii): The June 2014 eruption was preceded by a weak inflation only 11 days before the eruption. We attributed the summit cone inflation to the pressurization of a very shallow magmatic source (ca. 1.4-1.7 km below volcano summit) by Peltier et al., 2016. On one side, Froger et al., 2015 suggest that PdF hydrothermal system (and its potential sealing as well) was largely disrupted during the 2007 caldera collapse. Lénat et al. (2011) consider the hydrothermal system as a possible eruption trigger. In Lénat's model, thermal expansion of heated geothermal fluids induce rock fracturing by pore pressure increase. Hydrothermal fracturing would cause transient decompression of the magma reservoir, thus triggering vesiculation and starting magma ascent process. However, we found no evidences of new inputs of magma or fluids in the 2014 reservoir, that would have induced the pressurization of the hydrothermal system. Process (iv): finally, geodetic data show no evidence of flank sliding able to produce of the hydrothermal and magmatic system Process (v): The occurrence of deep (>10 km bsl) lateral magma transfer since March-April 2014 has been inferred by Boudoire et al., 2017 (GRL) on the basis of deep (mantle level) seismic swarms and soil $CO_2$ emissions on the distal western volcano flank. We suspect that these deep processes can have modified the shallower crustal stress field and favored magma vesiculation and eruption trigger. On one side, the 2014 eruption was the first of a long list of eruptions in the 2015-2017 period. On the other side, geophysical and geochemical data have permitted to track vertical magma and fluid transfer below the volcano summit in April 2015, that is about one year after the deep lateral magma transfer (Peltier et., al 2016). Deep processes cannot be detected by the OVPF geodetical network. The 11 days of weak summit volcano inflation, which preceded the 2014 eruption, possibly result from volatile exsolution and expansion of both the shallow magma reservoir and the hydrothermal system. Process (vi): Barometric data (Di Muro et al., 2014; 2016)

suggest that most magma reservoirs feeding the PdF eruptions are stored at shallow pressure (< 1-0.5 kbar). Water exsolution is strongly favored by low pressure and accelerates during magma transfer towards the surface. The 2014 magma erupted after an unusually long phase of quiescence and is chemically evolved, and records extensive magma cooling and crystallization. Extensive crystallization, clearly recorded in 2014 lava (we added the lava data in Figure 4 and in the text), can drive melt migration, volatile concentration and create the conditions favorable to second boiling (Tait et al., 1989). However, we suspect that stress field change related to deep magma transfer induced second boiling and rapid magma vesiculation and expansion, because the 2014 event represented the first of a long series of eruptions, whose magmas became progressively less evolved in time (Coppola et al., 2017). Process (vii):We stress that second boiling is possibly not the only process driving magma foaming in the reservoir. This is because we observe similar textural heterogeneities in 2009 and 2014 eruptive products, which represent the two chemical end-members of recent PdF activity.

Therefore, we suspect that magma storage at shallow depth favors volatile (mostly $H_2O$) exsolution at several steps during magma ponding, cooling and evolution and promotes fast magma response to external triggers (stress field changes; magma inputs). Without this external input we believe that the little reservoir of 2014 would have evolved in an intrusion (see the pervasive crystallization of the lava, one of the densest emitted from 2014 to 2017, see Figure 13 in Harris et al; 2017 + unpublished data, below)

See new Conclusions paragraph

In term of space and time evolution of eruptive dynamics and textures, we agree with Madison that we need to add a scheme to summarize our conclusions. We provide the new Figure 12

"The authors employ circularity to characterize different clast types. First, how many particles of each typology were measured using the Morphologi G3? I would also suggest the use of at least three shape descriptors, as recommended by Liu et al. 2015, to fully describe particle morphologies. Currently, the use of a shape descriptor in the interpretation of the eruptive products comes across as an afterthought. Because circularity is not really utilized in the description/interpretation of the products, the section could just be removed. "

We removed the Morphology G3 data, because these data are not so relevant for the whole story of the paper. In the submitted paper, we just wanted to show the methodology and the potential of these analyses. The instrument can measure up to 2000 fragments, so it is very robust in terms of statistics.

"I do think it would be interesting to see if other shape descriptors (such as solidity and convexity) may better describe the relationships between particle shapes and eruption styles."

We removed these data, but we completely agree with Amanda and we will use her precious comments for another paper (in progress) in which we discuss the ash dataset.

"Why were only 25 clasts from the Western Fracture analyzed versus 146 from the Main Vent? I'm speculative whether the number of clasts accurately samples the Western Fracture explosion. "

We explained the sampling strategy better (see from line 199) in the paper now, in order to clarify all these points and we moved the sampling strategy in the Methodology section. I would like to outline here, however, that three days after the eruption, when the deposits were still hot, difficult to reach etc, the strategy of the OVPF people was to collect as many samples as they could to be representative of the deposits. We stressed in the paper that the deposit from the Western Fracture were formed by scattered bombs and lapilli scorias, all fluidal and we believe our sampling is representative. To show the deposit at the Western Fracture we readjusted Figure 3c

"Also, I do not find it clear how clasts were picked for analyzing vesicle size distributions.

I find the Spiny Glass and Golden pumice density distributions to be slightly bimodal (Fig. 6c). Do the stars in Fig. 6 denote the mode determined for each component? This should be noted in the figure caption as well."

We explain it better in the text (lines 286-295), in the caption of Figure 6 and in the Figure 6. The choice of the clasts was made mostly on the typologies, rather than on each density distribution, in order to avoid the analysis of clasts with transitional characteristics. For example, two golden pumice fragments were selected from the largest clasts that were the less dense and didn't break, even if the values in vesicularity were similar. A larger number of fluidal fragments were chosen (even if the density distribution was unimodal) because this typology of clasts was the most abundant and was emitted all along the active fracture, so we did our best in order to study products representative of the Western Fracture, the Upper Fracture and the Main Vent activities. Only one spiny glassy and one spiny opaque were selected, because they were emitted only at the Main Vent.

"I do not see a table that includes all of the crystallinity data (vol.Crystallinity data could be inserted into Table 3 in the connected vesicle or isolated vesicle column, as it's not necessary to have both (connected/isolated) listed. There is some description in the results (phases present), but I find it difficult to follow without a table to reference/compare. "

The total crystallinity corrected for the porosity, and mesocrysts and microcrysts percentage (found using Higgins software) for plg and cpx are reported now as ranges in Figure 4 and we added for each sample the data in Table 3 as well.

"I would also be interested to see the phase abundances and aspect ratios. The amount of crystals (specifically high aspect ratio plagioclase) coupled with the vesicularity data, may give more insight into efficient vs. inefficient degassing in the different typologies (see Shea et al. 2017). The amount of crystals (depending on the aspect ratio) will influence degassing as well (Lindoo et al. 2017). "

Yes, we agree with these observations and actually the microcrysts that formed in the conduit are mostly sodic plagioclases; their abundance increases from the golden (high vesicularity and high vesicle number density) to the spiny (lower vesicularity coupled with lower vesicle number density); therefore, the increase in plg of microlites does favour an efficient degassing in the relatively crystal-rich magma, because of their low wet angles that favor degassing against nucleation (Shea 2017). We added and discussed these data in the text and we added Figure 10a.

"I would ask the authors to also consider the effect of crystals on the permeability of the "degassed, cooler reservoir" along with their interpretation of reservoir tapping. Crystals increasing bubble connectivity/permeability of the reservoir alone may contribute to extensive degassing and shifts in eruptive style."

Yes we do agree that syn-eruptive degassing is favored by bubble connectivity/permeability in the ascending magma, enhanced by syn-eruptive crystallization in the conduit (especially microcrysts of plg), even for magma at low vesicularity. However, we also support the idea of magma stratification in the reservoir. This stratification is probably mechanical and enhanced by melt-crystal separation during second boiling. From the data is evident that we have a melt (represented by golden and large part of the fluidal fragments) with scarce crystals. This crystal poor melt represents only a small volume and is associated (and followed in time by) with the main volume of magma that contains a larger amount of mesocrysts and forms the main volume of the lava flows. These larger crystals, absent in the golden, scares in the fluidal and more abundant in the spiny and lava consist in an equal percentage of plg and cpx and minor olivine, and they form in the reservoir, as shown by their different composition in respect to the microcrysts counterparts (we added a graph of plagioclase compositions, in Fig. 10) that formed in the conduit. However, a large amount of microcrysts in lava formed in the reservoir as well (as shown by their compositions, see Figure 10a). So, we have a range of crystallization conditions. The fact that the lighter plg are not concentrated in the upper portion can be due to the fact that often they are locked in clusters with the cpx and/or trapped by the microcrysts that in lava formed in the reservoir (see Figure 10a). Our dataset permits us to propose that the 2014 eruption was fed by a physically zoned magma reservoir with the lighter crystal poor magma erupted first (and possibly located in the upper part of the storage system) that ascends faster and feed the more energetic phase, the fountaining. This lighter magma is not more evolved than the spiny one (same bulk compositions) and it is not necessarily richer in dissolved volatile amounts; it is just poor in crystal. We conclude that the second boiling is responsible of the extraction of bubble rich melt from a crystal rich network. This last one will represent the main volume of erupted lava. Fast ascent of the foam hinders its crystallization and preserve high number of vesicles, high vesicularity and it is only little modified by post-fragmentation expansion. Decrease in initial overpressure translates in a progressive decrease in magma ascent rate and output rate (e.g. Coppola et al., 2017 and references therein). Nucleation of microcrysts is enhanced in melt ascending with lower speed and is mostly related to syneruptive degassing (for the spiny). The larger volume (dense lava) corresponds to crystallized and less vesiculated magma which experiences a slow ascent in the dyke and even further micro-crystallization during its subaerial emplacement.

"Section 5.2 might benefit from subsections or reorganization, perhaps divided by the different typologies, sampling area, or interpretation and comparison to other studies. There is a lot of information presented and comparison to other studies."

We did it, also following Madison suggestions. See the new 5.2 paragraph subdivided now in four subsections: 1)Background on the texture of clasts from Hawaiian and Strombolian activities; 2) The four typologies of clasts and their distribution in space and in time in the 2014 eruption at Pd; 3) Degassing-driven versus cooling-driven crystallization 4) Textural syn-eruptive versus post fragmentation modifications

"Lines 99-105 – Reassess/reorganize the questions posed here. There are 5 questions listed with (iv) and (v) attached to (iii). I suggest separating each question with a paragraph or do not separate them. Also, I do not think questions (i) or (v) were addressed in the discussion/conclusions section." We addressed these two questions now, see the new conclusions

"Table 3 does not need both connected vesicularity and isolated vesicularity listed." We deleted a column and we added the crystals parameters

"Figure 5c needs a more descriptive caption. I'm not sure what I and II refer to or the arrows (the clasts pictured?). I think the caption only describes one of the two graphs?" Figure 5c was removed

"Figure 6c – please clarify the meaning of the star symbols" We clarify it in the caption and we improved the figure

"Figure 11 could be redrafted to provide more clarity to the reader. I would move the references to the figure caption to make room for an inset similar to Stovall et al. 2011 to help the reader interpret trends." We did it, see new Figure 11

"89 references - I think the number of the references could be reduced."

I don't think that in a paper where we integrated field, physical textural, petrological and geochemical analyses we can reduce the references. With the corrections and the suggestion from the three reviewers we actually increased the references list. If the journal does not impose references limitations we are happy to try to acknowledge all the relevant contributions

"Is the amount and quality of supplementary material appropriate? Yes. Some formatting issues with supplementary tables." Yes, we readjust all the tables.

"Line 111 – I would recommend removing this final sentence. The authors make it clear earlier in the introduction the importance of the multi-disciplinary approach." Yes, we removed it

"Line 218 – Should reference Fig. 3e not 3f?" Yes

"Line 310 – Combine the two sentences with the rest of the paragraph." Yes, corrected

"Line 331 – Should reference Figure 3b?" Yes, thank you

"Line 510 – subscript "wr" in MgOwr." Yes, corrected

"Line 645 – reference numbers for comparison to Houghton et al. 2016. General - Vg/Vl should be Vg/Vl. Subscript "v" in Nv." We corrected it

"Figure 1 – An inset map of Reunion Island would be helpful. (1c) is very dark/difficult to see." Done

"Figure 3c – The pictures are so small it is difficult to see." We changed a lot in Figure 3, to better clarify the nature of the deposits

"Figure 3e – 2010, Fountaining is spelled wrong." We corrected it

"Figure 10 – Inconsistent figure formatting. Thick axes lines and bold axes values" We corrected it

"Missing or incorrect references:" Bombrun et al. 2015 (line 703) Added

Di Muro et al. 2012 (line 126) Deleted

Gurioli et al. 2008 (line 633) Added

Hammer et al. 1999 (line 750) Added

Inman 1952 (line 223) Added

Liuzzo et al. 2015 (line 134) Added

Morandi 2015 (line 72) Corrected

Line 58 – Taddeucci misspelled Done

Line 60 – Extra "and" Corrected

Line 60 – Eychenne misspelled. Corrected

Line 61 – Should read "Leibrandt and Le Pennec, 2015". Corrected

Line 600 – references in italics. Corrected

Line 971 – Should read "Lange, R.A.,. . ." Corrected

Line 1016/1020 – reference chronology inconsistent. Corrected

Line 1023 – delete "a" from reference. Corrected

References cited:

Lindoo, A., Larsen, J. F., Cashman, K. V., and Oppenheimer, J., 2017, Crystal controls on permeability development and degassing in basaltic andesite magma: Geology, 45(9), p. 831-834.

Liu, E. J., Cashman, K. V., Rust, A. C., 2015, Optimising shape analysis to quantify volcanic ash morphology: GeoResJ, 8, p. 14-30.

Parmigiani, A., Huber, C., Bachmann, O., and Chopard, B., 2011, Pore-scale mass and reactant transport in multiphase porous media flows: Journal of Fluid Mechanics, v. 686, p. 40-76.

Parmigiani, A., Faroughi, S., Huber, C., Bachmann, O., Su, Y, 2016, Bubble accumulation and its role in the evolution of magma reservoirs in the upper crust: Nature, 532,p. 492-494.

Spina, L., Cimarelli, C., Scheu, B., Di Genova, D., and Dingwell, D. B., 2016, On the slow decompressive response of volatile- and crystal-bearing magmas: An analogue experimental investigation: Earth and Planetary Science Letters, v. 433, p. 44-53.

We checked these papers and we added a few references from the list above and other useful ones founded in the papers

Please also note the supplement to this comment:
https://www.solid-earth-discuss.net/se-2017-99/se-2017-99-AC2-supplement.pdf

[Figure]

[Figure]

**Fig. 1.** Table S3

**Supplement:**

[revised manuscript text omitted]
 (4.6 years) and was preceded by only a few days of weak and short geophysical seismic, geodetic and geochemical precursors (Peltier et al., 2016 and Fig. 1d). This multidisciplinary approach provides new constraints on the mechanisms triggering such short-lived, small but potentially hazardous eruptions (STIAMO RIPETENDO TRE VOLTE GLI STESSI CONCETTI. RIASSUMI E TAGLIA).supposewas sittingcollingDuring the inflation period (b,itself allows exsolution, and magma with crystallization in situ of thelites, as observed in the lava sampleFig.This process favours a physical zonation of the shallow reservoir. Therefore, magma storage at shallow depth favours volatile (mostly H₂O) exsolution at several steps during magma ponding, cooling and evolution (Fig. 12b)~~

The occurrence of deep (>10 km bsl) lateral magma transfer since March-April 2014 has been inferred by Boudoire et al., (2017) on the basis of deep (mantle level) seismic swarms and increase in soil CO2 emissions on the distal western volcano flank. The incipit of magma transfer towards shallower crustal levels is potentially recorded by subtle volcano inflation about 11 days before the June 2014 eruptions (Figs. 1d and 12c). We suspect that these deep processes can have progressively modified the shallow crustal stress field and favoured magma vesiculation and melt-crystal separation. Second boiling could thus have over-pressured the shallow seated reservoir and triggered magma ascent  (Fig. 12c).

Without this deep magma transfers  we believe that the small reservoir activated in 2014 would have cooled down completely to form  an intrusion (as suggested by the pervasive crystallization of the lava, one of the densest emitted from 2014 to 2017).  The 2014 event represented instead the first of a long series of eruptions, whose magmas became progressively less evolved in time (Coppola et al., 2017). In this scenario the trigger mechanisms of 2014 activity are both internal and external in the sense that the small shallow reservoir hosting cooled magmas permitted to create the conditions favourable to a second boiling (Fig. 12c, and Tait et al., 1989). The second boiling was likely trigger by an almost undetectable stress field change, and was favoured by the shallow storage pressure of the magma (Fig. 12c) that promotes fast water exsolution and rapid magma response to external triggers. The second boiling possibly contributed to  the inflation registered 11 days before the eruption at 1.4-1.7 km (Fig. 12c) both by magma expansion and  transfer of hot fluids to the hydrothermal system (Lénat et al., 2011).

Our data permit to exclude (i) new magma input and/or to fluid inputs (CO2-rich fluids) from deep magmatic levels to trigger the June 2014 eruption. We also exclude (ii) heating and enhanced convection of the shallow magma reservoir (due to heat diffusion without fluid or mass transfer), because this process is very slow . Furthermore,   2014 minerals do not record evidences of  magma heating. We can exclude equally  (iii) deformation of the volcanic edifice and decompression of the magma reservoir and/or hydrothermal system due to flank sliding because geodetic data show no evidence of flank sliding able to produce stress change in  the hydrothermal and magmatic system. ~~Geochemical (bulk rock) and petrological (mineral composition and zoning) data, permit to exclude this hypothesis. The magma erupted in 2014 results to be one of the most evolved and cold magmas ever erupted at Piton de la Fournaise (Figs 8 and 10b); it is very homogeneous (Fig. 7), minerals do not exhibit reverse zoning and their compositional evolution from phenocrysts to microlites record magma cooling and final degassing (new Figure 10a).cannot bedetected by the OVPF geodetical~~ detect for any monitoring network.

We conclude that the overpressure, caused by the second boiling, triggered the ~~The 11 days of weak summit volcano inflation, which preceded the 2014 eruption, possibly result from volatile exsolution and expansion of both the shallow magma reservoir and the hydrothermal system (Fig. 12c). We also exclude (ii) heating and enhanced convection of the shallow magma reservoir (due to energy transfer without fluid or mass transfer, because rocess is very slow because of slow heat diffusion and 2014 minerals do not record evidences of slow magma heating. We can exclude equally (iii) deformation of the volcanic edifice and decompression of the magma reservoir and/or hydrothermal system due to flank sliding because geodetic data show no evidence of flank sliding able to produce decompression of the hydrothermal and magmatic system. However, it is necessary to discuss the pressurisation (volcano inflation) and/or depressurization (volcano deflation) of the (iv) hydrothermal system located between the Dolomieu crater and the roof of the shallow magma reservoir (Fig. 12c) as a possible eruption trigger, as suggested by Lénat et al. (2011). Expansion of the hydrothermal system is due to inputs of heat and fluids from the magma reservoir or deeper and pressurization is favored by its sealing (because of mineral precipitation; lava accumulation at the volcano top). Related to this point, Froger et al., (2015) suggest that PdF hydrothermal system (and its potential sealing as well) was largely disrupted during the 2007 caldera collapse. In Lénat's model, thermal expansion of heated geothermal fluids induce rock~~

fracturing by pore pressure increase. Hydrothermal fracturing would cause transient decompression of the magma reservoir, thus triggering vesiculation and starting magma ascent process. However, we found no evidences of new inputs of magma or fluids in the 2014 reservoir, that would have induced the pressurization of the hydrothermal system. So, in our model the combination of change of stress field (deep input) and the physical zonation of the shallow magma reservoir promote the second boiling that enhanced the foam accumulation. Our dataset permits us to propose that the 2014 eruption was fed by a physically zoned magma reservoir with the lighter crystal poor magma erupted first (and possibly located in the upper part of the storage system) that ascends faster and feed the more energetic phase, the fountaining (Fig. 12d). This lighter magma is not more evolved than the spiny one (same bulk compositions) and it is not necessarily richer in dissolved volatile amounts; it is just poor in crystal. We conclude that the second boiling is responsible of the extraction of bubble rich melt from a crystal rich network. This last one will represent the main volume of erupted lava. eEruption. The occurrence of a hydrous almost pure melt at shallow depth permitted its fast vesiculation upon ascent towards the surface. In turn, fast ascent of the foam (Fig. 12d) hindered its crystallization and preserved high number of vesicles, high vesicularity and it is only little modified by post fragmentation expansion. Decrease in initial overpressure translated in a progressive decrease in magma ascent rate and output rate (e.g. Coppola et al., 2017 and references therein) and a temporal transition from Hawaiian activity to Strombolian activity (Fig. 12 d). Nucleation of microcrysts was enhanced in melt ascending with lower speed and in turn this syn-eruptive crystallization favoured bubble connectivity/permeability in the ascending magma, even for magma at low vesicularity and was mostly controlled by syneruptive degassing. The larger volume (dense lava) corresponds to highly-crystallized and degassed magma already in the reservoir, and experienced a slower ascent in the dyke and even further micro-crystallisation during its subaerial emplacement.

The texture of the products allowed us to follow the dynamic evolution of the system in space, (from smooth fluidal scoria emitted from rapid jet of lava fromat the fractures, to a more stable activity at the Main Vvent, and in time. At the Main Vent, in fact, we observed the transition ) and in time, at the Main vent (fromfrom 
[revised manuscript text omitted]

---

## Author Comment (AC3) · 28 Nov 2017

Please find here our detailed list of responses. A few explanations are reported in Amanda and Madison responses. Attached is the manuscript with all the corrections and the new figures

"the eruption was triggered by pressurization due to bubble accumulation in a shallow magma reservoir, as opposed to magma chamber cooling or a new batch of magma flux into the reservoir In general, the outcomes of this study are not transparent with regards to the questions addressed in Lines 99-105. It seems that the paper includes a number of hypotheses while the validity of those are inadequately presented. I suggest

either rephrasing parts of the manuscript as applicable or provide some quantitative analysis in support of some of the conclusions. Also, I find a number of parameters in the figures are not defined properly in the text or in figure captions, making it difficult to follow at places. I hope the authors will find the following specific comments useful for further improvements. "

We added more explanations and data to support our hypothesis and we corrected all the Figures and captions

"Lines 801-807: Following my general remark, several possible scenarios have been proposed here without a reasonable justification. For example, "we found that this kind of eruption can be triggered solely by bubble accumulation and source pressurization" – The relationship of bubbles, pressure build-up and its extent for eruption triggering have not been demonstrated in this study. "

We explained all of this in Amanda and Madison responses, and we added the explanation in the text

"Lines 798-799: It seems like the hypotheses of a shallow magma reservoir and its pressurization are mostly driven by the weak and short geophysical precursors, which is not the focus of this study. In other words, the contribution of geochemical/petrological monitoring independent of geophysical signals – for tracking eruption triggers and dynamics are not transparent. "

As you can see from the previous explanations, the integration of the geophysical and the geochemical/petrological data allowed us to obtain the whole picture. Based on our findings we propose a scenario in which the trigger mechanisms of 2014 activity are both internal and external in the sense that the small shallow reservoir hosting cooled magmas permitted to create the conditions favourable to a second boiling. The second boiling was likely trigger by an almost undetectable stress field change, and was favoured by the shallow storage pressure of the magma (Fig. 12c) that promotes fast water exsolution and rapid magma response to external triggers. See the new

discussion and conclusions.

"Title: The title is too broad. Although it is catchy, but based on the previous two comments, neither the trigger nor the dynamics are adequately addressed in this study. "

We completely disagree and we leave this title, if the editor and the other authors agree.

"Lines 636-640 and 683-689: Isolated vesicles, also mentioned in some other parts of the manuscript, could simply be a result of post-coalescence surface tension forces, especially for low viscosity magmas due to relatively smaller viscous forces. Therefore it may not represent the low rate of deformation, and can even get overprinted during cooling of the pyroclasts. On the other hand, the presence of micro-crystals increase viscosity preserving the coalesced textures (see Moitra et al. 2013, Relating vesicle shapes in pyroclasts to eruption styles, Bull Volc, for a discussion), and therefore if syneruptive, it may not represent cooled magma and longer residence times. Therefore the implications/conclusions need to be more convincing, or a discussion on the various possibilities is required, also insightful, at the least."

Rapid re-annealing of pore throats between connected bubbles can happen due to short melt relaxation times (Lindoo et al; 2016). This phenomenology can explain the high amount of isolated vesicles in the fountaining samples. However, if you look at the vesicle distributions, they are almost perfect Gaussian curves, so it seems that if the relaxation process happens it just merged perfectly with the expected vesicle distribution. In contrast, you know well that secondary processes like coalescence and/or expansion (as we observe in the spiny) do not fit the curve. In the isolated vesicle rich samples, because of their high permeability, their high vesicularity and mostly their high number of vesicles, we do affirm that we have preserved the signature of the conduit before the explosion. We added this part in the discussion (from line 884)

"Figure 5c: There is no discussion on circularity? What about any other shape factor? What do they mean? " We removed these data

"Figure 6d: There are a number of solid lines drawn without a proper caption. Which diagonal line (and therefore the samples) represents equality and what are those various percentages? " We added explanation

Technical corrections:

"Line 75: space between grain and size" Done

"Line 81: weird spacing" Done "Line 189: Mm3 could be defined in line 188, where million m3 is first introduced, for better" clarity. Done

"Figure 1c caption: locations instead of location " Done

"Figure 4 caption: %cry and not %Cry to be consistent " Done

"Figure 9 – 'T' in FeOT should be in subscript " Done

"The name/expression "Piton de la Fournaise" is not consistent in the manuscript: 'La' is often used instead of 'la'" Corrected in captions text and references

"Figure subplots are sometimes labeled by capital letters, sometimes by small letters" Corrected

Please also note the supplement to this comment:
https://www.solid-earth-discuss.net/se-2017-99/se-2017-99-AC3-supplement.pdf

---

## Author Response (AR2)

[revised manuscript text omitted]

**Lucia Gurioli**
Physicienne

Laboratoire Magmas et Volcans
Université Clermont Auvergne –
CNRS - IRD, OPGC
Campus Universitaire des Cézeaux
Avenue Blaise Pascal
TSA 60026 - CS 60026
63178 AUBIERE Cedex
Tel: +33 (0)473346782
Fax: +33 (0)473346744
l.gurioli@opgc.univ-bpclermont.fr
WEB: http://www.opgc.univ-bpclermont.fr/

Date:13/12/2017

Dear Mike,

Please, find the revised version of the paper where we left all the corrections highlighted and the clean version. Figures and Supporting Material are the same.

We are very glad that you appreciated our corrections. We tried to do our best.

We went through your minor corrections carefully and we checked all the text and captions again.

Below you can follow our corrections:

Line 26: I suggest that you change "lightest" to "least dense". "Light" can also refer to colour.
Yes, we changed it everywhere in the text. Very good point.
Line 34: "led".
Changed
Line 70: "…the spatial and temporal evolution of magma…"
Changed
Line 78: References should be in chronological order.
I made all the references lists in chronological order
Line 108: "endogenic" rather than "intruded"?
Yes, I agree to avoid genetic interpretation
Line 202 and elsewhere: "PdF", no?
Yes, I changed everywhere
Line 225: "Sampling".
Corrected
Line 289: Should permeability not be mentioned in the subheading?
Added
Line 312: You should mention that the measurements were performed at atmospheric pressure (i.e. without confining pressure). You should also mention the pore fluid you used and whether/how you checked for turbulent flow (the Forchheimer effect). Please provide more details here.

I added more details. Could you check it please? Line 283

Line 315: Do not abbreviate the names of the minerals.

Yes, corrected everywhere

Line 458 and elsewhere: "Light" can also refer to colour. "Low-density" is a better descriptor, in my opinion.

We completely agree

Line 504: "Clusters".

Corrected

Line 506: Rephrase to avoid "picture".

We rephrased it

Line 524: "percent", not "percentage".

Corrected

Lines 527 and elsewhere: Are you referring here to a percentage or percentage points? If you're talking about percentage points, perhaps it's best to stick with "30 vol.%".

Yes, corrected everywhere in vol. %

Line 530: This is the results section. Since you now show values of permeability, there should be at least a couple of sentences describing the data.

I added more details. Could you check it please? Line 486

Line 668: Remove comma.

OK

Line 727: "experienced". In fact, please reword this sentence to improve clarity.

I reword it

Line 806: Please reword the sentence starting "The proportion…"

I reword it

Line 852 and elsewhere: Do not abbreviate the names of the minerals.

Corrected everywhere

Line 853: "indicates".

OK

Line 889: References for the published expansion signatures?

I reword it because the references are reported earlier and we wanted to refer to the evidence of the expansion in the graph of figure 11; see line 838

Line 916: Remove comma.

OK

Line 924: "and".

OK

Line 926: Impermeable means that fluid cannot pass. Please rephrase this sentence to use "low-permeability".

I reword it

Line 948: "pathway".

OK

Line 948: Do the authors mean "very low permeability" or "impermeable" here?

Actually the authors speak about a unmeasurable permeability and Amanda in her paper reports "impermeable samples" in Fig 1 (Lindoo et al; 2017). However, I added "almost" impermeable

Line 1041: "feeds".

OK

Line 1044: "crystals".

OK

Yes, we added to the previous paragraph

As I said, I read it carefully, my first two co-authors did the same, but no-one of us is English. We hope that we found all the errors, or at least all the inconsistency; see other corrections that we added

Figure 5: The figure still contains the Morphology G3 results. I suggest that the authors double-check all the figures and figure captions.
I fixed it

Sincerely

Lucia and co-authors